# VERIFYING GNNS WITH READOUT IS INTRACTABLE

## ABSTRACT

We introduce a logical language for reasoning about quantized aggregate-combine graph neural networks with global readout (ACR-GNNs). We provide a logical characterization and use it to prove that verification tasks for quantized GNNs with readout are (co)NEXPTIME-complete. This result implies that the verification of quantized GNNs is computationally intractable, prompting substantial research efforts toward ensuring the safety of GNN-based systems. We also experimentally demonstrate that quantized ACR-GNN models are lightweight while maintaining good accuracy and generalization capabilities with respect to non-quantized models.

## 1 INTRODUCTION

Graph neural networks (GNNs) are models used for classification and regression tasks on graph-structured data, including node-level and graph-level tasks. GNNs find applications in recommendation systems in social networks (Salamat et al., 2021), knowledge graphs (Ye et al., 2022), chemistry (Reiser et al., 2022), drug discovery (Xiong et al., 2021), etc. Like several other machine learning models, GNNs are difficult to interpret, understand, and verify. This poses a significant issue for their adoption, morally and legally, with the enforcement of regulatory policies such as the EU AI Act (European Parliament, 2024). Previous works lay the foundation for analyzing them using formal logic, as seen in Barceló et al. (2020), Nunn et al. (2024), or Benedikt et al. (2025). However, many of these approaches consider *idealized* GNNs in which numbers are either arbitrarily large integers or rationals, whereas in real-world implementations, GNNs are *quantized*; numbers are represented as Standard IEEE 754 64-bit floats, INT8, or FP8 (Micikevicius et al., 2022). Verification of quantized GNNs without global readout has been addressed by Sälzer et al. (2025). However, global readout is a crucial component of GNNs, particularly for graph classification (Xu et al., 2019).

In this paper, we consider three verification tasks for GNNs to check critical properties of graphs. Examples of such verification tasks are: "Is it the case that...

- (sufficiency) ... any power plant serving over a thousand users is classified by the GNN as essential?"
- (necessity) ... any power plant classified by the GNN as essential has at least 3 substations?"
- (consistency) ... a power plant classified by the GNN as essential can be a wind farm whose wind turbines have a total capacity of less than 12 MW?"

Verifying critical properties of GNNs helps ensure the safety of GNN-based systems, analogous to how correctness proofs are used to guarantee the safety of traditional computer programs Hoare (1969).

**Contribution.** The contribution is threefold. First, we show that verifying Aggregate-Combine Graph Neural Networks with global Readout (ACR-GNNs) is decidable and (co)NEXPTIME-complete, where NEXPTIME is the class of problems decidable by a non-deterministic algorithm running in exponential time in the size of its input. This contrasts with the PSPACE-completeness without global readout as demonstrated by Sälzer et al. (2025). In summary, this implies that global readout makes quantized GNN verification highly intractable[1]. To achieve this, we define the logic

---

[1]As pointed out in Jogl et al. (2023), any inference of a ACR-GNN can be simulated by a AC-GNN (Aggregate-Combine Graph Neural Networks *without* global Readout) where the global readout mechanism is replaced by an extra complete relation $V \times V$ where $V$ is the set of vertices of the input graph. So global

$q\mathcal{L}$, extending the one introduced by Sälzer et al. (2025) to capture global readout. It is expressive enough to capture quantized ACR-GNNs with arbitrary activation functions. Moreover, $q\mathcal{L}$ can serve as a flexible graph property specification language reminiscent of modal logics (Blackburn et al., 2001). The following example illustrates the use of $q\mathcal{L}$ for expressing graph properties.

**Example 1.** *Assume a class of knowledge graphs (KGs) representing communities of people and animals, where each node corresponds to an individual. Each individual can be Animal, Human, Leg, Fur, White, Black, etc. These concepts can be encoded with features $x_0, x_1, \ldots, x_5, \ldots$ respectively, taking values $0$ or $1$. Edges in a KG represent a generic 'has' relationship: a human can have an animal (pet); an animal can have a human (owner), a leg, a fur; a fur can have a color; etc. Suppose that $\mathcal{A}$ is a GNN processing those KGs and is trained to supposedly recognize dogs. We can verify that the nodes recognized by $\mathcal{A}$ are animals—arguably a critical property of the domain—by checking the validity (i.e., the non-satisfiability of the negation) of $\varphi_{\mathcal{A}} \to x_0 = 1$ where $\varphi_{\mathcal{A}}$ is a $q\mathcal{L}$-formula corresponding to $\mathcal{A}$'s computation, true in exactly the pointed graphs accepted by $\mathcal{A}$. Ideally, $\mathcal{A}$ should not overfit the concept of dog as a perfect prototypical animal. For instance, three-legged dogs do exist. We can verify that $\mathcal{A}$ lets it be a possibility by checking the satisfiability of the formula $\varphi_{\mathcal{A}} \wedge \Diamond^{\leq 3}(x_2 = 1)$. More complex $q\mathcal{L}$ formulas can be written to express graph properties to be evaluated against an ACR-GNN, which will be formalized later in Example 2: 1. Has a human owner, whose pets are all two-legged. 2. A human in a community that has more than twice as many animals as humans, and more than five animals without an owner. [2] 3. An animal in a community where some animals have both white and black fur.*

Firstly, to prove the (co)NEXPTIME upper bound, we reuse a concept from mathematical logic called Hintikka sets (Blackburn et al., 2001), which are complete sets of subformulas that can be true at a given vertex of a graph. We then introduce a quantized variant of Quantifier-Free Boolean Algebra with Presburger Arithmetic (QFBAPA) logic (Kuncak & Rinard, 2007), denoted by QFBAPA$_{\mathbb{K}}$, and prove that it is in NP, similar to the original QFBAPA on integers. We reduce the satisfiability problem of $q\mathcal{L}$ to that of QFBAPA$_{\mathbb{K}}$. On the other hand, (co)NEXPTIME-hardness is proven by reduction from a suitable tiling problem. Similarly, we also add global counting to the logic $K^{\sharp}$ previously introduced by Nunn et al. (2024). We show that it corresponds to AC-GNNs over $\mathbb{Z}$ with global readout and truncated ReLU activation functions. We prove that the satisfiability problem is NEXPTIME-complete, partially addressing a problem left open in the literature—that is, for the case of integer values and truncated ReLU activation functions (Benedikt et al., 2024; 2025). Details are in the appendix to keep the main text concise.

Secondly, since NEXPTIME is highly intractable—it is provably distinct from NP (Seiferas et al., 1978)—we relax the satisfiability problem of $q\mathcal{L}$ and ACR-GNNs, focusing on finding graph counterexamples where the number of vertices is bounded. This problem is NP-complete, and an implementation is provided as a proof of concept and a baseline for future research.

Finally, we experimentally demonstrate in Section 7 that the quantization of GNNs results in minimal accuracy degradation. Our results confirm that quantized models retain robust predictive performance while achieving substantial reductions in model size and inference cost. These findings demonstrate the practical viability of quantized ACR-GNNs for deployment in resource-constrained environments.

**Outline.** Section 3 defines the logic $q\mathcal{L}$, discusses its expressivity, and defines ACR-GNN verification tasks. Section 4 provides the (co)NEXPTIME membership for the satisfiability problem of $q\mathcal{L}$ and the ACR-GNN verification tasks. In Section 5, we show that these problems are (co)NEXPTIME-complete. Section 6 discusses the relaxation making the problems (co)NP-complete. Section 7 presents experimental results justifying the practical utility of quantized ACR-GNNs. We complement these results by reporting on the results of a prototype implementation (Appendix B.1) for verifying said GNNs in a bounded setting, highlighting the hardness of verifying quantized GNNs.

**Related work.** Barceló et al. (2020) showed that ACR-GNNs have the expressive power of FOC$_2$, that is, two-variable first-order logic with counting. Recent work has explored the logical expressiveness of GNN variants in more detail. Notably, Nunn et al. (2024) and Benedikt et al. (2024)

---

readout has no significant impact on the complexity of inference. On the contrary, verification tasks involve quantifications over the set of pointed graphs where this extra relation *must* be $V \times V$.

[2]Interestingly, $q\mathcal{L}$ goes beyond the capabilities of graded modal logic and even first-order logic (FOL). The property of Item 2 cannot be expressed in FOL.

introduced logics to exactly characterize the capabilities of different forms of GNNs. Similarly, Cucala & Grau (2024) analyzed Max-Sum-GNNs through the lens of Datalog. Sälzer et al. (2025) considered the expressivity of GNN with quantized parameters but without global readout. Ahvonen et al. (2024) offered several logical characterizations of recurrent GNNs over floats and real numbers.

On the verification side, Henzinger et al. (2021) studied the complexity of verification of quantized feedforward neural networks (FNNs), while Sälzer & Lange (2021); Sälzer & Lange (2023) investigated reachability and reasoning problems for general FNNs and GNNs. Approaches to verification are proposed via integer linear programming (ILP) by Huang et al. (2024) and Zhang et al. (2023), and via model checking by Sena et al. (2021).

From a logical perspective, reasoning over structures involving arithmetic constraints is closely tied to several well-studied logics. Relevant work includes Kuncak and Rinard's decision procedures for QFBAPA (Kuncak & Rinard (2007)), as well as further developments by Demri & Lugiez (2010), Baader et al. (2020), Bednarczyk et al. (2021), and Galliani et al. (2023). These logics form the basis for the characterizations established in Nunn et al. (2024); Benedikt et al. (2024).

Quantization techniques in neural networks exist, with surveys such as Gholami et al. (2022); Nagel et al. (2021) providing comprehensive overviews focused on maintaining model accuracy. Although most practical advancements target convolutional neural networks (CNNs), many of the underlying principles extend to GNNs as well (Zhou et al. (2020)). NVIDIA has demonstrated hardware-ready quantization strategies (Wu et al. (2020)), and frameworks like PyTorch (Ansel et al. (2024)) support both post-training quantization and quantization-aware training (QAT), the latter simulating quantization effects during training to improve low-precision performance. QAT has been particularly effective in closing the gap between quantized and full-precision models, especially for highly compressed or edge-deployed systems (Jacob et al. (2018)). In the context of GNNs, Tailor et al. (2021) proposed Degree-Quant, incorporating node degree information to mitigate quantization-related issues. Based on this, Zhu et al. (2023) introduced $A^2Q$, a mixed-precision framework that adapts bitwidths on graph topology to achieve high compression with minimal performance loss.

## 2 BACKGROUND

**Quantized numbers** Let $\mathbb{K}$ be a set of quantized numbers, and let $n$ denote the *bitwidth* of $\mathbb{K}$, which is the number of bits required to represent a number in $\mathbb{K}$. The bitwidth $n$ is written in unary; this is motivated by the fact that $n$ is small and that we would, in any case, need to allocate $n$-bit consecutive memory for storing a number. Formally, we consider a sequence $\mathbb{K}_1, \mathbb{K}_2, \ldots$ corresponding to bitwidths 1, 2, etc., but we retain the notation $\mathbb{K}$ for simplicity. We suppose that $\mathbb{K}$ saturates; for example, if $x \geq 0$ and $y \geq 0$, then $x + y \geq 0$ (i.e., there is no modulo behavior like in `int` in C, for instance). We assume that $1 \in \mathbb{K}$.

We consider Aggregate-Combine Graph Neural Networks with global Readout (ACR-GNNs), a standard class of message-passing GNNs (Barceló et al., 2020; Gilmer et al., 2017).

**ACR-GNNs** An ACR-GNN layer with input dimension $m$ and output dimension $m'$ is defined by a triple $(comb, agg, agg_\forall)$, where $comb : \mathbb{K}^{3m} \to \mathbb{K}^{m'}$ is a combination function, and $agg, agg_\forall$ are local and global aggregation functions that map multisets of vectors in $\mathbb{K}^m$ to a single vector in $\mathbb{K}^m$. An ACR-GNN consists of a sequence of layers $(\mathcal{L}^{(1)}, \ldots, \mathcal{L}^{(L)})$, followed by a classification function $cls : \mathbb{K}^m \to \{0, 1\}$. Given a graph $G = (V, E)$ with an initial vertex labeling $x_0 : V \to \{0, 1\}^k$, the state of a vertex $u$ at layer $i$ is defined recursively as:

$$x_i(u) = comb(x_{i-1}(u), \ agg(\{\{x_{i-1}(v) \mid uv \in E\}\}), \ agg_\forall(\{\{x_{i-1}(v) \mid v \in V\}\}))$$

The final output of the GNN for a pointed graph $(G, u)$ is $\mathcal{A}(G, u) = cls(x_L(u))$.

We concentrate on a specific subclass where both $agg$ and $agg_\forall$ perform summation over vectors, and where $comb(x, y, z) = \vec{\sigma}(xC + yA_1 + zA_2 + b)$, using $m \times m'$-matrices $C, A_1, A_2$ and a bias $1 \times m'$-vector $b$, all with entries from $\mathbb{K}$, with the componentwise application of an activation function $\sigma$. The classification function is a linear threshold: $cls(x) = \sum_i a_i x_i \geq 1$ with weights $a_i \in \mathbb{K}$. We assume that all arithmetic operations are executed according to the arithmetic related to $\mathbb{K}$. It is assumed that the context makes clear the $\mathbb{K}$ and arithmetic being used. We note $[[\mathcal{A}]]$ the set of pointed graphs $(G, u)$ such that $\mathcal{A}(G, u) = 1$. An ACR-GNN $\mathcal{A}$ is satisfiable if $[[\mathcal{A}]]$ is non-empty. The *satisfiability problem* for ACR-GNNs is: Given a ACR-GNN $\mathcal{A}$, decide whether $\mathcal{A}$ is satisfiable.

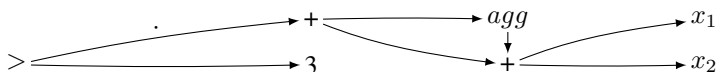

Figure 1: DAG data structure for the formula $agg(x_1 + x_2) + (x_1 + x_2) \geq 3$.

## 3 LOGIC $q\mathcal{L}$

We set up a logical framework called $q\mathcal{L}$ extending the logic introduced in Sälzer et al. (2025) with global aggregation: it is a *lingua franca* to represent GNN computations and properties on graphs.

**Syntax.** Let $F$ be a finite set of features and $\mathbb{K}$ be some finite-width arithmetic. We consider a set of *expressions* defined by the following grammar in Backus-Naur form:

$$\vartheta ::= c \mid x_i \mid \alpha(\vartheta) \mid agg(\vartheta) \mid agg_\forall(\vartheta) \mid \vartheta + \vartheta \mid c \times \vartheta$$

where $c$ is a number in $\mathbb{K}$, $x_i$ is a feature in $F$, $\alpha$ is a symbol for denoting the activation function, and $agg$ and $agg_\forall$ denote the aggregation function for local and global readout respectively. A *formula* is a construction of the formula $\vartheta \geq k$ where $\vartheta$ is an expression and $k$ is an element of $\mathbb{K}$. If $-1 \in \mathbb{K}$, and $-\vartheta$ is not, we can write $-\vartheta$ instead of $(-1) \times \vartheta$. Other standard abbreviations can be used.

Formulas are represented as directed acyclic graphs, aka circuits, meaning that we do not repeat the same expressions several times. For instance, the formula $agg(x_1 + x_2) + (x_1 + x_2) \geq 3$ can be represented as the DAG given in Figure 1. Formulas can also be represented by a sequence of assignments via new fresh intermediate variables. For instance: $y := x_1 + x_2, z := agg(y) + y, res := z \geq 3$.

**Semantics.** Consider a graph $G = (V, E)$, where vertices in $V$ are labeled via a labeling function $\ell : V \rightarrow \mathbb{K}^n$ with feature values. The value of an expression $\vartheta$ in a vertex $u \in V$ is denoted by $[[\vartheta]]_{G,u}$ and is defined by induction on $\vartheta$:

$$[[c]]_{G,u} = c,$$
$$[[x_i]]_{G,u} = \ell(u)_i,$$
$$[[\vartheta + \vartheta']]_{G,u} = [[\vartheta]]_{G,u} +_\mathbb{K} [[\vartheta']]_{G,u},$$

$$[[c \times \vartheta]]_{G,u} = c \times_\mathbb{K} [[\vartheta]]_{G,u},$$
$$[[\alpha(\vartheta)]]_{G,u} = [[\alpha]]([[\vartheta]]_{G,u}),$$
$$[[agg(\vartheta)]]_{G,u} = \Sigma_{v|uEv}[[\vartheta]]_{G,v},$$
$$[[agg_\forall(\vartheta)]]_{G,u} = \Sigma_{v \in V}[[\vartheta]]_{G,v},$$

We define $[[\vartheta \geq k]] = \{G, u \mid [[\vartheta]]_{G,u} \geq_\mathbb{K} [[k]]_{G,u}\}$ (we write $\geq$ for the symbol in the syntax and $\geq_\mathbb{K}$ for the comparison in $\mathbb{K}$). A formula $\varphi$ is satisfiable if $[[\varphi]]$ is non-empty. The *satisfiability problem* for $q\mathcal{L}$ is: Given a $q\mathcal{L}$-formula $\varphi$, decide whether $\varphi$ is satisfiable.

**Simulating a modal logic in the logic $q\mathcal{L}$.** We show that extending $q\mathcal{L}$ with modal operators Blackburn et al. (2001) does not increase the expressivity. We can even compute an equivalent $q\mathcal{L}$ without Boolean connectives and without modal operators in poly-time. It means that formulas like $\varphi_{\mathcal{A}_1} \rightarrow x_0 = 1$ or $\varphi_{\mathcal{A}_1} \wedge \Diamond^{\leq 3}(x_2 = 1)$ have poly-size equivalent formulas in $q\mathcal{L}$.

Assume that $\alpha$ is ReLU. Let $Atm_0$ be the set of atomic formulas of $q\mathcal{L}$ of the form $\vartheta \geq 0$. We suppose that $\vartheta$ takes integer values. In general, $\vartheta \geq k$ is an atomic formula equivalent to $\vartheta - k \geq 0$. Without loss of generality, we thus assume that formulas of $q\mathcal{L}$ are over $Atm_0$. Let modal $q\mathcal{L}$ be the propositional logic on $Atm_0$ extended with modalities and a restricted variant of graded modalities where number $k$ in $\mathbb{K}$ in the following way:

$$[[\Box\varphi]] = \{G, u \mid G, v \in [[\varphi]] \text{ for every } v \text{ s.t. } uEv\}$$
$$[[\Box_g\varphi]] = \{G, u \mid G, v \in [[\varphi]] \text{ for every } v \text{ in } V\}$$
$$[[\Diamond^{\geq k}\varphi]] = \{G, u \mid |\{G, v \mid uEv \text{ and } G, v \in [[\varphi]]\}| \geq_\mathbb{K} k\}$$
$$[[\Diamond_g^{\geq k}\varphi]] = \{G, u \mid |[[\varphi]]| \geq_\mathbb{K} k\}$$

and modalities $\Diamond^{\leq k}\varphi$ and $\Diamond_g^{\leq k}\varphi$ defined the same way but with $\leq_\mathbb{K}$.

**Example 2** (continuation of Example 1). *We first define a few simple formulas to characterize the concepts of the domain. Let $\varphi_A := x_0 = 1$ (Animal), $\varphi_H := x_1 = 1$ (Human), $\varphi_L := x_2 = 1$ (Leg), $\varphi_F := x_3 = 1$ (Fur), $\varphi_W := x_4 = 1$ (White), and $\varphi_B := x_5 = 1$ (Black).*

1. *Has a human owner, whose pets are all two-legged:* $\Diamond(\varphi_H \wedge \Box(\varphi_A \rightarrow \Diamond^{=2}\varphi_L))$.
2. *A human in a community that has more than twice as many animals as humans, and more than five animals without an owner:*

$$\varphi_H \wedge (agg_\forall(x_0) - 2 \times agg_\forall(x_1) \geq 0) \wedge \Diamond_g^{\geq 5}(\varphi_A \wedge \Box(\neg\varphi_H)).$$

3. *An animal in a community where some animals have white and black fur:*

$$\varphi_A \wedge \Diamond_g(\Diamond(\varphi_F \wedge \Diamond\varphi_W) \wedge \Diamond(\varphi_F \wedge \Diamond\varphi_B)).$$

We can see the boolean operator $\neg$, and the various modalities as functions from $\mathrm{Atm}_0$ to $\mathrm{Atm}_0$, and the boolean operator $\vee$ as a function from $\mathrm{Atm}_0 \times \mathrm{Atm}_0$ to $\mathrm{Atm}_0$.

$$f_\neg(\vartheta \geq 0) := -\vartheta - 1 \geq 0$$
$$f_\vee(\vartheta_1 \geq 0, \vartheta_2 \geq 0) := \vartheta_1 + ReLU(\vartheta_2 - \vartheta_1) \geq 0$$
$$f_\Box(\vartheta \geq 0) := agg(-ReLU(-\vartheta)) \geq 0$$
$$f_{\Diamond^{\geq k}}(\vartheta \geq 0) := agg(ReLU(\vartheta + 1) - ReLU(\vartheta)) - k \geq 0$$
$$f_{\Diamond^{\leq k}}(\vartheta \geq 0) := k - agg(ReLU(\vartheta + 1) - ReLU(\vartheta)) \geq 0$$

For the corresponding global modalities ($f_{\Box_g}(\vartheta \geq 0)$, $f_{\Diamond^{\geq k}}(\vartheta \geq 0)$, and $f_{\Diamond^{\leq k}}(\vartheta \geq 0)$), it suffices to use $agg_\forall$ in place of $agg$. The previous transformations can be generalized to arbitrary formulas of modal $q\mathcal{L}$ as follows.

$$mod2expr(\vartheta \geq 0) := \vartheta \geq 0$$
$$mod2expr(\neg\varphi) := f_\neg(mod2expr(\varphi))$$
$$mod2expr(\varphi_1 \vee \varphi_2) := f_\vee(mod2expr(\varphi_1), mod2expr(\varphi_2))$$
$$mod2expr(\boxplus\varphi) := f_\boxplus(mod2expr(\varphi)),$$

where $\boxplus \in \{\Box, \Box_g, \Diamond^{\geq k}, \Diamond_g^{\geq k}, \Diamond^{\leq k}, \Diamond_g^{\leq k}\}$.

We can show that formulas of modal $q\mathcal{L}$ can be captured by a single expression $\vartheta \geq 0$. This is a consequence of the following lemma [3].

**Lemma 3.** *Let $\varphi$ be a formula of modal $q\mathcal{L}$. The formulas $\varphi$ and $mod2expr(\varphi)$ are equivalent.*

## 3.1 CONNECTION BETWEEN $q\mathcal{L}$ AND ACR-GNN VERIFICATION

We focus on the following decision problems in context of the verification of ACR-GNNs:

- (VT1, sufficiency) Given a GNN $\mathcal{A}$ and a $q\mathcal{L}$ formula $\varphi$, decide whether $[[\varphi]] \subseteq [[\mathcal{A}]]$.
- (VT2, necessity) Given a GNN $\mathcal{A}$ and a $q\mathcal{L}$ formula $\varphi$, decide whether $[[\mathcal{A}]] \subseteq [[\varphi]]$.
- (VT3, consistency) Given a GNN $\mathcal{A}$ and a $q\mathcal{L}$ formula $\varphi$, decide whether $[[\varphi]] \cap [[\mathcal{A}]] \neq \emptyset$.

Informally, these problems are described as follows: problem VT1 consists of verifying that pointed graphs satisfying the specification $\varphi$ are classified positively by the GNN $\mathcal{A}$, while VT2 is the reverse, namely verifying that any pointed graphs positively classified by $\mathcal{A}$ satisfies $\varphi$. Problem VT3 consists in verifying whether $\mathcal{A}$ can classify some pointed graph satisfying $\varphi$.

Essential to our results is the straightforward insight that $q\mathcal{L}$ and ACR-GNN are equally expressive.

**Theorem 4.** *Given a ACR-GNN $\mathcal{A}$, we can compute $q\mathcal{L}$-formula in poly-time in the size of $\mathcal{A}$ with $[[\mathcal{A}]] = [[\varphi_\mathcal{A}]]$, and vice-versa.*

The proof follows the same line of reasoning as in expressivity results of previous works such as Sälzer et al. (2025); Nunn et al. (2024); Barceló et al. (2020). However, the key insight needed here is that $q\mathcal{L}$ is tailored to capture the computation of an ACR-GNN, as indicated by the following example[4]:

---

[3]For simplicity, we do not present how to handle $\vartheta \geq 0$ when $\vartheta$ is not an integer. We could introduce several activation functions $\alpha$ in $q\mathcal{L}$, one of them could be interpreted as the Heaviside step function. In the sequel Definition 7, Point 4 is just repeated for each $\alpha$.

[4]To complement this straightforward argument, we provide in Appendix C and D a formal proof for an instantiation of $q\mathcal{L}$ using integers and truncated ReLU, which is then equal to an extension of the logic presented in Nunn et al. (2024) with global aggregation.

**Example 5.** *To reason formally about ACR-GNNs, we represent their computations using $q\mathcal{L}$. Consider an ACR-GNN $\mathcal{A}$ with with two layers of input and output dimension 2, using summation for aggregation, truncated ReLU as activation $\sigma(x) = \max(0, \min(1, x)) = [[\alpha]](x)$, and a classification function $2x_1 - x_2 \geq 1$. The combination functions are:*

$$comb_1((x_1, x_2), (y_1, y_2), (z_1, z_2)) := \begin{pmatrix} \sigma(2x_1 + x_2 + 5y_1 - 3y_2 + 1) \\ \sigma(-x_1 + 4x_2 + 2y_1 + 6y_2 - 2) \end{pmatrix}^T,$$

$$comb_2((x_1, x_2), (y_1, y_2), (z_1, z_2)) := \begin{pmatrix} \sigma(3x_1 - y_1 + 2z_2) \\ \sigma(-2x_1 + 5y_2 + 4z_1) \end{pmatrix}^T.$$

*Note that this assumes that $\mathcal{A}$ operates over $\mathbb{K}$ with at least three bits. Then, the corresponding $q\mathcal{L}$ formula $\varphi_{\mathcal{A}}$ is given by:*

$$\psi_1 := \alpha(2x_1 + x_2 + 5agg(x_1) - 3agg(x_2) + 1)$$
$$\psi_2 := \alpha(-x_1 + 4x_2 + 2agg(x_1) + 6agg(x_2) - 2)$$
$$\chi_1 := \alpha(3\psi_1 - agg(\psi_1) + 2(agg_\forall(\psi_2)))$$
$$\chi_2 := \alpha(-2\psi_1 + 5(agg(\psi_2)) + 4agg_\forall(\psi_1))$$
$$\varphi_{\mathcal{A}} := 2(\chi_1) - \chi_2 \geq 1.$$

*To sum up, given a GNN $\mathcal{A}$, we compute $q\mathcal{L}$-formula in poly-time in the size of $\mathcal{A}$ with $[[\mathcal{A}]] = [[\varphi_{\mathcal{A}}]]$.*

Finally, ACR-GNN verification tasks can be solved by reduction to the satisfiability problem of $q\mathcal{L}$:

- VT1 by checking that $\varphi \wedge \neg\varphi_{\mathcal{A}}$ is not satisfiable
- VT2 by checking that $\neg\varphi \wedge \varphi_{\mathcal{A}}$ is not satisfiable
- VT3 by checking that $\varphi \wedge \varphi_{\mathcal{A}}$ is satisfiable

## 4   COMPLEXITY UPPER BOUND OF THE VERIFICATION TASKS

In this section, we prove the NEXPTIME membership of reasoning in modal quantized logic, and also of solving of ACR-GNN verification tasks (by reduction to the former). Remember that the activation function $\alpha$ can be arbitrary in our setting. Our result holds with the loose restriction that $[[\alpha]]$ is computable in exponential-time in the bit-width $n$ of $\mathbb{K}$.

**Theorem 6.** *The satisfiability problem of $q\mathcal{L}$ is decidable and in NEXPTIME, and so VT3. VT1 and VT2 are in coNEXPTIME.*

In order to prove Theorem 6, we adapt the NEXPTIME membership of the description logic $\mathcal{ALCSCC}^{++}$ from Baader et al. (2020) to logic $q\mathcal{L}$. The difference resides in the definition of Hintikka sets and the treatment of quantization. The idea is to encode the constraints of a $q\mathcal{L}$-formula $\varphi$ in a formula of exponential length of a quantized version of QFBAPA, that we prove to be in NP.

### 4.1   HINTIKKA SETS

Consider $q\mathcal{L}$-formula $\varphi$. Let $E(\varphi)$ be the set of subexpressions in $\varphi$. For instance, if $\varphi$ is $agg(\alpha(x_2 + agg_\forall(x_1))) \geq 5$ then $E(\varphi) = \{agg(\alpha(x_2 + agg_\forall(x_1))), \alpha(x_2 + agg_\forall(x_1)), x_2, agg_\forall(x_1), x_1\}$. From now on, we consider equality subformulas that are of the form $\vartheta{=}k$ where $\vartheta$ is a subexpression of $\varphi$ and $k \in \mathbb{K}$.

**Definition 7.** *A Hintikka set $H$ for $\varphi$ is a subset of subformulas of $\varphi$ such that:*

1. *For all $\vartheta \in E(\varphi)$, there is a unique value $k \in \mathbb{K}$ such that $\vartheta = k \in H$*
2. *For all $\vartheta_1{+}\vartheta_2 \in E(\varphi)$, if $\vartheta_1{=}k_1, \vartheta_2{=}k_2 \in H$ then $\vartheta_1{+}\vartheta_2{=}k_1{+}k_2 \in H$*
3. *For all $c \times \vartheta \in E(\varphi)$, if $\vartheta = k \in H$ then $c \times \vartheta{=}k' \in H$ where $k' = c \times_{\mathbb{K}} k$*
4. *For all $\alpha(\vartheta) \in E(\varphi)$, $\vartheta{=}k \in H$ and $\alpha(\vartheta){=}k' \in H$ implies $k' = [[\alpha]](k)$*

Informally, a *Hintikka set* contains equality subformulas obtained from a choice of a value for each subexpression of $\varphi$ (point 1), provided that the set is consistent *at the current vertex* (points 2–4). The notion of Hintikka set does not take any constraints about $agg$ and $agg_\forall$ into account since checking consistency of aggregation would require information about the neighbor or the whole graph.

**Example 8.** *If $\varphi$ is $agg(\alpha(x_2 + agg_\forall(x_1))) \geq 5$, then an example of Hintikka set is: $\{agg(\alpha(x_2 + agg_\forall(x_1)) = 8, \alpha(x_2 + agg_\forall(x_1)) = 9, x_2 + agg_\forall(x_1) = 9, x_2 = 7, agg_\forall(x_1) = 2, x_1 = 5\}$.*

**Proposition 9.** *The number of Hintikka sets is bounded by $2^{n|\varphi|}$ where $|\varphi|$ is the size of $\varphi$, and $n$ is the bitwidth of $\mathbb{K}$.*

### 4.2 Quantized Version of QFBAPA (Quantifier-free Boolean Algebra and Presburger Arithmetics)

A QFBAPA formula is a propositional formula where each atom is either an inclusion of sets or equality of sets or linear constraints (Kuncak & Rinard (2007), and Appendix E.2.1). Sets are denoted by Boolean algebra expressions, e.g., $(S \cup S') \setminus S''$, or $\mathcal{U}$ where $\mathcal{U}$ denotes the set of all points in some domain. Here $S$, $S'$, etc. are set variables. Linear constraints are over $|S|$ denoting the cardinality of the set denoted by the set expression $S$. For instance, the QFBAPA-formula $(pianist \subseteq happy) \wedge (|happy| + |\mathcal{U} \setminus pianist| \geq 6) \wedge (|happy| < 2)$ is read as 'all pianists are happy and the number of happy persons plus the number of persons that are not pianists is greater than 6 and the number of happy persons is smaller than 2'.

We now introduce a *quantized* version QFBAPA$_\mathbb{K}$ of QFBAPA. It has the same syntax as QFBAPA except that numbers in expressions are in $\mathbb{K}$. Semantically, every numerical expression is interpreted in $\mathbb{K}$. For each set expression $S$, the interpretation of $|S|$ is not the cardinality $c$ of the interpretation of $S$, but the result of the computation $1 + 1 + \ldots + 1$ in $\mathbb{K}$ with $c$ occurrences of 1 in the sum.

We consider that $\mathbb{K}$ that saturates, meaning that if $x + y$ exceed the upper bound limit of $\mathbb{K}$, there is a special value denoted by $+\infty$ such that $x + y = +\infty$.

**Proposition 10.** *If bitwidth $n$ is in unary, and if $\mathbb{K}$ saturates, then satisfiability in QFBAPA$_\mathbb{K}$ is in NP.*

### 4.3 Reduction to QFBAPA$_\mathbb{K}$

Let $\varphi$ be a formula of $q\mathcal{L}$. For each Hintikka set $H$, we introduce the set variable $X_H$ that intuitively represents the $H$-vertices, i.e., the vertices in which subformulas of $H$ hold. The following QFBAPA$_\mathbb{K}$-formulas say that the interpretation of $X_H$ form a partition of the universe. For each subformula $\vartheta' = k$, we introduce the set variable $X_{\vartheta'=k}$ that intuitively represents the vertices in which $\vartheta' = k$ holds. Equation (1) expresses that $\{X_H\}_H$ form a partition of the universe. Equation (2) makes the bridge between variables $X_{\vartheta'=k}$ and $X_H$.

$$\left( \bigwedge_{H \neq H'} X_H \cap X_{H'} = \emptyset \right) \wedge \left( \bigcup_H X_H = \mathcal{U} \right) \quad (1) \qquad \bigwedge_{\vartheta' \in E(\varphi)} \bigwedge_{k \in \mathbb{K}} \left( X_{\vartheta'=k} = \bigcup_{H | \vartheta'=k \in H} X_H \right) \quad (2)$$

We introduce also a variable $S_H$ that denotes the set of all successors of some $H$-vertex. If there is no $H$-vertex then the variable $S_H$ is just irrelevant.

The following QFBAPA$_\mathbb{K}$-formula (Equation (3)) encodes the semantics of $agg(\vartheta)$. More precisely, it says that for all subexpressions $agg(\vartheta)$, for all values $k$, for all Hintikka sets $H$ containing formula $agg(\vartheta) = k$, if there is some $H$-vertex (i.e., some vertex in $X_H$), then the aggregation obtained by summing over the successors of some $H$-vertex is $k$.

$$\bigwedge_{agg(\vartheta) \in E(\varphi)} \bigwedge_{k \in \mathbb{K}} \bigwedge_{\substack{\text{Hintikka set } H \\ | \; agg(\vartheta)=k \; \in \; H}} \left[ (X_H \neq \emptyset) \rightarrow \sum_{k' \in \mathbb{K}} |S_H \cap X_{\vartheta=k'}| \times k' = k \right] \quad (3)$$

In the previous sum, we partition $S_H$ into subsets $S_H \cap X_{\vartheta=k'}$ for all possible values $k'$. Each contribution for a successor in $S_H \cap X_{\vartheta=k'}$ is $k'$. We rely here on the fact[5] that $(1+1+\ldots+1) \times k' = k' + k' + \ldots + k'$. We also fix a specific order over values $k'$ in the summation (it means that $agg(\vartheta)$ is computed as follows: first order the successors according to the taken values of $\vartheta$ in that specific order, then perform the summation). Finally, the semantics of $agg_\forall$ is captured by the formula:

$$\bigwedge_{agg_\forall(\vartheta) \in E(\varphi)} \bigwedge_{k \in \mathbb{K}} X_{agg_\forall(\vartheta)=k} \neq \emptyset \rightarrow \sum_{k' \in \mathbb{K}} |X_{\vartheta=k'}| \times k' = k \quad (4)$$

---

[5]This is true for some fixed-point arithmetics but not for floating-point arthmetics. See Appendix B.2.

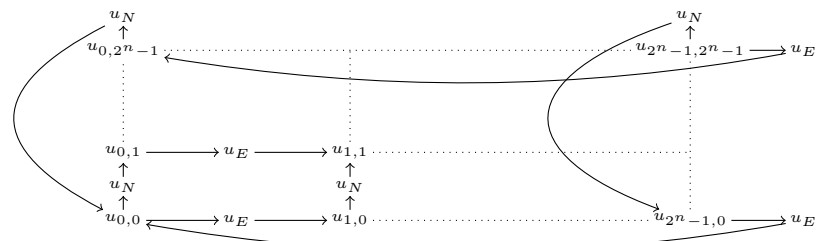

Figure 2: Encoding a torus of exponential size with (modal) $q\mathcal{L}$ formulas. Vertices $u_{x,y}$ correspond to locations $(x, y)$ in the torus while $u_N$ and $u_E$ denote intermediate vertices indicating the direction (resp., north and east).

Note that intuitively Equation (4) implies that for $X_{agg_\forall(\vartheta)=k}$ is interpreted as the universe, for the value $k$ which equals the semantics of $\sum_{k' \in \mathbb{K}} |X_{\vartheta=k'}| \times k'$.

Given $\varphi = \vartheta \geq k$, we define $tr(\varphi) := \psi \wedge \bigvee_{k' \geq k} X_{\vartheta=k'} \neq \emptyset$ where $\psi$ the conjunction of Formulas 1–4. The function $tr$ requires to compute all the Hintikka sets. So we need in particular to check Point 4 of Definition 7 and we get the following when $[[\alpha]]$ is computable in exponential time in $n$.

**Proposition 11.** $tr(\varphi)$ *is computable in exponential-time in* $|\varphi|$ *and* $n$.

**Proposition 12.** *Let* $\varphi$ *be a formula of* $q\mathcal{L}$. *$\varphi$ is satisfiable iff* $tr(\varphi)$ *is* QFBAPA$_\mathbb{K}$ *satisfiable.*

Finally, in order to check whether a $q\mathcal{L}$-formula $\varphi$ is satisfiable, we construct a QFBAPA$_\mathbb{K}$-formula $tr(\varphi)$ in exponential time. As the satisfiability problem of QFBAPA$_\mathbb{K}$ is in NP, we obtain that the satisfiability problem of $q\mathcal{L}$ is in NEXPTIME. We proved Theorem 6,

**Remark 13.** *Our methodology can be generalized to reason in subclasses of graphs. For instance, we may tackle the problem of satisfiability in a graph where vertices are of bounded degree bounded by $d$. To do so, we add the constraint $\bigwedge_H |S_H| \leq d$.*

## 5 COMPLEXITY LOWER BOUND OF THE VERIFICATION TASKS

The NEXPTIME upper-bound is tight. Having defined modalities in $q\mathcal{L}$ and stated Lemma 3, Theorem 14 is proven by adapting the proof of NEXPTIME-hardness of deciding the consistency of $\mathcal{ALCQ}$-$T_C$Boxes presented in Tobies (2000). So we already have the hardness result for ReLU.

NEXPTIME-hardness is proven via a reduction from the tiling problem by Wang tiles of a torus of size $2^n \times 2^n$. A Wang tile is a square with colors, e.g., ◩, ◪, etc. The problem takes as input a number $n$ in unary, and Wang tile types, and an initial condition—let us say the bottom row is already given. The objective is to decide whether the torus of $2^n \times 2^n$ can be tiled while colors of adjacent Wang tiles match. A slight difficulty resides in adequately capturing a two-dimensional grid structure—as in Figure 2—with only a single relation. To do that, we introduce special formulas $\varphi_E$ and $\varphi_N$ to indicate the direction (east or north). In the formula computed by the reduction, we also need to bound the number of vertices corresponding to tile locations by $2^n \times 2^n$. Thus $\mathbb{K}$ needs to encode $2^n \times 2^n$. We need a bit-width of at least $2n$.

**Theorem 14.** *The satisfiability problem in $q\mathcal{L}$ is NEXPTIME-hard, and so is the ACR-GNN verification task* VT3. *The ACR-GNN verification tasks* VT1 *and* VT2 *are coNEXPTIME-hard.*

**Remark 15.** *It turns out that the verification task only needs the fragment of $q\mathcal{L}$ where agg is applied directly on an expression $\alpha(..)$. Indeed, this is the case when we represent a GNN in $q\mathcal{L}$ or when we translate logical formulas in $q\mathcal{L}$ (Lemma 3). Reasoning about $q\mathcal{L}$ when $\mathbb{K} = \mathbb{Z}$ and the activation function is truncated ReLU is also NEXPTIME-complete (see Appendix F).*

## 6 BOUNDING THE NUMBER OF VERTICES

The satisfiability problem is NEXPTIME-complete, thus far from tractable. The high complexity arises because counterexamples can be arbitrary large graphs. However, one usually looks for small

counterexamples. Let $\mathcal{G}^{\leq N}$ be the set of pointed graphs with at most $N$ vertices. We consider the $q\mathcal{L}$ and ACR-GNN *satisfiability problem with a bound on the number of vertices* and ACR-GNNs verification tasks: given a number $N$ given in unary, 1. given a $q\mathcal{L}$-formula $\varphi$, is it the case that $[[\varphi]] \cap \mathcal{G}^{\leq N} \neq \emptyset$, 2. given an ACR-GNN $\mathcal{A}$, is it the case that $[[\mathcal{A}]] \cap \mathcal{G}^{\leq N} \neq \emptyset$. In the same way, we introduce the following verification tasks. Given a GNN $\mathcal{A}$, a $q\mathcal{L}$ formula $\varphi$, and a number $N$ in unary: (VT'1, sufficiency) Do we have $[[\varphi]] \cap \mathcal{G}^{\leq N} \subseteq [[\mathcal{A}]] \cap \mathcal{G}^{\leq N}$? (VT'2, necessity) Do we have $[[\mathcal{A}]] \cap \mathcal{G}^{\leq N} \subseteq [[\varphi]] \cap \mathcal{G}^{\leq N}$? (VT'3, consistency) Do we have $[[\varphi]] \cap [[\mathcal{A}]] \cap \mathcal{G}^{\leq N} \neq \emptyset$?

**Theorem 16.** *The satisfiability problems with bounded number of vertices are NP-complete, so is ACR-GNN verification task* VT'3*, while the verification tasks* VT'1 *and* VT'2 *are coNP-complete.*

It is then possible to extend the methodology of Sena et al. (2021) but for verifying GNNs. An efficient SMT encoding of GNN verification tasks would be a contribution of its own. We merely propose a proof of concept and a baseline for future research. See Appendix B.1 for details.Our implementation proposal is a Python program that takes a quantized GNN $\mathcal{A}$ as an input, a precondition, a postcondition and a bound $N$. It then produces a C program that mimics the execution of $\mathcal{A}$ on an arbitrary graph with at most $N$ vertices, and embeds the pre/postcondition. We then apply ESBMC (SMT-based context-bounded model checker) (Menezes et al., 2024) on the C program.

## 7 QUANTIZATION EFFECTS ON ACCURACY, PERFORMANCE AND MODEL SIZE

We now investigate the application of dynamic Post-Training Quantization (PTQ) to ACR-GNNs. As a reference, we used the models described and analyzed in Barceló et al. (2020), using their implementation (Barceló et al., 2021) as the baseline. Experiments used two datasets: one synthetic (Erdös–Rényi model) and one real-world dataset (Protein-Protein Interactions (PPI) benchmark by Zitnik & Leskovec (2017)). In the original work, experiments were made with two activation functions: Rectified Linear Unit (ReLU) and truncated ReLU. Since the models of Section 2 can handle arbitrary activation functions, in our experiments, we used several types of activation function: Piecewise linear (ReLU, ReLU6, and trReLU), Smooth unbounded (GELU and SiLU), Smooth bounded (Sigmoid), and Smooth ReLU-like (Softplus and ELU). The quantization method is implemented in PyTorch (Ansel et al., 2024; PyTorch Team, 2024a), dynamic PTQ transforms a pre-trained floating-point model into a quantized version without requiring retraining. In this approach, model weights are statically quantized to INT8, while activations remain in floating-point format until they are dynamically quantized at compute time. This hybrid representation enables efficient low-precision computation using INT8-based matrix operations, thereby reducing the memory footprint and improving the inference speed. The PyTorch implementation applies per-tensor quantization to the weights and stores the activations as floating-point values between operations to strike a balance between accuracy and performance.

Synthetic graphs (Table 2 of Appendix G) were generated using the dense Erdös–Rényi model, a classical approach to constructing random graphs. Each graph includes five initial node colors, encoded as one-hot feature vectors. Following Barceló et al. (2020), labels were assigned using formulas from the logic fragment FOC$_2$. Specifically, a hierarchy of classifiers $\alpha_i(x)$ was defined as follows: $\alpha_0(x) := Blue(x)$ and $\alpha_{i+1}(x) := \exists^{[N,M]} y\,(\alpha_i(y) \wedge \neg E(x,y))$, where $\exists^{[N,M]}$ denotes the quantifier "there exist between $N$ and $M$ nodes" satisfying a given condition. Each classifier $\alpha_i(x)$ can be expressed within FOC$_2$, as the bounded quantifier can be rewritten using $\exists^{\geq N}$ and $\neg\exists^{\geq M+1}$. Each property $p_i$ corresponds to a classifier $\alpha_i$ with $i \in \{1,2,3\}$. For the analysis, we collected training time, model size, and accuracy for both datasets. We list the principal findings of the analysis. More detailed statistics can be found in the Appendix G. According to our experimental flow, we first examine the training time. For both datasets, we found that piecewise-linear activation functions consistently achieve the shortest training time (Table 4 and Table 29 of Appendix G). Moreover, for computational efficiency, we found that the model with the Softplus activation function was consistently the slowest, regardless of the datasets (Table 5 and Table 30 of Appendix G). We computed the Reduction (in percentage) of the model size: $\approx 60\%$ for Erdös–Rényi (Table 8) and $\approx 74\%$ for PPI (Table 33). We calculated the mean speed-up of models after dynamic PTQ, defined as the ratio of the non-quantized execution time to the quantized one. We observed that dynamic PTQ does not accelerate inference for both datasets (Figures 6 (for Erdös–Rényi) and 9 (for PPI) in Appendix G). We performed a detailed analysis of the accuracy of both data (Tables 9–18 and Tables 34–43 in Appendix G). Across all activation families, the observed $\Delta_{acc}$ values are generally within $\pm 1\%$. In addition, we observed a drop in the accuracy of the baseline models after two layers.

These findings highlight the advantages of quantized ACR-GNN models of Section 2 with respect to non-quantized models, striking a remarkable balance between model size and accuracy.

To complete our quantization study, we evaluated ACR-GNNs under a range of bit-width configurations (32, 16, 8, 7, 6, 5, 4, 2), and additionally under finer-grained reductions between 8 and 4 bits. Across both datasets, 16- and 8-bit models preserved accuracy within a small margin of the full-precision baseline. For the synthetic dataset (Tables 19–26 in Appendix G), notable accuracy drops emerged only below 8 bits, with consistent degradation from 6 to 5 and from 5 to 4 bit transitions across classifiers and activation functions. In contrast, the PPI dataset (Tables 44–51 in Appendix G) exhibits strong robustness to dynamic post-training quantization: performance is preserved down to 6 bits across all activation functions, with only minor deviations (less than 3%) observed in the from 7 to 6 and from 6 to 5 bit transitions. The only marked degradation appears when moving into the very low-precision range (5 bits and below).

## 8 CONCLUSION AND FUTURE WORK

The main result is the (co)NEXPTIME-completeness of verification tasks for GNNs with global readout. It helps to understand the inherent complexity, and demonstrates that the verification of ACR-GNNs is highly intractable. To further guide future research, we provide extensive experiments demonstrating that quantised GNNs are essential. However, naive approaches to verifying GNNs, even over a set of graphs with a bounded number of vertices, quickly reach their limits, as expected, given the inherently high complexity. This prompts significant efforts of the research community towards ensuring the safety of quantised GNN-based systems.

There are many directions to go. First, characterizing the modal flavor of $q\mathcal{L}$—a powerful graph property specification language—for other activation functions than ReLU. New extensions of $q\mathcal{L}$ could also be proposed to tackle other classes of GNNs. Verification of neural networks is challenging and is currently tackled by the verification community (Cordeiro et al., 2025). So it will be for GNNs as well. Our verification tool with a bound on the number of vertices is still preliminary and a mere baseline for future research. One obvious path would be to improve it, to compare different approaches (bounded model checking vs. linear programming as in Huang et al. (2024)) and to apply it to real GNN verification scenarios. Designing a practical verification procedure in the general case (without any bound on the number of vertices) and overcoming the high computational complexity is an exciting challenge for future research towards the verification of GNNs.

**Limitations.** Section 4 and 5 reflect theoretical results. Some practical implementations of GNNs may not fully align with them. In particular, the order in the (non-associative) summation over values in $\mathbb{K}$ is fixed in formulas (3) and (4). It means that we suppose that the aggregation $agg(\vartheta)$ is computed in that order too (we sort the successors of a vertex according to the values of $\vartheta$ and then perform the summation).

## ETHICS STATEMENT

This research poses no significant ethical concerns. But it warns the research community that the safety of GNN-based systems is hard to ensure, and prompts further research.

## REPRODUCIBILITY STATEMENT

Full proofs are in the appendix.

The code of the verification prototype is in the supplementary material, and located in the folder 'src_verificationtool'.

The reproducibility package for the experimental evaluation of ACR-GNN quantization is provided in the folder 'code_notebooks_csv'. The 'Code' subfolder contains the Python implementation, along with a 'README.md' file that provides a detailed description of the files and step-by-step instructions for reproducing the experiments. The 'Notebooks' subfolder includes the analysis scripts, all corresponding '.csv' files, and an additional file with usage instructions and descriptions.

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

## A    PROOFS OF STATEMENTS IN THE MAIN TEXT

**Lemma 3.** *Let $\varphi$ be a formula of modal $q\mathcal{L}$. The formulas $\varphi$ and $mod2expr(\varphi)$ are equivalent.*

*Proof.* We have to prove that for all $G, u$, we have $G, u \models \varphi$ iff $G, u \models mod2expr(\varphi)$. We proceed by induction on $\varphi$.

- The base case is obvious: $G, u \models \varphi$ iff $G, u \models mod2expr(\varphi)$ is $G, u \models \varphi$ iff $G, u \models mod2expr(\varphi)$.

- $G, u \models \neg\varphi$ iff $G, u \not\models \varphi$

  iff (by induction) $G, u \not\models mod2expr(\varphi)$

  iff (by writing $mod2expr(\varphi) = \vartheta \geq 0$) $G, u \not\models \vartheta \geq 0$

  iff $G, u \models \vartheta < 0$

  iff $G, u \models \vartheta \leq -1$ (because we suppose that $\vartheta$ takes its value in the integers

  iff $G, u \models \vartheta + 1 \leq 0$

  iff $G, u \models -\vartheta - 1 \geq 0$.

- $G, u \models (\varphi_1 \vee \varphi_2)$

  iff $G, u \models \varphi_1$ or $G, u \models \varphi_2$

  iff $G, u \models (\vartheta_1 \geq 0)$ or $G, u \models (\vartheta_2 \geq 0)$

  iff $G, u \models \vartheta_1 + ReLU(\vartheta_2 - \vartheta_1) \geq 0$

  Indeed, ($\Rightarrow$) if $G, u \models (\vartheta_1 \geq 0)$ then $G, u \models \vartheta_1 + ReLU(\vartheta_2 - \vartheta_1) \geq \vartheta_1 \geq 0$.

  If $G, u \models (\vartheta_2 \geq 0)$ and $G, u \models (\vartheta_1 < 0)$ then $G, u \models \vartheta_1 + ReLU(\vartheta_2 - \vartheta_1) = \vartheta_1 + \vartheta_2 - \vartheta_1 = \vartheta_2 \geq 0$.

  ($\Leftarrow$) Conversely, by contrapositive, if $G, u \models (\vartheta_2 < 0)$ and $G, u \models (\vartheta_1 < 0)$, then $G, u \models \vartheta_1 + ReLU(\vartheta_2 - \vartheta_1) = \vartheta_1 + \vartheta_2 - \vartheta_1 = \vartheta_2 < 0$ or $G, u \models \vartheta_1 + ReLU(\vartheta_2 - \vartheta_1) = \vartheta_1 + 0 = \vartheta_1 < 0$. In the two cases, $G, u \models \vartheta_1 + ReLU(\vartheta_2 - \vartheta_1) < 0$.

- $G, u \models \Diamond^{\geq k}\varphi$ iff the number of vertices $v$ that are successors of $u$ and with $G, v \models \varphi$ is greater than $k$

  iff the number of vertices $v$ that are successors of $u$ and with $G, v \models mod2expr(\varphi)$ is greater than $k$

  iff (written $\vartheta \geq 0$) iff the number of vertices $v$ that are successors of $u$ and with $G, v \models \vartheta \geq 0$ is greater than $k$

  iff the number of vertices $v$ that are successors of $u$ and with $G, v \models ReLU(\vartheta + 1) - ReLU(\vartheta) = 1$ is greater than $k$ (since we know by defining of modal $q\mathcal{L}$ that $\vartheta$ takes its value in integers)

  iff $G, u \models agg(ReLU(\vartheta + 1) - ReLU(\vartheta) \geq k$

  iff $G, u \models mod2expr(\Diamond^{\geq k}\varphi)$

- Other cases are similar.

$\square$

**Proposition 9.** *The number of Hintikka sets is bounded by $2^{n|\varphi|}$ where $|\varphi|$ is the size of $\varphi$, and $n$ is the bitwidth of $\mathbb{K}$.*

*Proof.* For each expression $\vartheta$, we choose a number in $\mathbb{K}$. There is $2^n$ different numbers. There are $|\varphi|$ number of expressions. So we get $(2^n)^{|\varphi|} = 2^{n|\varphi|}$ possible choices for a Hintikka set. $\square$

**Proposition 10.** *If bitwidth $n$ is in unary, and if $\mathbb{K}$ saturates, then satisfiability in QFBAPA$_{\mathbb{K}}$ is in NP.*

*Proof.* Here is a non-deterministic algorithm for the satisfiability problem in QFBAPA$_{\mathbb{K}}$.

1. Let $\chi$ be a QFBAPA$_{\mathbb{K}}$ formula.

2. For each set expression $B$ appearing in some $|B|$, guess a non-negative integer number $k_B$ in $\mathbb{K}$.

3. Let $\chi'$ be a (grounded) formula in which we replaced $|B|$ by $k_B$.

4. Check that $\chi'$ is true (can be done in poly-time since $\chi'$ is a grounded formula, it is a Boolean formula on variable-free equations and inequations in $\mathbb{K}$).

5. If not we reject.

6. We now build a standard QFBAPA formula $\delta = \bigwedge_B constraint(B)$ where:

$$constraint(B) = \begin{cases} |B| = k_B \text{ if } k_B < \infty_{\mathbb{K}} \\ |B| \geq limit \text{ if } k_B = +\infty_{\mathbb{K}} \end{cases}$$

where $limit$ is the maximum number that is considered as infinity in $\mathbb{K}$.

7. Run a non-deterministic poly-time algorithm for the QFBAPA satisfiability on $\delta$. Accepts if it accepts. Otherwise reject.

The algorithm runs in poly-time. Guessing a number $n_B$ is in poly-time since it consists in guessing $n$ bits ($n$ in unary). Step 4 is just doing the computations in $\mathbb{K}$. In Step 6, $\delta$ can be computed in poly-time.

If $\chi$ is QFBAPA$_{\mathbb{K}}$ satisfiable, then there is a solution $\sigma$ such that $\sigma \models \chi$. At step 2, we guess $n_B = |\sigma(B)|_{\mathbb{K}}$. The algorithm accepts the input.

Conversely, if the algorithm accepts its input, $\chi'$ is true for the chosen values $n_B$. $\delta$ is satisfiable. So there is a solution $\sigma$ such that $\sigma \models \delta$. By the definition of $constraint$, $\sigma \models \chi$. $\qquad\square$

**Remark 17.** *If the number $n$ of bits to represent $\mathbb{K}$ is given in unary and if $\mathbb{K}$ is "modulo", then the satisfiability problem in QFBAPA$_{\mathbb{K}}$ is also in NP. The proof is similar except than now $constraint(B) = (|B| = k_B + Ld_B)$ where $d_B$ is a new variable.*

**Proposition 11.** *$tr(\varphi)$ is computable in exponential-time in $|\varphi|$ and $n$.*

*Proof.* In order to create $tr(\varphi)$, we write an algorithm where each big conjunction, big disjunction, big union and big sum is replaced by a loop. For instance, $\bigwedge_{H \neq H'}$ is replaced by two inner loops over Hintikka sets. Note that we create check whether a candidate $H$ is a Hintikka set in exponential time in $n$ since Point 4 can be checked in exponential time in $n$ (thanks to our loose assumption on the computability of $[[\alpha]]$ in exponential time in $n$. There are $2^{n|\varphi|}$ many of them. In the same way, $\bigwedge_{k \in \mathbb{K}}$ is a loop over $2^n$ values. There is a constant number of nested loops, each of them iterating over an exponential number (in $n$ and $|\varphi|$ of elements. QED. $\qquad\square$

**Proposition 12.** *Let $\varphi$ be a formula of q$\mathcal{L}$. $\varphi$ is satisfiable iff $tr(\varphi)$ is QFBAPA$_{\mathbb{K}}$ satisfiable.*

*Proof.* $\boxed{\Rightarrow}$ Let $G, u$ such that $G, u \models \varphi$. We set $\sigma(X_{\vartheta'=k}) := \{v \mid [[\vartheta']]_{G,v} = k\}$ and $\sigma(X_H) = \{v \mid G, v \models H\}$ where $G, u \models H$ means that for all $\vartheta' = k \in H$, we have $[[\vartheta']]_{G,v} = k$. For all Hintikka sets $H$ such that there is $v$ such that $G, v \models H$, we set: $\sigma(S_H) := \{w \mid vEw\}$.

We check that $\sigma \models tr(\varphi)$. First, $\sigma$ satisfies Formulas 1 and 2 by definition of $\sigma$. Now, $\sigma$ also satisfies Formula 3. Indeed, if $agg(\vartheta') = k \in H$, then if there is no $H$-vertex in $G$ then the implication is true. Otherwise, consider the $H$-vertex $v$. But, then by definition of $X_{agg(\vartheta')=k}$, $[[agg(\vartheta')]]_{G,v} = k$. But then the semantics of $agg$ exactly corresponds to $\sum_{k' \in \mathbb{K}} |S_H \cap X_{\vartheta=k'}| \times k' = k$. Indeed, each $S_H \cap X_{\vartheta=k'}$-successor contributes with $k'$. Thus, the contribution of successors where $\vartheta$ is $k'$ is $|S_H \cap X_{\vartheta=k'}| \times k'$.

Formula 4 is also satisfied by $\sigma$. Actually, let $k$ such that $\sigma \models X_{agg_\forall(\vartheta)=k} = \mathcal{U}$. This means that the value of $agg_\forall(\vartheta)$ (which does not depend on a specific vertex $u$ but only on $G$) is $k$. The sum $\sum_{k' \in \mathbb{K}} |X_{\vartheta=k'}| \times k' = k$ is the semantics of $agg_\forall(\vartheta) = k$.

Finally, as $G, u \models \varphi$, and $\varphi$ is of the form $\vartheta \geq k$, there is $k' \geq k$ such that $[[\vartheta]]_{G,u} = k'$. So $X_{\vartheta=k'} \neq \emptyset$.

$\boxed{\Leftarrow}$ Conversely, consider a solution $\sigma$ of $tr(\varphi)$. We construct a graph $G = (V, E)$ as follows.

$$V := \sigma(\mathcal{U})$$
$$E := \{(u, v) \mid \text{for some } H, u \in \sigma(X_H) \text{ and } v \in \sigma(S_H)\}$$
$$\ell(v)_i := k \text{ where } v \in X_{x_i=k}$$

i.e., the set of vertices is the universe, and we add an edge between any $H$-vertex $u$ and a vertex $v \in \sigma(S_H)$, and the labeling for features is directly given $X_{x_i=k}$. Note that the labeling is well-defined because of formulas 1 and 2.

As $\sigma \models |X_\varphi| \geq 1$, there exists $u \in \sigma(X_\varphi)$. Let us prove that $G, u \models \varphi$. By induction on $\vartheta'$, we prove that $u \in X_{\vartheta'=k}$ implies $[[\vartheta']]_{G,u} = k$. The base case is obtained via the definition of $\ell$. Cases for $+$, $\times$ and $\alpha$ are obtained because each vertices is in some $\sigma(X_H)$ for some $H$. As the definition of Hintikka set takes care of the semantics of $+$, $\times$ and $\alpha$, we have $[[\vartheta_1 + \vartheta_2]]_{G,u} = [[\vartheta_1]]_{G,u} + [[\vartheta_2]]_{G,u}$, etc.

$[[agg(\vartheta)]]_{G,u} = \Sigma_{v|uEv}[[\vartheta]]_{G,v}$ and $[[agg_\forall(\vartheta)]]_{G,u} = \Sigma_{v \in V}[[\vartheta]]_{G,v}$ hold because of $\sigma$ satisfies respectively formula 3 and 4. $\qquad \square$

**Theorem 14.** *The satisfiability problem in $q\mathcal{L}$ is NEXPTIME-hard, and so is the ACR-GNN verification task* VT3. *The ACR-GNN verification tasks* VT1 *and* VT2 *are coNEXPTIME-hard.*

*Proof.* We reduce the NEXPTIME-hard problem of deciding whether a domino system $\mathcal{D} = (D, V, H)$, given an initial condition $w_0 \ldots w_{n-1} \in D^n$, can tile an exponential torus Tobies (2000). In the domino system, $D$ is the set of tile types, and $V$ and $H$ respectively are the respectively vertical and horizontal color compatibility relations. We are going to write a set of modal $q\mathcal{L}$ formulas that characterize the torus $\mathbb{Z}^{2n+1} \times \mathbb{Z}^{2n+1}$ and the domino system. We use $2n + 2$ features. We use $x_0, \ldots x_{n-1}$, and $y_0, \ldots, y_{n-1}$, to hold the (binary-encoded) coordinates of vertices $u_{x,y}$ in the torus. We use the feature $x_N$ to denote a vertex $u_N$ 'on the way north' (when $x_N = 1$) and $x_E$ to denote a vertex $u_E$ 'on the way east' (when $x_E = 1$), with abbreviations $\varphi_N := x_N = 1$, and $\varphi_E := x_E = 1$. See Figure 2.

For every $n \in \mathbb{N}$, we define the following set of formulas. $T_n =$

$$\begin{array}{lll}
\{ & \Box_g(x_N = 1 \vee x_N = 0) & , & \Box_g(x_E = 1 \vee x_E = 0), \\
& \Box_g(\bigwedge_{k=0}^{n-1}(x_i = 1 \vee x_i = 0)) & , & \Box_g(\bigwedge_{k=0}^{n-1}(y_i = 1 \vee y_i = 0)), \\
& \Box_g(\neg(x_N = 1 \wedge x_E = 1)) & , & \Box_g(\neg(\varphi_N \vee \varphi_E) \rightarrow agg(1) = 2), \\
& \Box_g(\neg(\varphi_N \vee \varphi_E) \rightarrow (agg(x_N) = 1)) & , & \Box_g(\neg(\varphi_N \vee \varphi_E) \rightarrow (agg(x_E) = 1)), \\
& \Box_g(\varphi_N \rightarrow agg(1) = 1) & , & \Box_g(\varphi_E = 1 \rightarrow agg(1) = 1), \\
& \Diamond_g^{=1}\varphi_{(0,0)} & , & \Diamond_g^{=1}\varphi_{(2^n-1,2^n-1)}, \\
& \Box_g(\neg(\varphi_N \vee \varphi_E) \rightarrow \varphi_{east}) & , & \Box_g(\neg(\varphi_N \vee \varphi_E) \rightarrow \varphi_{north}), \\
& \Diamond_g^{\leq 2^n \times 2^n}\neg(\varphi_N \vee \varphi_E), & \Diamond_g^{\leq 2^n \times 2^n}\varphi_N, & \Diamond_g^{\leq 2^n \times 2^n}\varphi_E \quad \}
\end{array}$$

where $\varphi_{(0,0)} := \bigwedge_{k=0}^{n-1} x_i = 0 \wedge \bigwedge_{k=0}^{n-1} y_i = 0$, and $\varphi_{(2^n-1,2^n-1)} := \bigwedge_{k=0}^{n-1} x_i = 1 \wedge \bigwedge_{k=0}^{n-1} y_i = 1$ represent two nodes, namely those at coordinates $(0,0)$ and $(2^n-1, 2^n-1)$. The formulas $\varphi_{north}$ and $\varphi_{east}$ enforce constraints on the coordinates of states, such that going north increases the coordinate encoding using the $x_i$ features by one, leaving the $y_i$ features unchanged, and going east increases coordinate encoding using the $y_i$ features by one, leaving the $x_i$ features unchanged. For every

formula $\varphi$, $\forall east.\varphi$ stands for $\Box(\varphi_E \to \Box\varphi)$ and $\forall north.\varphi$ stands for $\Box(\varphi_N \to \Box\varphi)$.

$$\varphi_{north} := \bigwedge_{k=0}^{n-1} (\bigwedge_{j=0}^{k-1}(x_j = 1)) \to (((x_k = 1) \to \forall north.(x_k = 0)) \land ((x_k = 0) \to \forall north.(x_k = 1)))\land$$

$$\bigwedge_{k=0}^{n-1} (\bigvee_{j=0}^{k-1}(x_j = 0)) \to (((x_k = 1) \to \forall north.(x_k = 1)) \land ((x_k = 0) \to \forall north.(x_k = 0)))\land$$

$$\bigwedge_{k=0}^{n-1} (((y_k = 1) \to \forall north.(y_k = 1)) \land ((y_k = 0) \to \forall north.(y_k = 0)))$$

$$\varphi_{east} := \bigwedge_{k=0}^{n-1} (\bigwedge_{j=0}^{k-1}(y_j = 1)) \to (((y_k = 1) \to \forall east.(y_k = 0)) \land ((y_k = 0) \to \forall east.(y_k = 1)))\land$$

$$\bigwedge_{k=0}^{n-1} (\bigvee_{j=0}^{k-1}(y_j = 0)) \to (((y_k = 1) \to \forall east.(y_k = 1)) \land ((y_k = 0) \to \forall east.(y_k = 0)))\land$$

$$\bigwedge_{k=0}^{n-1} (((x_k = 1) \to \forall east.(x_k = 1)) \land ((x_k = 0) \to \forall east.(x_k = 0)))$$

The problem of deciding whether a domino system $\mathcal{D} = (D, V, H)$, given an initial condition $w_0 \ldots w_{n-1} \in D^n$, can tile a torus of exponential size can be reduced to the problem satisfiability in $q\mathcal{L}$, checking the satisfiability of the set of formulas $T(n, \mathcal{D}, w) = T_n \cup T_{\mathcal{D}} \cup T_w$, where $T_n$ is as above, $T_{\mathcal{D}}$ encodes the domino system, and $T_w$ encodes the initial condition as follows. We define

$$T_{\mathcal{D}} = \{ \quad \Box_g(\bigwedge_{d \in D}(x_d = 1 \lor x_d = 0)),$$
$$\Box_g(\neg(\varphi_N \lor \varphi_E) \to (\bigvee_{d \in D}\varphi_d)),$$
$$\Box_g(\neg(\varphi_N \lor \varphi_E) \to (\bigwedge_{d \in D}\bigwedge_{d' \in D\setminus\{d\}}\neg(\varphi_d \land \varphi_{d'}))),$$
$$\Box_g(\bigwedge_{d \in D}(\varphi_d \to (\forall east.\bigvee_{(d,d') \in H}\varphi_{d'}))),$$
$$\Box_g(\bigwedge_{d \in D}(\varphi_d \to (\forall north.\bigvee_{(d,d') \in V}\varphi_{d'}))) \quad \}$$

where for every $d \in D$, there is a feature $x_d$ and $\varphi_d := x_d = 1$. Finally, we define

$$T_w = \{ \quad \Box_g(\varphi_{(0,0)} \to \varphi_{w_0}), \ldots, \Box_g(\varphi_{(n-1,0)} \to \varphi_{w_{n-1}}) \quad \}$$

The size of $T(n, \mathcal{D}, w)$ is polynomial in the size of the tiling problem instance, that is in $|D| + |H| + |V| + n$. The rest of the proof is analogous to the proof of (Tobies, 2000, Corollary 3.9). The NEXPTIME-hardness of $q\mathcal{L}$ follows from Lemma 3 and (Tobies, 2000, Corollary 3.3) stating the NEXPTIME-hardness of deciding whether a domino system with initial condition can tile a torus of exponential size.

For the complexity of ACR-GNN verification tasks, we observe the following.

1. We reduce the satisfiability problem in (modal) $q\mathcal{L}$ (restricted to graded modal logic + graded universal modality, because it is sufficient to encode the tiling problem) to VT3 in poly-time as follows. Let $\varphi$ be a $q\mathcal{L}$. We build in poly-time an ACR-GNN $\mathcal{A}$ that recognizes all pointed graphs. We have $\varphi$ is satisfiable iff $[[\varphi]] \cap [[\mathcal{A}]] \neq \emptyset$ So VT3 is NEXPTIME-hard.

2. The validity problem of $q\mathcal{L}$ (dual problem of the satisfiability problem, i.e., given a formula $\varphi$, is $\varphi$ true in all pointed graphs $G, u$?) is coNEXPTIME-hard. We reduce the validity problem of $q\mathcal{L}$ to VT2. Let $\varphi$ be a $q\mathcal{L}$ formula. We construct an ACR-GNN $\mathcal{A}$ that accepts all pointed graphs. We have $\varphi$ is valid iff $[[\mathcal{A}]] \subseteq [[\varphi]]$. So VT2 is coNEXPTIME-hard.

3. We reduce the validity problem of $q\mathcal{L}$ to VT1. Let $\psi$ be a $q\mathcal{L}$ formula. (again restricted to graded modal logic + graded global modalities as for point 1). So following Barceló et al. (2020), we can construct in poly-time an ACR-GNN $\mathcal{A}$ that is equivalent to $\psi$ (by Barceló et al. (2020)). We have $\psi$ is valid iff $[[\top]] \subseteq [[\mathcal{A}]]$. So VT1 is coNEXPTIME-hard.

$\square$

**Theorem 16.** *The satisfiability problems with bounded number of vertices are NP-complete, so is ACR-GNN verification task* VT'3, *while the verification tasks* VT'1 *and* VT'2 *are coNP-complete.*

*Proof.* NP upper bound is obtained by guessing a graph with at most $N$ vertices and then check that $\varphi$ holds. The obtained algorithm is non-deterministic, runs in poly-time and decides the satisfiability problem with bounded number of vertices. NP-hardness already holds for $agg$-free formulas by reduction from SAT for propositional logic (the reduction is $mod2expr$, see Lemma 3).

For the complexity of the bounded ACR-GNN verification tasks, we observe the following.

1. NP upper bound is also obtained by guessing a graph with at most $N$ vertices and then check. For the lower bound, we reduce (propositional) SAT to VT'3 in poly-time as follows. Let $\varphi$ be a propositional formula. We build in poly-time an ACR-GNN $\mathcal{A}$ that recognizes all pointed graphs. We have $\varphi$ is satisfiable iff $[[\varphi]] \cap [[\mathcal{A}]] \neq \emptyset$ So VT'3 is NP-hard.

2. coNP upper bound corresponds to a NP upper bound for the dual problem: guessing a graph with at most $N$ vertices which is recognizes by $\mathcal{A}$ but in which $\varphi$ does not hold. The validity problem of propositional logic (dual problem of the satisfiability problem, i.e., given a formula $\varphi$, is $\varphi$ true for all valuations) is coNP-hard. We reduce the validity problem of propositional logic to VT'2. Let $\varphi$ be a propositional formula. We construct an ACR-GNN $\mathcal{A}$ that accepts all pointed graphs. We have $\varphi$ is valid iff $[[\mathcal{A}]] \subseteq [[\varphi]]$. So VT'2 is coNP-hard.

3. coNP upper bound is obtained similarly. For the lower bound, we reduce the validity problem of propositional logic to VT'1. Let $\psi$ be a propositional formula. So following Barceló et al. (2020), we can construct in poly-time an ACR-GNN $\mathcal{A}$ that is equivalent to $\psi$ (by Barceló et al. (2020)). We have $\psi$ is valid iff $[[\top]] \subseteq [[\mathcal{A}]]$. So VT'1 is coNP-hard.

□

# B PROTOTYPE VERIFICATION OF ACR-GNNS

## B.1 PERFORMANCE

We propose the first implementation to serve as a proof of concept, and a baseline for future research. The prototype directly transforms an instance of a ACR-GNN satisfiability problem into a C program. The C program is then verified by the model checker ESBMC (Menezes et al., 2024).

Just to get an idea, we report in Table 1 the performance of our prototype on a very small ACR-GNN $\mathcal{A}_{test}$ (three layers of input and output dimensions of three).

| Number of vertices | Time (s) |
|---|---|
| 1 | 0.089 |
| 2 | 0.103 |
| 3 | 0.845 |
| 4 | 2.576 |
| 5 | 10.406 |
| 6 | 32.667 |

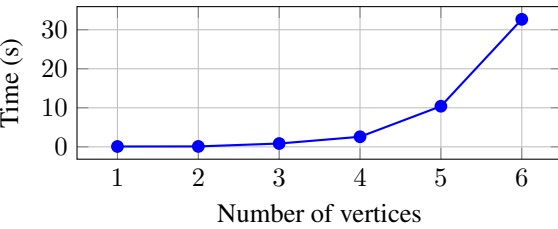

Table 1: Time for solving the ACR-GNN satisfiability problem on the ACR-GNN $\mathcal{A}_{test}$, with varying number of vertices.

Expectedly, the experimental results reveal a bad scalability. Efficient encoding into Satisfiability Modulo Theory (SMT) is a research area of its own, and we hope that the machine learning and verification research communities will find interesting the challenge of making GNN verification practical.

### B.2 CHECKING DISTRIBUTIVITY

We provide C source code for checking distributivity. The reader may run the model checker ESBMC on it to see whether distributivity holds or not.

## C EXTENSION OF LOGIC $K^\sharp$ AND ACR-GNNS OVER $\mathbb{Z}$

A *(labeled directed) graph* $G$ is a tuple $(V, E, \ell)$ such that $V$ is a finite set of vertices, $E \subseteq V \times V$ a set of directed edges and $\ell$ is a mapping from $V$ to a valuation over a set of atomic propositions. We write $\ell(u)(p) = 1$ when atomic proposition $p$ is true in $u$, and $\ell(u)(p) = 0$ otherwise. Given a graph $G$ and vertex $u \in V$, we call $(G, u)$ a *pointed graph*.

### C.1 LOGIC

Consider a countable set $Ap$ of propositions. We define the language of logic $K^{\sharp, \sharp_g}$ as the set of formulas generated by the following BNF:

$$\varphi ::= p \mid \neg\varphi \mid \varphi \vee \varphi \mid \xi \geq 0$$
$$\xi ::= c \mid \mathbb{1}\varphi \mid \sharp\varphi \mid \sharp_g\varphi \mid \xi + \xi \mid c \times \xi$$

where $p$ ranges over $Ap$, and $c$ ranges over $\mathbb{Z}$. We assume that all formulas $\varphi$ are represented as directed acyclic graph (DAG) and refer by *the size of* $\varphi$ to the size of its DAG representation.

Atomic formulas are propositions $p$, inequalities and equalities of linear expressions. We consider linear expressions over $\mathbb{1}\varphi$ and $\sharp\varphi$ and $\sharp_g\varphi$. The number $\mathbb{1}\varphi$ is equal to 1 if $\varphi$ holds in the current world and equal 0 otherwise. The number $\sharp\varphi$ is the number of successors in which $\varphi$ hold. The number $\sharp_g\varphi$ is the number of worlds in the model in which $\varphi$ hold. The language seems strict but we write $\xi_1 \leq \xi_2$ for $\xi_2 - \xi_1 \geq 0$, $\xi = 0$ for $(\xi \geq 0) \wedge (-\xi \geq 0)$, etc.

As in modal logic, a formula $\varphi$ is evaluated in a pointed graph $(G, u)$ (also known as pointed Kripke model). We define the truth conditions $(G, u) \models \varphi$ ($\varphi$ is true in $u$) by

$$
\begin{aligned}
(G, u) &\models p && \text{if} && \ell(u)(p) = 1, \\
(G, u) &\models \neg\varphi && \text{if} && \text{it is not the case that } (G, u) \models \varphi, \\
(G, u) &\models \varphi \wedge \psi && \text{if} && (G, u) \models \varphi \text{ and } (G, u) \models \psi, \\
(G, u) &\models \xi \geq 0 && \text{if} && [[\xi]]_{G,u} \geq 0,
\end{aligned}
$$

and the semantics $[[\xi]]_{G,u}$ (the value of $\xi$ in $u$) of an expression $\xi$ by mutual induction on $\varphi$ and $\xi$ as follows.

$$
\begin{aligned}
[[c]]_{G,u} &= c, \\
[[\xi_1 + \xi_2]]_{G,u} &= [[\xi_1]]_{G,u} + [[\xi_2]]_{G,u}, \\
[[c \times \xi]]_{G,u} &= c \times [[\xi]]_{G,u}, \\
[[\mathbb{1}\varphi]]_{G,u} &= \begin{cases} 1 & \text{if } (G, u) \models \varphi \\ 0 & \text{otherwise,} \end{cases} \\
[[\sharp\varphi]]_{G,u} &= |\{v \in V \mid (u, v) \in E \text{ and } (G, v) \models \varphi\}| \\
[[\sharp_g\varphi]]_{G,u} &= |\{v \in V \mid (G, v) \models \varphi\}|.
\end{aligned}
$$

A local modality $\square\varphi$ can be defined as $\square\varphi := (-1) \times \sharp(\neg\varphi) \geq 0$. That is, to say that $\varphi$ holds in all successors, we say that the number of successors in which $\neg\varphi$ holds is zero. Similarly, a global/universal modality can be defined as $\square_g\varphi := (-1) \times \sharp_g(\neg\varphi) \geq 0$.

### C.2 AGGREGATE-COMBINE GRAPH NEURAL NETWORKS

In this section, we consider a detailed definition of quantized (global) Aggregate-Combine GNNs (ACR-GNN) Barceló et al. (2020), also called message passing neural networks Gilmer et al. (2017).

A *(global) ACR-GNN layer* $\mathcal{L} = (comb, agg, agg_\forall)$ is a tuple where $comb : \mathbb{K}^{3m} \to \mathbb{K}^{m'}$ is a so-called *combination function*, $agg$ is a so-called *local aggregation function*, mapping multisets of vectors from $\mathbb{R}^m$ to a single vector from $\mathbb{K}^{m'}$, $agg_\forall$ is a so-called *global aggregation function*,

also mapping multisets of vectors from $\mathbb{K}^m$ to a single vector from $\mathbb{K}^n$. We call $m$ the *input dimension* of layer $\mathcal{L}$ and $m'$ the *output dimension* of layer $\mathcal{L}$. Then, a *(global) ACR-GNN* is a tuple $(\mathcal{L}^{(1)}, \ldots, \mathcal{L}^{(L)}, cls)$ where $\mathcal{L}^{(1)}, \ldots, \mathcal{L}^{(L)}$ are $L$ ACR-GNN layers and $cls : \mathbb{K}^m \to \{0, 1\}$ is a *classification function*. We assume that all GNNs are well-formed in the sense that output dimension of layer $\mathcal{L}^{(i)}$ matches input dimension of layer $\mathcal{L}^{(i+1)}$ as well as output dimension of $\mathcal{L}^{(L)}$ matches input dimension of $cls$.

Let $G = (V, E)$ be a graph with atomic propositions $p_1, \ldots, p_k$ and $\mathcal{A} = (\mathcal{L}^{(1)}, \ldots, \mathcal{L}^{(L)}, cls)$ an ACR-GNN. We define $x_0 : V \to \{0, 1\}^k$, called the *initial state of $G$*, as $x_0(u) := (\ell(u)(p_1), \ldots, \ell(u)(p_k))$ for all $u \in V$. Then, the $i$-th layer of $\mathcal{A}$ computes an updated state of $G$ by

$$x_i(u) := comb(x_{i-1}(u), agg(\{\{x_{i-1}(v) \mid uv \in E\}\}), agg_\forall(\{\{x_{i-1}(v) \mid v \in V\}\}))$$

where $agg$, $agg_\forall$, and $comb$ are respectively the local aggregation, global aggregation and combination function of the $i$-th layer. Let $(G, u)$ be a pointed graph. We write $\mathcal{A}(G, u)$ to denote the application of $\mathcal{A}$ to $(G, u)$, which is formally defined as $\mathcal{A}(G, u) = cls(x_L(u))$ where $x_L$ is the state of $G$ computed by $\mathcal{A}$ after layer $L$. Informally, this corresponds to a binary classification of vertex $u$.

We consider the following form of ACR-GNN $\mathcal{A}$: all local and global aggregation functions are given by the sum of all vectors in the input multiset, all combination functions are given by $comb(x, y, z) = \vec{\sigma}(xC + yA_1 + zA_2 + b)$ where $\vec{\sigma}(x)$ is the componentwise application of the activation function $\sigma(x)$ with matrices $C$, $A_1$ and $A_2$ and vector $b$ of $\mathbb{K}$ parameters, and where the classification function is $cls(x) = \sum_i a_i x_i \geq 1$, where $a_i$ are from $\mathbb{K}$ as well.

We note $[[\mathcal{A}]]$ the set of pointed graphs $(G, u)$ such that $\mathcal{A}(G, u) = 1$. An ACR-GNN $\mathcal{A}$ is satisfiable if $[[\mathcal{A}]]$ is non-empty. The *satisfiability problem* for ACR-GNNs is: Given a ACR-GNN $\mathcal{A}$, decide whether $\mathcal{A}$ is satisfiable.

# D    CAPTURING GNNs WITH $K^{\sharp, \sharp_g}$

In this section, we exclusively consider ACR-GNNs where $\mathbb{K} = \mathbb{Z}$ and $\sigma$ is *truncated ReLU* $\sigma(x) = max(0, min(1, x))$.

We demonstrate that the expressive power of (global) ACR-GNNs over $\mathbb{Z}$, with truncated ReLU activation functions, and $K^{\sharp, \sharp_g}$, is equivalent. Informally, this means that for every formula $\varphi$ of $K^{\sharp, \sharp_g}$, there exists an ACR-GNNs $\mathcal{A}$ that expresses the same query, and vice-versa. To achieve this, we define a translation of one into the other and substantiate that this translation is efficient. This enables ways to employ $K^{\sharp, \sharp_g}$ for reasoning about ACR-GNN.

We begin by showing that global ACR-GNNs are at least as expressive as $K^{\sharp, \sharp_g}$. We remark that the arguments are similar to the proof of Theorem 1 in Nunn et al. (2024).

**Theorem 18.** *Let $\varphi \in K^{\sharp, \sharp_g}$ be a formula. There is an ACR-GNN $\mathcal{A}_\varphi$ such that for all pointed graphs $(G, u)$ we have $(G, u) \models \varphi$ if and only if $\mathcal{A}_\varphi(G, u) = 1$. Furthermore, $\mathcal{A}_\varphi$ can be built in polynomial time regarding the size of $\varphi$.*

*Proof sketch.* We construct a GNN $\mathcal{A}_\varphi$ that evaluates the semantics of a given $K^{\sharp, \sharp_g}$ formula $\varphi$ for some given pointed graph $(G, v)$. The network consists of $n$ layers, one for each of the $n$ subformulas $\varphi_i$ of $\varphi$, ordered so that the subformulas are evaluated based on subformula inclusion. The first layer evaluates atomic propositions, and each subsequent messages passing layer $l_i$ uses a fixed combination and fixed aggregation function to evaluate the semantics of $\varphi_i$.

The correctness follows by induction on the layers: the $i$-th layer correctly evaluates $\varphi_i$ at each vertex of $G$, assuming all its subformulas are correctly evaluated in previous layers. Finally, the classifying function $cls$ checks whether the $n$-th dimension of the vector after layer $l_n$, corresponding to the semantics of $\varphi_n$ for the respective vertex $v$, indicates that $\varphi_n = \varphi$ is satisfied by $(G, v)$. The network size is polynomial in the size of $\varphi$ due to the fact that the total number of layers and their width is polynomially bounded by the number of subformulas of $\varphi$. A full formal proof is given in Appendix D.1. $\qquad \square$

**Theorem 19.** *Let $\mathcal{A}$ be a GNN. We can compute in polynomial time wrt. $|\mathcal{A}|$ a $K^{\sharp,\sharp_g}$-formula $\varphi_{\mathcal{A}}$, represented as a DAG, such that $[[\mathcal{A}]] = [[\varphi_{\mathcal{A}}]]$.*

*Proof sketch.* We construct a $K^{\sharp,\sharp_g}$-formula $\varphi_{\mathcal{A}}$ that simulates the computation of a given GNN $\mathcal{A}$. For each layer $l_i$ of the GNN, we define a set of formulas $\varphi_{i,j}$, one per output dimension, that encode the corresponding node features using linear threshold expressions over the formulas from the previous layer. At the base, the input features are the atomic propositions $p_1, \ldots, p_{m_1}$.

Each formula $\varphi_{i,j}$ mirrors the computation of the GNN layer, including combination, local aggregation, and global aggregation. The final classification formula $\varphi_{\mathcal{A}}$ encodes the output of the linear classifier on the top layer features. Correctness follows from the fact that all intermediate node features remain Boolean under message passing layers with integer parameters and truncated ReLU activations. This allows expressing each output as a Boolean formula over the input propositions. The construction is efficient: by reusing shared subformulas via a DAG representation, the total size remains polynomial in the size of $\mathcal{A}$. A more complete proof is given in Appendix D.2. □

### D.1 PROOF OF THEOREM 18

*Proof of Theorem 18.* Let $\varphi$ be a $K^{\sharp,\sharp_g}$ formula over the set of atomic propositions $p_1, \ldots, p_m$. Let $\varphi_1, \ldots, \varphi_n$ denote an enumeration of the subformulas of $\varphi$ such that $\varphi_i = p_i$ for $i \leq m$, $\varphi_n = \varphi$, and whenever $\varphi_i$ is a subformula of $\varphi_j$, it holds that $i \leq j$. Without loss of generality, we assume that all subformulas of the form $\xi \geq 0$ are written as

$$\sum_{j \in J} k_j \cdot \mathbb{1}\varphi_j + \sum_{j' \in J'} k_{j'} \cdot \sharp\varphi_{j'} + \sum_{j'' \in J''} k_{j''} \cdot \sharp_g\varphi_{j''} - c \geq 0,$$

for some index sets $J, J', J'' \subseteq \{1, \ldots, n\}$.

We construct the GNN $\mathcal{A}_{\varphi}$ in a layered manner. Note that $\mathcal{A}_{\varphi}$ is fully specified by defining the combination function $comb_i$, including its local and global aggregation, for each layer $l_i$ with $i \in \{1, \ldots, n\}$ and the final classification function $cls$. Each $comb_i$ produces output vectors of dimension $n$. The first layer has input dimension $m$, and $comb_1$ is defined by $comb_1(x, y, z) = (x, 0, \ldots, 0)$, ensuring that the first $m$ dimensions correspond to the truth values of the atomic propositions $p_1, \ldots, p_m$, while the remaining entries are initialized to zero. Note that $comb_1$ is easily realized by an FNN with trReLU activations. For $i > 1$, the combination function $comb_i$ is defined as

$$comb_i(x, y, z) = \vec{\sigma}(xC + yA_1 + zA_2 + b),$$

where $C, A_1, A_2$ are $n \times n$ matrices corresponding to self, local (neighbor), and global aggregation respectively, and $b \in \mathbb{Z}^n$ is a bias vector. The parameters are defined sparsely as follows:

- $C_{ii} = 1$ for all $i \leq m$ (preserving the atomic propositions),

- If $\varphi_i = \neg\varphi_j$, then $C_{ji} = -1$ and $b_i = 1$,

- If $\varphi_i = \varphi_j \vee \varphi_l$, then $C_{ji} = C_{li} = 1$, and

- If $\varphi_i = \sum_{j \in J} k_j \cdot 1_{\varphi_j} + \sum_{j' \in J'} k_{j'} \cdot \sharp\varphi_{j'} + \sum_{j'' \in J''} k_{j''} \cdot \sharp_g\varphi_{j''} - c \geq 0$, then

$$C_{ji} = k_j, \quad A_{1,j'i} = k_{j'}, \quad A_{2,j''i} = k_{j''}, \quad b_i = -c + 1.$$

Note that each $comb_i$ has the same functional form, differing only in the non-zero entries of its parameters. The classification function is defined by $cls(x) = x_n \geq 1$.

Let $l_i$ denote the $i$th layer of $\mathcal{A}_{\varphi}$, and fix a vertex $v$ in some input graph. We show, by induction on $i$, that the following invariant holds: for all $j \leq i$, $(x_i(v))_j = 1$ if and only if $v \models \varphi_j$, and $(x_i(v))_j = 0$ otherwise. Assume that $i = 1$. By construction, $x_1(v)$ contains the truth values of the atomic propositions $p_1, \ldots, p_m$ in its first $m$ coordinates. Thus, the statement holds at layer 1. Next, assume the statement holds for layer $x_{i-1}$. Let $j < i$. By assumption, the semantics of $\varphi_j$ are already correctly encoded in $x_{j-1}$ and preserved by $comb_i$ due to the fixed structure of $C, A_1, A_2$, and $b$. Now consider $j = i$. The semantics of all subformulas of $\varphi_i$ are captured in $x_{i-1}$, either at the current vertex or its neighbors. By the design of $comb_i$, which depends only on the values of

relevant subformulas, we conclude that $\varphi_i$ is correctly evaluated. This holds regardless of whether $\varphi_i$ is a negation, disjunction, or numeric threshold formula. Thus, the statement holds for all $i$, and in particular for $x_n(v)$ and $\varphi_n = \varphi$. Finally, the classifier $cls$ evaluates whether $x_n(v)_n \geq 1$, which is equivalent to $G, v \models \varphi$. The size claim is obvious given that $n$ depends polynomial on the size of $\varphi$. We note that this assumes that the enumeration of subformulas of $\varphi$ does not contain duplicates. $\quad\square$

## D.2 Proof of Theorem 19

*Proof of Theorem 19.* Let $\mathcal{A}$ be a GNN composed of layers $l_1, \ldots, l_k$, where each $comb_i$ has input dimension $2m_i$, output dimension $n_i$, and parameters $C_i$, $A_{i,1}$, $A_{i,2}$, and $b_i$. The final classification is defined via a linear threshold function $cls(x) = a_1 x_1 + \cdots + a_{n_k} x_{n_k} \geq 1$. We assume that the dimensionalities match across layers, i.e., $m_i = n_{i-1}$ for all $i \geq 2$, so that the GNN is well-formed.

We construct a formula $\varphi_{\mathcal{A}}$ over the input propositions $p_1, \ldots, p_{m_1}$ inductively, mirroring the structure of the GNN computation.

We begin with the first layer $l_1$. For each $j \in \{1, \ldots, n_1\}$, we define:

$$\varphi_{1,j} = \sum_{k=1}^{m_1} (C_1)_{kj} \cdot \mathbb{1} p_k + (A_{1,1})_{kj} \cdot \sharp p_k + (A_{1,2})_{kj} \cdot \sharp_g p_k + (b_1)_j \geq 1.$$

Now suppose that we have already constructed formulas $\varphi_{i-1,1}, \ldots, \varphi_{i-1,n_{i-1}}$ for some layer $i \geq 2$. Then, for each output index $j \in \{1, \ldots, n_i\}$, we define:

$$\varphi_{i,j} = \sum_{k=1}^{m_i} (C_i)_{kj} \cdot \mathbb{1} \varphi_{i-1,k} + (A_{i,1})_{kj} \cdot \sharp \varphi_{i-1,k} + (A_{i,2})_{kj} \cdot \sharp_g \varphi_{i-1,k} + (b_i)_j \geq 1.$$

Once all layers have been encoded in this way, we define the final classification formula as

$$\varphi_{\mathcal{A}} = a_1 \mathbb{1} \varphi_{k,1} + \cdots + a_{n_k} \mathbb{1} \varphi_{k,n_k} \geq 1.$$

Let $G, v$ be a pointed graph. The correctness of our translation follows directly from the following observations: all weights and biases in $\mathcal{A}$ are integers, and the input vectors $x_0(u)$ assigned to nodes $u$ in $G$ are Boolean. Moreover, each layer applies a linear transformation followed by a pointwise truncated ReLU, which preserves the Boolean nature of the node features. It follows that the intermediate representations $x_i(v)$ remain in $\{0,1\}^{n_i}$ for all $i$. Consequently, each such feature vector can be expressed via a set of Boolean $K^{\sharp, \sharp_g}$-formulas as constructed above. Taken together, this ensures that the overall formula $\varphi_{\mathcal{A}}$ faithfully simulates the GNN's computation.

It remains to argue that this construction can be carried out efficiently. Throughout, we represent the (sub)formulas using a shared DAG structure, avoiding duplication of equivalent subterms. This ensures that subformulas $\varphi_{i-1,k}$ can be reused without recomputation. For each layer, constructing all $\varphi_{i,j}$ requires at most $n_i \cdot m_i$ steps, plus the same order of additional operations to account for global aggregation terms. Since the number of layers, dimensions, and parameters are bounded by $|\mathcal{A}|$, and each operation can be performed in constant or linear time, the total construction is polynomial in the size of $\mathcal{A}$. $\quad\square$

# E DESCRIPTION LOGICS WITH CARDINALITY CONSTRAINTS

## E.1 $\mathcal{ALCQ}$ AND $T_C$BOXES CONSISTENCY

$\mathcal{ALCQ}$ is the Description Logic adding qualified number restrictions to the standard Description Logic $\mathcal{ALC}$, analogously to how Graded Modal Logic extends standard Modal Logic with graded modalities.

Let $N_C$ and $N_R$ be two non-intersecting sets of concept names, and role names respecively. A concept name $A \in N_C$ is an $\mathcal{ALCQ}$ concept expressions of $\mathcal{ALCQ}$. If $C$ is an $\mathcal{ALCQ}$ concept expression, so is $\neg C$. If $C_1$ and $C_2$ are $\mathcal{ALCQ}$ concept expressions, then so is $C_1 \sqcap C_2$. If $C$ is an $\mathcal{ALCQ}$ concept expression, $R \in N_R$, and $n \in \mathbb{N}$, then $\geq n\,R.C$ is an $\mathcal{ALCQ}$ concept expression.

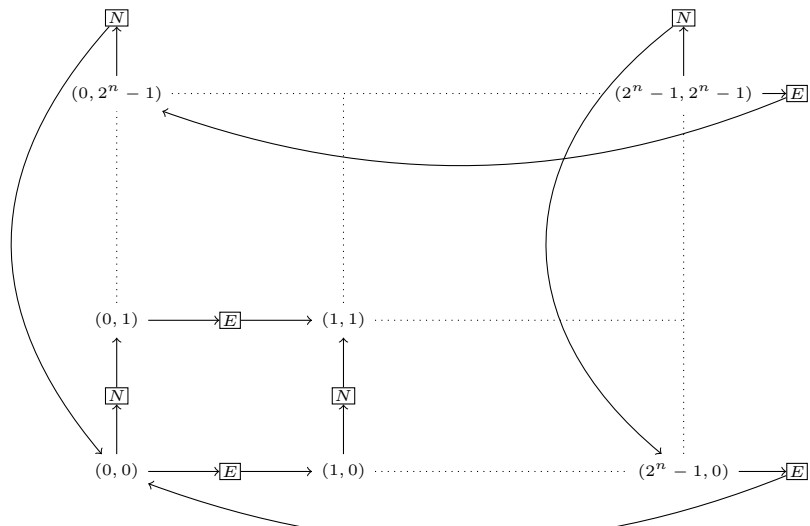

Figure 3: Encoding a torus of exponential size with an $\mathcal{ALCQ}$-$T_C$Box with one role.

A *cardinality restriction* of $\mathcal{ALCQ}$ is is an expression of the form $(\geq\ n\ C)$ or $(\leq\ n\ C)$, where $C$ an $\mathcal{ALCQ}$ concept expression and $n \in \mathbb{N}$.

An $\mathcal{ALCQ}$-$T_C$Box is a finite set of cardinality restrictions.

An *interpretation* is a pair $I = (\Delta^I, \cdot^I)$, where $\Delta^I$ is a non-empty set of individuals, and $\cdot^I$ is a function such that: every $A \in N_C$ is mapped to $A^I \subseteq \Delta^I$, and every $R \in N_R$ is mapped to $R^I \subseteq \Delta^I \times \Delta^I$. Given an element of $d \in \Delta^I$, we define $R^I(d) = \{d' \mid (d, d') \in R^I\}$. An interpretation $I$ is extended to complex concept descriptions as follows: $(\neg C)^I = \Delta^I \setminus C^I$; $(C_1 \sqcap C_2)^I = C_1^I \cap C_2^I$; and $(\geq\ n\ R.C)^I = \{d \mid |R^I(d) \cap C^I| \geq n\}$.

An interpretation $I$ satisfies the cardinality restriction $(\geq\ n\ C)$ iff $|C^I| \geq n$ and it satisfies the cardinality restriction $(\leq\ n\ C)$ iff $|C^I| \leq n$. A $T_C$Box $TC$ is *consistent* if there exists an interpretation that satisfies all the cardinality restrictions in $TC$.

**Theorem 20** (Tobies (2000)). *Deciding the consistency of $\mathcal{ALCQ}$-$T_C$Boxes is NEXPTIME-hard.*

The proof can be slightly adapted to show that the result holds even when there is only one role.

Some abbreviations are useful. For every pair of concepts $C$ and $D$, $C \to D$ stands for $\neg C \sqcup D$. For every concept $C$, role $R$, and non-negative integer $n$, we define: $(\leq n\ R.C) := \neg(\geq (n + 1)\ R.C)$, $(\forall\ R.C) := (\leq 0\ R.\neg C)$, $(\forall\ C) := (\leq 0\ \neg C)$, $(= n\ R.C) := (\geq n\ R.C) \sqcap (\leq n\ R.C)$, and $(= n\ C) := (\geq n\ C) \sqcap (\leq n\ C)$.

**Theorem 21.** *Deciding the consistency of $\mathcal{ALCQ}$-$T_C$Boxes is NEXPTIME-hard even if $|N_R| = 1$.*

*Proof.* Let $next$ be the unique role in $N_R$. We use the atomic concepts $N$ to denote an individual 'on the way north' and $E$ to denote an individual 'on the way east'. See Figure 3.

For every $n \in \mathbb{N}$, we define the following $\mathcal{ALCQ}$-$T_C$Box.

$$
\begin{aligned}
T_n = \{ \quad & (\forall \neg(N \sqcup E) \to (= 1\ next.N)) & , & \quad & (\forall \neg(N \sqcup E) \to (= 1\ next.E)) \\
& (\forall\ N \to (= 1\ next.\top)) & , & & (\forall\ E \to (= 1\ next.\top)) \\
& (= 1\ C_{(0,0)}) & , & & (= 1\ C_{(2^n-1,2^n-1)}) \\
& (\forall \neg(N \sqcup E) \to D_{east}) & , & & (\forall \neg(N \sqcup E) \to D_{north}) \\
& (\leq (2^n \times 2^n)\ \neg(N \sqcup E)), & & (\leq (2^n \times 2^n)\ N), & (\leq (2^n \times 2^n)\ E) \quad \}
\end{aligned}
$$

such that the concepts $C_{(0,0)}$, $C_{(2^n-1,2^n-1)}$ are defined like in (Tobies, 2000, Figure 3), and so are the concepts $D_{north}$ and $D_{east}$, except that for every concept $C$, $\forall east.C$ now stands for $\forall next.(E \to \forall next.C)$ and $\forall north.C$ now stands for $\forall next.(N \to \forall next.C)$.

The problem of deciding whether a domino system $\mathcal{D} = (D, V, H)$, given an initial condition $w_0 \ldots w_{n-1}$, can tile a torus of exponential size can be reduced to the problem of consistency of $\mathcal{ALCQ}\text{-}T_C$Boxes, checking the consistency of $T(n, \mathcal{D}, w) = T_n \cup T_{\mathcal{D}} \cup T_w$, where $T_n$ is as above, $T_{\mathcal{D}}$ encodes the domino system, and $T_w$ encodes the initial condition as follows.

$$T_{\mathcal{D}} = \{ \quad (\forall \neg (N \sqcup E) \rightarrow (\bigsqcup_{d \in D} C_d)),$$
$$(\forall \neg (N \sqcup E) \rightarrow (\bigsqcap_{d \in D} \bigsqcap_{d' \in D \setminus \{d\}} \neg(C_d \sqcap C_{d'}))),$$
$$(\forall \bigsqcap_{d \in D}(C_d \rightarrow (\forall east. \bigsqcup_{(d,d') \in H} C_{d'}))),$$
$$(\forall \bigsqcap_{d \in D}(C_d \rightarrow (\forall north. \bigsqcup_{(d,d') \in V} C_{d'}))) \quad \}$$

$$T_w = \{ \quad (\forall C_{(0,0)} \rightarrow C_{w_0}), \ldots, (\forall C_{(n-1,0)} \rightarrow C_{w_{n-1}}) \quad \}$$

The rest of the proof remains unchanged. $\qquad \square$

### E.2  DESCRIPTION LOGICS WITH GLOBAL AND LOCAL CARDINALITY CONSTRAINTS

The Description Logic $\mathcal{ALCSCC}^{++}$ (Baader et al., 2020) extends the basic Description Logic $\mathcal{ALC}$ (Baader et al., 2017) with concepts that capture cardinality and set constraints expressed in the quantifier-free fragment of Boolean Algebra with Presburger Arithmetic (QFBAPA) (Kuncak & Rinard, 2007).

We assume that we have a set of *set variables* and a set of *integer constants*.

#### E.2.1  QFBAPA

A QFBAPA *formula* is a Boolean combination ($\wedge$, $\vee$, $\neg$) of *set constraints* and *cardinality constraints*.

A *set term* is a Boolean combination ($\cup$, $\cap$, $\overline{\cdot}$) of *set variables*, and *set constants* $\mathcal{U}$, and $\emptyset$. If $S$ is a set term, then its cardinality $|S|$ is an *arithmetic expressions*. Integer constants are also arithmetic expressions. If $T_1$ and $T_2$ are arithmetic expressions, so is $T_1 + T_2$. If $T$ is an arithmetic expression and $c$ is an integer constant, then $c \cdot T$ is an arithmetic expression.

Given two set terms $B_1$ and $B_2$, the expressions $B_1 \subseteq B_2$ and $B_1 = B_2$ are *set constraints*. Given two arithmetic expressions $T_1$ and $T_2$, the expressions $T_1 < T_2$ and $T_1 = T_2$ are *cardinality constraints*. Given an integer constant $c$ and an arithmetic expression $T$, the expression $c \; dvd \; T$ is a *cardinality constraint*.

A *substitution* $\sigma$ assigns $\emptyset$ to the set constant $\emptyset$, a finite set $\sigma(\mathcal{U})$ to the set constant $\mathcal{U}$, and a subset of $\sigma(\mathcal{U})$ to every set variable. A substitution is first extended to set terms by applying the standard set-theoretic semantics of the Boolean operations. It is further extended to map arithmetic expressions to integers, in such that way that every integer constant $c$ is mapped to $c$, for every set term $B$, the arithmetic expression $|B|$ is mapped to the cardinality of the set $\sigma(B)$, and the standard semantics for addition and multiplication is applied.

The substitution $\sigma$ *(QFBAPA) satisfies* the set constraint $B_1 \subseteq B_2$ if $\sigma(B_1) \subseteq \sigma(B_2)$, the set constraint $B_1 = B_2$ if $\sigma(B_1) = \sigma(B_2)$, the cardinality constraint $T_1 < T_2$ if $\sigma(T_1) < \sigma(T_2)$, the cardinality constraint $T_1 = T_2$ if $\sigma(T_1) = \sigma(T_2)$, and the cardinality constraint $c \; dvd \; T$ if $c$ divides $\sigma(T)$.

#### E.2.2  $\mathcal{ALCSCC}^{++}$

We can now define the syntax of $\mathcal{ALCSCC}^{++}$ concept descriptions and their semantics. Let $N_C$ be a set of concept names, and $N_R$ be a set of role names, such that $N_C \cap N_R = \emptyset$. Every $A \in N_C$ is a *concept description* of $\mathcal{ALCSCC}^{++}$. Moreover, if $C, C_1, C_2, \ldots$ are *concept descriptions* of $\mathcal{ALCSCC}^{++}$, then so are: $C_1 \sqcap C_2, C_1 \sqcup C_2, \neg C$, and $\mathsf{sat}(\chi)$, where $\chi$ is a set or cardinality QFBAPA constraint, with elements of $N_R$ and concept descriptions $C_1, C_2, \ldots$ used in place of set variables.

A *finite interpretation* is a pair $I = (\Delta^I, \cdot^I)$, where $\Delta^I$ is a finite non-empty set of individuals, and $\cdot^I$ is a function such that: every $A \in N_C$ is mapped to $A^I \subseteq \Delta^I$, and every $R \in N_R$ is mapped to $R^I \subseteq \Delta^I \times \Delta^I$. Given an element of $d \in \Delta^I$, we define $R^I(d) = \{d' \mid (d, d') \in R^I\}$.

The semantics of the language of $\mathcal{ALCSCC}^{++}$ makes use QFBAPA substitutions to interpret QFBAPA constraints in terms of $\mathcal{ALCSCC}^{++}$ finite interpretations. Given an element $d \in \Delta^I$, we can define

the substitution $\sigma_d^I$ in such a way that: $\sigma_d^I(\mathcal{U}) = \Delta^I$, $\sigma_d^I(\emptyset) = \emptyset$, and $A \in N_C$ and $R \in N_R$ are considered QFBAPA set variables and substituted as $\sigma_d^I(A) = A^I$, and $\sigma_d^I(R) = R^I(d)$.

The finite interpretation $I$ and the QFBAPA substitutions $\sigma_d^I$ are mutually extended to complex expressions such that: $\sigma_d^I(C_1 \sqcap C_2) = (C_1 \sqcap C_2)^I = C_1^I \cap C_2^I$; $\sigma_d^I(C_1 \sqcup C_2) = (C_1 \sqcup C_2)^I = C_1^I \cup C_2^I$; $\sigma_d^I(\neg C) = (\neg C)^I = \Delta^I \setminus C^I$; and $\sigma_d^I(\mathsf{sat}(\chi)) = (\mathsf{sat}(\chi))^I = \{d' \in \Delta^I \mid \sigma_{d'}^I \text{ (QFBAPA) satisfies } \chi\}$.

**Definition 22.** *The $\mathcal{ALCSCC}^{++}$ concept description $C$ is satisfiable if there is a finite interpretation $I$ such that $C^I \neq \emptyset$.*

**Theorem 23** (Baader et al. (2020))**.** *The problem of deciding whether an $\mathcal{ALCSCC}^{++}$ concept description is satisfiable is NEXPTIME-complete.*

## F    COMPLEXITY OF THE SATISFIABILITY OF $K^{\sharp, \sharp_g}$ AND ITS IMPLICATIONS FOR ACR-GNN VERIFICATION

In this section, we establish the complexity of reasoning with $K^{\sharp, \sharp_g}$.

Instrumentally, we first show that every $K^{\sharp, \sharp_g}$ formula can be translated into a $K^{\sharp, \sharp_g}$ formula that is equi-satisfiable, and has a tree representation of size at most polynomial in the size of the original formula. An analogous result was obtained in Nunn et al. (2024) for $K^{\sharp}$. It can be shown using a technique reminiscent of Tseitin (1983) and consisting of factorizing subformulas that are reused in the DAG by introducing a fresh proposition that is made equivalent. Instead of reusing a 'possibly large' subformula, a formula then reuses the equivalent 'small' atomic proposition.

**Lemma 24.** *The satisfiability problem of $K^{\sharp, \sharp_g}$ reduces to the satisfiability of $K^{\sharp, \sharp_g}$ with tree formulas in polynomial time.*

*Proof.* Let $\varphi$ be a $K^{\sharp, \sharp_g}$ formula represented as a DAG. For every subformula $\psi$ (i.e., for every node in the DAG representation of $\varphi$), we introduce a fresh atomic proposition $p_\psi$. We can capture the meaning of these new atomic propositions with the formula $\Phi := \bigwedge_{\psi \text{ node in the DAG}} sem(\psi)$ where:

$$sem(\psi \vee \chi) := p_{\psi \vee \chi} \leftrightarrow (p_\psi \vee p_\chi)$$
$$sem(\neg \psi) := p_{\neg \psi} \leftrightarrow \neg p_\psi$$
$$sem(\xi \geq 0) := p_{\xi \geq 0} \leftrightarrow \xi' \geq 0$$

$$(c)' := c \quad (\xi_1 + \xi_2)' := \xi_1' + \xi_2' \quad (c \times \xi)' := c \times \xi'$$
$$(\mathbb{1}\psi)' := \mathbb{1}p_\psi \quad (\sharp \psi)' := \sharp p_\psi \quad (\sharp_g \psi)' := \sharp_g p_\psi$$

Now, define $\varphi_t := p_\varphi \wedge \Box_g \Phi$, where $\Box_g \Phi := (-1) \times \sharp_g(\neg \Phi) \geq 0$, enforcing the truth of $\Phi$ in every vertex. The size of its tree representation is polynomial in the size of $\varphi$. Moreover, $\varphi_t$ is satisfiable iff $\varphi$ is satisfiable.

$\square$

**Theorem 25.** *The satisfiability problem of $K^{\sharp, \sharp_g}$ with tree formulas is NEXPTIME-complete.*

*Proof.* For membership, we translate the problem into the NEXPTIME-complete problem of concept description satisfiability in the Description Logics with Global and Local Cardinality Constraints Baader et al. (2020), noted $\mathcal{ALCSCC}^{++}$. The Description Logic $\mathcal{ALCSCC}^{++}$ uses the Boolean Algebra with Presburger Arithmetic Kuncak & Rinard (2007), noted QFBAPA, to formalize cardinality constraints. See Section E.2 for a presentation of $\mathcal{ALCSCC}^{++}$ and QFBAPA.

Let $\varphi_0$ be a $K^{\sharp, \sharp_g}$ formula.

For every proposition $p$ occurring in $\varphi_0$, let $A_p$ be an $\mathcal{ALCSCC}^{++}$ concept name. Let $R$ be an $\mathcal{ALCSCC}^{++}$ role name. For every occurrence of $\mathbb{1}\varphi$ in $\varphi_0$, let $ZOO_\varphi$ be an $\mathcal{ALCSCC}^{++}$ role name. $ZOO$-roles stand for 'zero or one'. The rationale for introducing $ZOO$-roles is to be able to capture the value of $\mathbb{1}\varphi$ in $\mathcal{ALCSCC}^{++}$ making it equal to the number of successors of the role $ZOO_\varphi$ which

can then be used in QFBAPA constraints. A similar trick was used, in another context, in Galliani et al. (2023). Here, we enforce this with the QFBAPA constraint

$$\chi_0 = \bigwedge_{\mathbb{1}\varphi \in \varphi_0} \left( (|ZOO_\varphi| = 0 \vee |ZOO_\varphi| = 1) \wedge \overline{\tau}(\varphi) = \mathsf{sat}(|ZOO_\varphi| = 1) \right)$$

which states that $ZOO_\varphi$ has zero or one successor, and has one successor exactly when (the translation of) $\varphi$ is true. The concept descriptions $\overline{\tau}(\varphi)$ and arithmetic expressions $\overline{\tau}(\xi)$ are defined inductively as follows:

$$\begin{array}{rcl}
\overline{\tau}(p) & = & A_p \\
\overline{\tau}(\neg\varphi) & = & \neg\overline{\tau}(\varphi) \\
\overline{\tau}(\varphi \vee \psi) & = & \overline{\tau}(\varphi) \sqcup \overline{\tau}(\psi) \\
\overline{\tau}(\xi \geq 0) & = & \mathsf{sat}(-1 < \overline{\tau}(\xi)) \\
\overline{\tau}(c) & = & c \\
\overline{\tau}(\xi_1 + \xi_2) & = & \overline{\tau}(\xi_1) + \overline{\tau}(\xi_2) \\
\overline{\tau}(c \times \xi) & = & \overline{\tau}(c \cdot \xi) \\
\overline{\tau}(\sharp\varphi) & = & |R \cap \overline{\tau}(\varphi)| \\
\overline{\tau}(\mathbb{1}\varphi) & = & |ZOO_\varphi| \\
\overline{\tau}(\sharp_g\varphi) & = & |\overline{\tau}(\varphi)|
\end{array}$$

Finally, we define the $\mathcal{ALCSCC}^{++}$ concept description $C_{\varphi_0} = \overline{\tau}(\varphi_0) \sqcap \mathsf{sat}(\chi_0)$.

**Claim 26.** *The concept description $C_{\varphi_0}$ is $\mathcal{ALCSCC}^{++}$-satisfiable iff the formula $\varphi_0$ is $K^{\sharp,\sharp_g}$-satisfiable. Moreover, the concept description $C_{\varphi_0}$ has size polynomial in the size of $\varphi_0$.*

*Proof.* From right to left, suppose that $\varphi_0$ is $K^{\sharp,\sharp_g}$-satisfiable. It means that there is a pointed graph $(G, u)$ where $G = (V, E)$ and $u \in V$, such that $(G, u) \models \varphi_0$. Let $I_0 = (\Delta^{I_0}, \cdot^{I_0})$ be the $\mathcal{ALCSCC}^{++}$ interpretation over $N_C$ and $N_R$, such that $N_C = \{A_p \mid p \text{ a proposition in } \varphi_0\}$, $N_R = \{R\} \cup \{ZOO_\varphi \mid \mathbb{1}\varphi \in \varphi_0\}$, $\Delta^{I_0} = V$, $A_p^{I_0} = \{v \mid v \in V, (G, v) \models p\}$ for every $p$ in $\varphi_0$, $R^{I_0} = E$, $ZOO_\varphi^{I_0} = \{(v, v) \mid v \in V, (G, v) \models \varphi\}$ for every $\mathbb{1}\varphi$ in $\varphi_0$. We can show that $u \in C_{\varphi_0}^{I_0}$. Basically $I^0$ is like $G$ with the addition of adequately looping $ZOO$-roles. An individual in $\Delta^{I_0}$ has exactly one $ZOO_\varphi$-successor (itself), exactly when $\varphi$ is true, and no successor otherwise; $A_p$ is true exactly where $p$ is true, and the role $R$ corresponds exactly to $E$.

From left to right, suppose that $C_{\varphi_0}$ is $\mathcal{ALCSCC}^{++}$-satisfiable. It means that there is an $\mathcal{ALCSCC}^{++}$ finite interpretation $I_0 = (\Delta^{I_0}, \cdot^{I_0})$ and an individual $d \in \Delta^{I_0}$ such that $d \in C_{\varphi_0}^{I_0}$. Let $G = (V, E)$ be a graph such that $V = \Delta^{I_0}$, $E = R^{I_0}$, and $\ell(d)(p) = 1$ iff $d \in A_p^{I_0}$. We can show that $(G, d) \models \varphi_0$.

Since there are at most $|\varphi_0|$ subformulas in $\varphi_0$, the representation of $ZOO_\varphi$ for every subformula $\varphi$ of $\varphi_0$ can be done in size $\log_2(|\varphi_0|)$. For every formula $\varphi$, the size of the concept description $\overline{\tau}(\varphi)$ is polynomial (at most $O(n \log(n))$). The overall size of $\overline{\tau}(\varphi_0)$ is polynomial in the size of $\varphi_0$, and so is the size of $\mathsf{sat}(\xi_0)$ (at most $O(n^2(\log(n))^2)$). $\qquad\square$

The NEXPTIME-membership follows from Claim 26 and the fact that the concept satisfiability problem in $\mathcal{ALCSCC}^{++}$ is in NEXPTIME (Theorem 23).

For the hardness, we reduce the problem of consistency of $\mathcal{ALCQ}$-$T_C$Boxes which is NEXPTIME-hard (Tobies, 2000, Corollary 3.9). See Section E.1 and Theorem 21 that slightly adapts Tobies' proof to show that the problem is hard even with only one role.

We define the translation $\underline{\tau}$ from the set of $\mathcal{ALCQ}$ concept expressions and $\mathcal{ALCQ}$ cardinality constraints, with only one role $R$.

$$\begin{array}{rcl}
\underline{\tau}(A) & = & p_A \\
\underline{\tau}(\neg C) & = & \neg\underline{\tau}(C) \\
\underline{\tau}(C_1 \sqcup C_2) & = & \underline{\tau}(C_1) \vee \underline{\tau}(C_2) \\
\underline{\tau}(\geq n\, R.C) & = & \sharp\underline{\tau}(C) + (-1) \times n \geq 0 \\
\underline{\tau}(\geq n\, C) & = & \sharp_g\underline{\tau}(C) + (-1) \times n \geq 0 \\
\underline{\tau}(\leq n\, C) & = & (-1) \times \sharp_g\underline{\tau}(C) + n \geq 0
\end{array}$$

It is routine to check the following claim.

**Claim 27.** *Let $TC$ be an $\mathcal{ALCQ}$-$T_C$Box. $TC$ is consistent iff $\bigwedge_{\chi \in TC} \underline{\tau}(\chi)$ is $K^{\sharp, \sharp_g}$-satisfiable.*

Moreover, the reduction is linear. Hardness thus follows from the NEXPTIME-hardness of consistency of $\mathcal{ALCQ}$-$T_C$Boxes. □

Lemma 24 and Theorem 25 yield the following corollary.

**Corollary 28.** *The $K^{\sharp, \sharp_g}$-satisfiability problem is NEXPTIME-complete.*

Furthermore, from Theorem 18 and Corollary 28, we obtain the complexity of reasoning with ACR-GNNs with truncated ReLU and integer weights.

**Corollary 29.** *Satisfiability of ACR-GNN with global readout, over $\mathbb{Z}$ and with truncated ReLU is NEXPTIME-complete.*

The decidability of the problem is left open in Benedikt et al. (2024) and in the recent long version Benedikt et al. (2025) when the weights are rational numbers. Corollary 29 answers it positively in the case of integer weights and pinpoints the computational complexity.

## G   EXPERIMENTAL DATA AND FURTHER ANALYSES

In this section, we report on the application of dynamic Post-Training Quantization (PTQ) to Aggregate-Combined Readout Graph Neural Networks (ACR-GNNs). Implemented in PyTorch Ansel et al. (2024); PyTorch Team (2024a), dynamic PTQ transforms a pre-trained floating-point model into a quantized version without requiring retraining. In this approach, model weights are statically quantized to INT8, while activations remain in floating-point format until they are dynamically quantized at compute time. This hybrid representation enables efficient low-precision computation using INT8-based matrix operations, thereby reducing memory footprint and improving inference speed. PyTorch's implementation applies per-tensor quantization to weights and stores activations as floating-point values between operations to balance precision and performance.

We adopt INT8 and QINT8 representations as the primary quantization format. According to the theory, INT8 refers to 8-bit signed integers that can encode values in the range $[-128, 127]$. In contrast, QINT8, as defined in the PyTorch documentation Ansel et al. (2024); PyTorch Team (2024b;c), is a quantized tensor format that wraps INT8 values together with quantization metadata: a scale (defining the float value represented by one integer step) and a zero-point (the INT8 value corresponding to a floating-point zero). This additional information allows QINT8 tensors to approximate floating-point representations efficiently while enabling high-throughput inference.

To evaluate the practical impact of quantization, we conducted experiments on both synthetic and real datasets. The synthetic data setup was based on the benchmark introduced by Barceló et al. (2020). Graphs were generated using the dense Erdös–Rényi model, a classical method for constructing random graphs, and each graph was initialized with five node colours encoded as one-hot feature vectors. The dataset is structured as follows, as shown in Table 2. The training set consists of 5000 graphs, each with 40 to 50 nodes and between 560 and 700 edges. The test set is divided into two subsets. The first subset comprises 500 graphs with the same structure as the training set, featuring 40 to 50 nodes and 560 to 700 edges. The second subset contains 500 larger graphs, with 51 to 69 nodes and between 714 and 960 edges. This design allows us to evaluate the model's generalization capability to unseen graph sizes.

For this experiment, we used simple ACR-GNN models with the following specifications. We applied the *sum* function for both the aggregation and readout operations. The combination function was defined as: $comb(x, y, z) = \vec{\sigma}(xC + yA + zR + b)$, where $\vec{\sigma}$ denotes the component-wise application of the activation function. Following the original work, we set the hidden dimension to 64, used a batch size of 128, and trained the model for 20 epochs using the Adam optimizer with default PyTorch parameters.

We trained ACR-GNN on complex formulas $FOC_2$ for labeling. They are presented as a classifier $\alpha_i(x)$ that constructed as:

$$\alpha_0(x) := \text{Blue}(x), \alpha_{i+1}(x) := \exists^{[N,M]} y\, (\alpha_i(y) \wedge \neg E(x, y))$$

Table 2: Dataset statistics summary.

| | | Node | | | Edge | | |
|---|---|---|---|---|---|---|---|
| Classifier | Dataset | Min | Max | Avg | Min | Max | Avg |
| | Train | 40 | 50 | 45 | 560 | 700 | 630 |
| $p_1$ | Test1 | 40 | 50 | 45 | 560 | 700 | 633 |
| | Test2 | 51 | 60 | 55 | 714 | 960 | 832 |
| | Train | 40 | 50 | 45 | 560 | 700 | 630 |
| $p_2$ | Test1 | 40 | 50 | 44 | 560 | 700 | 628 |
| | Test2 | 51 | 60 | 55 | 714 | 960 | 832 |
| | Train | 40 | 50 | 44 | 560 | 700 | 629 |
| $p_2$ | Test1 | 40 | 50 | 45 | 560 | 700 | 630 |
| | Test2 | 51 | 60 | 55 | 714 | 960 | 831 |

where $\exists^{[N,M]}$ stands for "there exist between $N$ and $M$ nodes". satisfying a given property.

Observe that each $\alpha_i(x)$ is in FOC$_2$, as $\exists^{[N,M]}$ can be expressed by combining $\exists^{\geq N}$ and $\neg\exists^{\geq M+1}$.

The data set has the following specifications: Erdös–Rényi graphs and is labeled according to $\alpha_1(x)$, $\alpha_2(x)$, and $\alpha_3(x)$:

- $\alpha_0(x) := \text{Blue}(x)$

- $p_1 : \alpha_1(x) := \exists^{[8,10]} y \left(\alpha_0(y) \wedge \neg E(x,y)\right)$

- $p_2 : \alpha_2(x) := \exists^{[10,30]} y \left(\alpha_1(y) \wedge \neg E(x,y)\right)$

- $p_3 : \alpha_3(x) := \exists^{[10,30]} y \left(\alpha_2(y) \wedge \neg E(x,y)\right)$

In the original work, the authors made experiments with the two activation functions: ReLU and trReLU (truncated ReLU). The truncated ReLU, also referred to as ReLU1, clips activations to the interval $[0, 1]$. It is equivalent to the HardTanh function restricted to this range:

$$\text{trReLU}(x) = \begin{cases} 0, & \text{if } x < 0, \\ x, & \text{if } 0 \leq x \leq 1, \\ 1, & \text{if } x > 1. \end{cases}$$

In our experiments, we employ the strategy described in Barceló et al. (2020), where accuracy is calculated as the total number of correctly classified nodes across all nodes in all graphs in the dataset. In these experiments, we used several types of activation functions: Piecewise linear (ReLU, ReLU6, and trReLU), Smooth unbounded (GELU and SiLU), Smooth bounded (Sigmoid), and Smooth ReLU-like (Softplus and ELU). The activation functions here influence node-level message aggregation, feature combination, and global graph-level representation (readout). Here we present the description for each of the activation functions. We present eight non-linear activation functions in Figure 4 considered in our experiments (we used the implementation of PyTorch).

We presented the key aspects of each activation function (A/F) in Table 3.

Table 3: Comparison of activation functions used in ACR-GNN experiments.

| A/F | Range | Smoothness | Key Properties / Notes |
|---|---|---|---|
| ReLU | $[0, \infty)$ | Non-smooth (kink at 0) | Simple, sparse activations; unbounded above. |
| ReLU6 | $[0, 6]$ | Non-smooth (kinks at 0 and 6) | Bounded version of ReLU; robust under quantization. |
| trReLU | $[0, 1]$ | Non-smooth (piecewise linear) | Clipped ReLU; equivalent to HardTanh restricted to $[0, 1]$. |
| GELU | $(-\infty, \infty)$ | Smooth | Probabilistic ReLU; smoother transitions. |
| Sigmoid | $(0, 1)$ | Smooth | Squashing nonlinearity; prone to vanishing gradients. |
| SiLU (Swish) | $(-\infty, \infty)$ | Smooth | Combines ReLU and Sigmoid; unbounded; performs well in deep models. |
| Softplus | $(0, \infty)$ | Smooth | Smooth approximation of ReLU; strictly positive outputs. |
| ELU | $(-\alpha, \infty)$ | Smooth (except at 0) | Allows negative values; improves gradient flow compared to ReLU. |

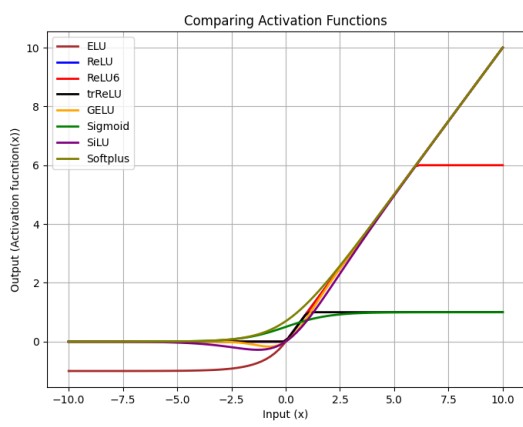

Figure 4: Non-linear activation functions that influence were analyzed.

We trained the models on the dataset and collected the training time. This data is the first preliminary step to analyze the influence of the activation function. Based on the data obtained, we can identify the slowest and fastest activation functions. Table 4 presents the training times of ACR-GNN model across

Table 4: Training time (s) per classifier and activation function

|       | ReLU    | ReLU6   | trReLU  | GELU    | Sigmoid | SiLU    | Softplus | ELU     |
|-------|---------|---------|---------|---------|---------|---------|----------|---------|
| $p_1$ | 1315.00 | 1349.94 | 1422.26 | 1386.15 | 3382.77 | 2886.43 | 3867.97  | 2667.60 |
| $p_2$ | 1665.08 | 1374.38 | 1450.75 | 1386.31 | 3078.93 | 2681.81 | 5737.33  | 2852.38 |
| $p_3$ | 1472.99 | 1528.14 | 1443.93 | 1563.24 | 2842.25 | 2632.73 | 3806.12  | 3007.42 |

datasets for classifiers $p_i$ and different activation functions. The results reveal substantial variability depending on the activation function. Standard piecewise-linear activations, such as ReLU, ReLU6, and trReLU, consistently achieve the shortest training times, with values between 1315–1665,s. In contrast, smoother nonlinearities such as SiLU, Softplus, and Sigmoid incur significantly longer training times, often exceeding 2500s and reaching as high as 5737s for Softplus on $p_2$. GELU and ELU fall between these extremes, with moderate training costs (around 1380–3000s). Across datasets, $p_2$ is generally the most computationally demanding, while $p_1$ remains the least expensive for most activations. Overall, the results indicate that the computational efficiency of training is strongly activation-dependent, with simpler functions such as ReLU and ReLU6 offering the best efficiency, while smoother activations introduce significant overhead.

Table 5: Slowest and fastest activation functions across layers and classifiers.

|       | $p_1$   |          | $p_2$   |          | $p_3$   |          |
|-------|---------|----------|---------|----------|---------|----------|
| Layer | Fastest | Slowest  | Fastest | Slowes   | Fastest | Slowes   |
| 1     | GELU    | Softplus | ReLU    | Softplus | ReLU6   | trReLU   |
| 2     | ReLU6   | Softplus | ReLU    | Softplus | ReLU6   | Sigmoid  |
| 3     | ReLU6   | Sigmoid  | trReLU  | Softplus | ReLU6   | Softplus |
| 4     | trReLU  | Softplus | trReLU  | Softplus | ReLU6   | Softplus |
| 5     | ReLU6   | Softplus | trReLU  | Softplus | ReLU    | Softplus |
| 6     | GELU    | Softplus | ReLU6   | Softplus | trReLU  | Softplus |
| 7     | ReLU    | SiLU     | GELU    | Softplus | ReLU    | Softplus |
| 8     | ReLU    | Softplus | GELU    | Softplus | trReLU  | Softplus |
| 9     | ReLU    | Softplus | ReLU6   | Softplus | ReLU6   | Softplus |
| 10    | ReLU    | Softplus | ReLU6   | Softplus | ReLU6   | ELU      |

Based on the results in Table 4 and Table 5, we identify the fastest and slowest activation functions across layers for the three classifiers. Several consistent trends can be observed along four dimensions: fastest activation, slowest activation, cross-classifier comparison, and depth effect.

*Fastest activations.* Across classifiers and depths, the fastest activations are more diverse, with ReLU, ReLU6, GELU, and trReLU each appearing in multiple layers.

*Slowest activations.* In contrast, Softplus consistently emerges as the slowest activation across nearly all classifiers and depths. Occasional exceptions include ELU in the deepest layer of $p_3$ and SiLU in $p_1$ (layer 7). This trend highlights the relatively high computational cost of smooth unbounded activations compared to piecewise-linear ones.

*Cross-classifier comparison.* While the fastest activations vary considerably by classifier and depth, the slowest activations remain remarkably stable: Softplus dominates across all three classifiers. This suggests that runtime inefficiency of smooth functions is robust to task differences, whereas the speed of simpler functions like ReLU is more context-dependent.

*Depth effect.* As depth increases, variability in the fastest activations decreases, with ReLU becoming dominant in deeper layers across classifiers. The slowest activation, however, remains almost exclusively Softplus, independent of depth.

In summary, Softplus demonstrates the highest runtime cost across classifiers and depths, whereas piecewise-linear activations such as ReLU and ReLU6 offer consistently faster training and inference performance.

We measured the size of the model (in Table 6) and obtained the results that the choice of activation function does not influence on the size of the model.

Table 6: Model size in MB as a function of the number of layers.

| Layers | 1 | 2 | 3 | 4 | 5 | 6 | 7 | 8 | 9 | 10 |
|---|---|---|---|---|---|---|---|---|---|---|
| Size (MB) | 0.06 | 0.11 | 0.16 | 0.22 | 0.27 | 0.32 | 0.38 | 0.43 | 0.49 | 0.54 |

We list the statistics at the microlevel (mean between all nodes) of accuracy. For better representation of the statistics, we present the information in a tabular way. For each layer of the ACR-GNN, we present the accuracy of three formulas $FOC_2$ for the Train, Test1 (the same number of nodes as the Train) and Test2 (larger number of nodes and edges than the Train) as specified in Table 2 and accordingly, the activation function (A/F) that was used for the calculations experiments.

As was mentioned before, we trained ten models for each activation function. For better visualization, we present the benchmark accuracy using heatmaps, which reveal several trends. Specifically, the ACR-GNN generally loses accuracy on Test2 (where the number of edges was increased) and shows a further decline in accuracy as the complexity of the formula increases.

The heatmaps in Figure 5 visualize how the accuracy of the ACR-GNN varies with respect to the number of layers and the choice of activation function. The figure is organized in a $3 \times 3$ grid: rows represent different evaluation metrics (Train accuracy, Test 1 accuracy, and Test 2 accuracy), and columns represent the three datasets classifiers ($p_1$, $p_2$, $p_3$). Each cell encodes accuracy values as a function of the number of layers (y-axis) and activation functions (x-axis). This visualization allows for a direct comparison of performance trends, highlighting, for example, activation functions that maintain stable accuracy across increasing depth or those that degrade sharply.

Generally, the trend that is common for all models: the number of edges, has a significant influence on the final classification results, which can be an indication of how robust the model is. Generally, a robust model performs reliably across different datasets, test conditions, or noise levels (not just on the data it was trained on). A non-robust model may work very well in the training set, but its accuracy drops sharply when evaluated on slightly different or more challenging test sets. In the case of this analysis, robustness is the ability of the model to use different activation functions to keep accuracy stable on Test2 compared to training.

The analysis of accuracy across different network depths and activation functions highlights several clear patterns. In shallow architectures (one and two layers), all activation functions achieve strong performance, with GELU and SiLU in particular showing near-perfect training accuracy and superior

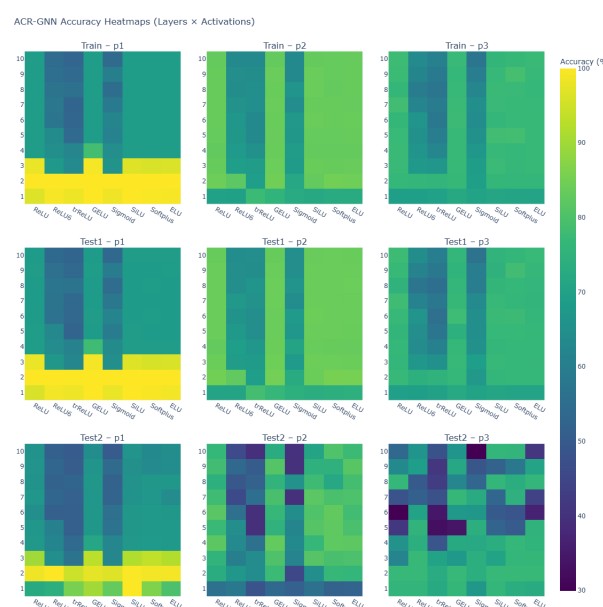

Figure 5: Heatmaps of ACR-GNN accuracy across activation functions and network depth. Each row corresponds to a metric (Train, Test1, Test2), while each column corresponds to a dataset classifier ($p_1$, $p_2$, $p_3$). Color intensity indicates classification accuracy.

generalization across all partitions. Even saturating functions such as Sigmoid and Softplus remain competitive at this stage, which indicates that at low depth the model capacity is sufficient to accommodate a wide range of nonlinearities without severe degradation.

As the number of layers increases (three to five), differences among activation functions become more pronounced. GELU, SiLU, and Softplus consistently emerge as the most robust choices, maintaining higher accuracy on the more challenging test sets. In contrast, ReLU and ELU provide stable but less competitive generalization, while ReLU6, trReLU, and Sigmoid begin to exhibit performance collapse, especially in the $p_2$ partition. At this stage, overfitting effects are also more evident, with models achieving very high training accuracy but diverging in their ability to generalize depending on the activation function.

In deeper architectures (six to eight layers), the gap between smooth activations and saturating or clipped activations widens significantly. ReLU6 and trReLU often collapse below 55% accuracy on the more difficult partitions, whereas GELU and SiLU sustain substantially higher values, often in the 70–80% range. Softplus demonstrates remarkable stability in this regime, occasionally outperforming GELU and SiLU, particularly on $p_2$. These results emphasize the role of smooth, non-saturating nonlinearities in maintaining expressive power and preventing gradient-related issues as depth increases.

Finally, in very deep architectures (nine and ten layers), the failure of saturating activations becomes evident, with Sigmoid and trReLU collapsing to near-random performance on certain partitions. In contrast, Softplus, SiLU, and GELU remain the only viable options, with Softplus providing the strongest generalization on $p_2$ and SiLU maintaining robustness on $p_3$. ReLU and ELU offer moderate results but are consistently outperformed by smoother activations.

Overall, these findings confirm that activation choice becomes increasingly critical with depth, and that smooth functions such as Softplus, SiLU, and GELU provide clear advantages in terms of stability and accuracy across different evaluation settings.

To assess how well the models generalize beyond the training data, we report two complementary metrics: Generalization Ratio and Generalization Gap. The generalization ratio measures the relative

closeness between training and test performance:

$$\text{Generalization Ratio} = \frac{\text{Test Accuracy}}{\text{Train Accuracy}}$$

If the ratio is close to 1, the model generalizes well (Test $\approx$ Train). If it is much less than 1, the model is overfitting (Train $\gg$ Test).

The generalization gap quantifies the absolute drop in performance from training to test:

$$\Delta_{gen} = \text{Train Accuracy} - \text{Test Accuracy}.$$

A small gap reflects strong generalization, while larger gaps highlight overfitting.

In our case, we compute both values separately for Test1 and Test2. After analysis, we obtain the following results. Shallow networks (1–2 layers): SiLU, Softplus, and ELU dominate. Moderate depths (3–5 layers): Softplus (3–4 layers) and ELU (5 layers) are the most reliable. Intermediate-deep networks (6–8 layers): Smooth activations (Softplus, SiLU, GELU, ELU) outperform sharp ones (ReLU, ReLU6, trReLU). Deep networks (9–10 layers): ELU peaks at 9 layers, but at 10 layers, Softplus and SiLU emerge as the only consistently stable choices, while Sigmoid and ELU collapse.

To assess the computational efficiency of dPTQ, we measured the inference time of each model across different activation functions and dataset classifiers.

Table 7: Running time (s) per classifier and activation function. Total running time for the model before (o) and after dPTQ (q) in seconds across the layers.

|       |   | ReLU  | ReLU6 | trReLU | GELU  | Sigmoid | SiLU  | Softplus | ELU   |
|-------|---|-------|-------|--------|-------|---------|-------|----------|-------|
| $p_1$ | o | 20.00 | 53.40 | 21.60  | 24.60 | 28.70   | 29.30 | 45.60    | 25.90 |
|       | q | 22.50 | 69.30 | 29.10  | 28.70 | 35.30   | 29.60 | 47.60    | 30.70 |
| $p_2$ | o | 20.70 | 89.00 | 23.10  | 26.50 | 26.60   | 30.70 | 49.10    | 27.80 |
|       | q | 22.40 | 92.50 | 27.70  | 31.50 | 31.20   | 32.60 | 53.90    | 32.60 |
| $p_3$ | o | 21.30 | 57.00 | 23.80  | 22.70 | 23.40   | 28.10 | 44.80    | 26.60 |
|       | q | 23.60 | 51.10 | 29.80  | 28.40 | 32.20   | 30.80 | 48.70    | 30.40 |

Table 7 reports the inference times of the ACR-GNN models across datasets for the classifiers $p_i$ before (o) and after applying dynamic Post-Training Quantization (q). The results show that quantization does not uniformly reduce runtime; in fact, for several activation functions (e.g., ReLU6, trReLU, GELU, and Sigmoid), the quantized models exhibit increased execution time compared to their original counterparts across most datasets. For example, ReLU6 increases from 53.4s to 69.3s on $p_1$, and from 89.0s to 92.5s on $p_2$. A similar pattern is observed for GELU and Sigmoid, where quantization consistently adds between 4–6 overhead. However, some cases highlight improved efficiency after quantization: most notably, ReLU on $p_3$ (from 57.0s down to 51.1s) and, to a smaller extent, SiLU and Softplus exhibit negligible changes across datasets. These results suggest that the computational impact of dPTQ is activation-dependent and dataset-dependent, with smoother nonlinearities (e.g., SiLU, Softplus) showing greater stability. In contrast, activations with more complex or nonlinear behavior (e.g., GELU, ReLU6, Sigmoid) tend to incur additional overhead after quantization.

We report the mean dynamic PTQ speedup across 10 layers, defined as the ratio of non-quantized to quantized execution time (original time / dPTQ time), which indicates whether dynamic PTQ reduces or increases runtime.

Figure 6 shows the mean speed-up of dynamic PTQ relative to non-quantized execution, grouped by activation function and datasets for classifiers $p_i$. The results indicate that dynamic PTQ does not universally accelerate inference: in most cases, execution times slightly increase (speed-up $< 1$). However, for $p_3$ with ReLU6, dynamic PTQ achieves a 23% speed-up, highlighting that the benefits depend on both the activation function and the characteristics of the dataset. Sigmoid consistently underperforms under PTQ, suggesting limited suitability for quantized execution.

Table 8 reports the size of the model before and after applying dynamic Post-Training Quantization. We used two metrics to measure the values: Reduction and $\Delta_{Size}$. We calculate each of these two

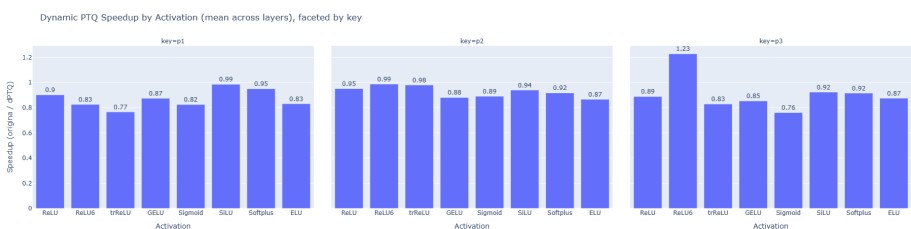

Figure 6: Mean dPTQ speedup (non-dPTQ time / dPTQ time) by activation function and classifier.

metrics in the following way:

$$\text{Reduction} = \frac{\text{Value}_{dPTQ} - \text{Value}_{original}}{\text{Value}_{original}} \times 100\%$$

and

$$\Delta_{Size} = \text{Value}_{original} - \text{Value}_{dPTQ}$$

respectfully.

The results (Table8) show a consistent and substantial reduction across all configurations, with quantized models achieving reductions between approximately 59.6% and 62.3% compared to their original size. This indicates that dynamic PTQ provides highly effective compression with nearly constant proportional savings, independent of the initial model size. Such reductions highlight the suitability of dynamic PTQ for resource-constrained environments, where storage and memory efficiency are critical.

Table 8: Model sizes before and after dynamic Post-Training Quantization.

| Layer | Original Size (MB) | Quantized Size (MB) | $\Delta_{Size}$ (MB) | Reduction (%) |
|-------|--------------------|--------------------|--------------------|--------------------|
| 1 | 0.057 | 0.023 | 0.034 | -59.604 |
| 2 | 0.112 | 0.044 | 0.068 | -60.993 |
| 3 | 0.167 | 0.064 | 0.103 | -61.559 |
| 4 | 0.221 | 0.085 | 0.137 | -61.804 |
| 5 | 0.276 | 0.105 | 0.171 | -61.975 |
| 6 | 0.331 | 0.126 | 0.206 | -62.068 |
| 7 | 0.386 | 0.146 | 0.240 | -62.148 |
| 8 | 0.441 | 0.167 | 0.274 | -62.194 |
| 9 | 0.496 | 0.187 | 0.309 | -62.230 |
| 10 | 0.551 | 0.208 | 0.343 | -62.251 |

Figure 7 illustrates the layer-wise accuracy of ACR-GNN across datasets for classifiers $p_i$ and activation functions, before and after dynamic Post-Training Quantization. The heatmaps show consistent patterns across Train and Test1, where most activations achieve high accuracies in the lower layers (1-3) but gradually decrease as the number of layers increases. This trend is most pronounced for ReLU, ReLU6, and trReLU, which exhibit sharp drops in accuracy, particularly in $p_1$. In contrast, smoother activations such as GELU, SiLU, Softplus, and ELU demonstrate more stable performance across depth, indicating better resilience to increased model complexity.

For Test2, the heatmaps reveal greater variation between activations and datasets. Piecewise-linear activations (ReLU, ReLU6, trReLU) consistently show lower and less stable accuracies, while GELU, SiLU, and Softplus retain higher performance across most layers. The visual differences are especially apparent in $p_2$ and $p_3$, where ReLU-based functions degrade more quickly compared to smoother nonlinearities.

For a more detailed analysis, we constructed tables with specific structural requirements to better examine the influence of dynamic PTQ. The impact of dynamic PTQ was assessed by calculating the Generalization Ratio (GR), the Generalization Gap ($\Delta_{gen}$), and the accuracy difference between the original and quantized models ($\Delta_{acc}$). The results are presented in Tables 9 –Tables 18. Below, we summarize the principal observations layer by layer.

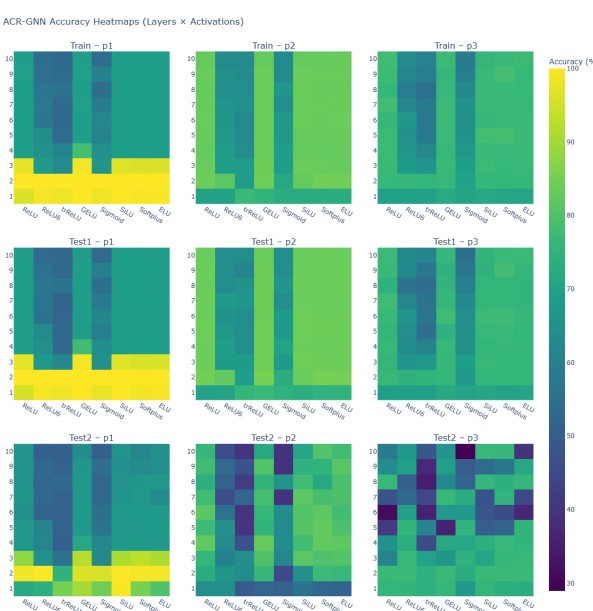

Figure 7: Heatmaps of ACR-GNN accuracy after applying the dynamic PTQ across activation functions and network depth. Each row corresponds to a metric (Train, Test1, Test2), while each column corresponds to a dataset classifier ($p_1$, $p_2$, $p_3$). Color intensity indicates classification accuracy.

Table 9: Accuracy differences ($\Delta_{acc}$, %) and generalization metrics (GR, $\Delta_{gen}$) per activation and classifier for one-layer ACR-GNN after applying the dynamic PTQ.

| A/F | $p_1$ | | | | | | $p_2$ | | | | | | $p_3$ | | | | | |
| | Test1 | | | Test2 | | | Test1 | | | Test2 | | | Test1 | | | Test2 | | |
| | GR | $\Delta_{gen}$ | $\Delta_{acc}$ | GR | $\Delta_{gen}$ | $\Delta_{acc}$ | GR | $\Delta_{gen}$ | $\Delta_{acc}$ | GR | $\Delta_{gen}$ | $\Delta_{acc}$ | GR | $\Delta_{gen}$ | $\Delta_{acc}$ | GR | $\Delta_{gen}$ | $\Delta_{acc}$ |
|---|---|---|---|---|---|---|---|---|---|---|---|---|---|---|---|---|---|---|
| ReLU | 0.994 | +0.561 | +0.411% | 0.743 | +24.702 | +0.915% | 1.003 | -0.176 | +1.270% | 0.920 | +5.559 | -7.760% | 0.996 | +0.265 | +0.328% | 1.095 | -6.536 | +0.134% |
| ReLU6 | 1.001 | -0.082 | -0.150% | 0.744 | +25.379 | +0.267% | 1.013 | -0.919 | +0.517% | 0.932 | +4.710 | -6.676% | 0.999 | +0.043 | +0.080% | 1.093 | -6.396 | +0.220% |
| trReLU | 1.001 | -0.145 | -0.146% | 0.866 | +13.204 | +0.494% | 0.988 | +0.900 | +2.768% | 0.640 | +27.678 | -1.196% | 0.994 | +0.420 | +0.968% | 1.067 | -4.673 | -1.689% |
| GELU | 1.000 | -0.014 | -0.004% | 0.901 | +9.936 | -0.050% | 1.018 | -1.365 | -0.134% | 0.695 | +23.025 | +0.958% | 0.997 | +0.186 | -0.062% | 1.058 | -4.159 | +0.018% |
| Sigmoid | 0.994 | +0.587 | +0.186% | 0.858 | +13.967 | -1.376% | 1.025 | -1.788 | -0.526% | 0.698 | +21.999 | +0.749% | 0.997 | +0.197 | +0.262% | 0.952 | +3.368 | +0.090% |
| SiLU | 1.000 | +0.000 | +0.000% | 0.995 | +0.472 | +0.342% | 1.021 | -1.489 | -0.076% | 0.777 | +16.083 | +0.000% | 0.997 | +0.200 | +0.209% | 1.066 | -4.666 | +0.336% |
| Softplus | 0.993 | +0.673 | +0.880% | 0.857 | +14.224 | +1.279% | 1.023 | -1.685 | -0.588% | 0.714 | +20.832 | +0.097% | 0.995 | +0.372 | +0.124% | 1.038 | -2.688 | +0.018% |
| ELU | 0.998 | +0.243 | -0.402% | 0.805 | +19.405 | -0.458% | 1.025 | -1.802 | -0.334% | 0.714 | +20.774 | +0.349% | 1.000 | -0.033 | +0.004% | 1.036 | -2.543 | -0.401% |

For layer 1 (Table 9), across Test1, most activations show GR $\approx 1$ and small $|\Delta_{gen}|$. On Test2, several activations exhibit lower GR and larger positive $\Delta_{gen}$, indicating stronger train–eval gaps (e.g., ReLU and trReLU on $p_2$ and $p_3$). $\Delta_{acc}$ is generally small (mostly within $\pm 1\%$), with Softplus and ReLU showing a few larger but still moderate deviations depending on classifier and data split.

Table 10: Accuracy differences ($\Delta_{acc}$, %) and generalization metrics (GR, $\Delta_{gen}$) per activation and classifier for two-layer ACR-GNN after applying the dynamic PTQ.

| A/F | $p_1$ | | | | | | $p_2$ | | | | | | $p_3$ | | | | | |
| | Test1 | | | Test2 | | | Test1 | | | Test2 | | | Test1 | | | Test2 | | |
| | GR | $\Delta_{gen}$ | $\Delta_{acc}$ | GR | $\Delta_{gen}$ | $\Delta_{acc}$ | GR | $\Delta_{gen}$ | $\Delta_{acc}$ | GR | $\Delta_{gen}$ | $\Delta_{acc}$ | GR | $\Delta_{gen}$ | $\Delta_{acc}$ | GR | $\Delta_{gen}$ | $\Delta_{acc}$ |
|---|---|---|---|---|---|---|---|---|---|---|---|---|---|---|---|---|---|---|
| ReLU | 1.000 | -0.006 | -0.004% | 0.986 | +1.445 | -0.025% | 1.008 | -0.678 | +0.013% | 0.866 | +11.265 | +0.389% | 1.001 | -0.074 | +0.342% | 1.006 | -0.438 | +0.025% |
| ReLU6 | 1.000 | +0.001 | +0.000% | 0.999 | +0.086 | +0.058% | 1.008 | -0.677 | +0.254% | 0.815 | +15.121 | -0.047% | 0.973 | +2.038 | -0.102% | 1.006 | -0.494 | +0.025% |
| trReLU | 1.000 | +0.018 | +0.027% | 0.746 | +25.384 | +20.200% | 1.001 | -0.085 | +0.615% | 0.932 | +4.648 | +0.313% | 0.996 | -0.316 | -0.093% | 0.981 | +1.464 | -1.187% |
| GELU | 1.000 | +0.000 | +0.000% | 0.962 | +3.815 | +1.286% | 1.009 | -0.759 | +0.045% | 0.906 | +7.885 | +1.160% | 0.996 | +0.288 | +0.431% | 1.006 | -0.426 | -0.011% |
| Sigmoid | 1.000 | +0.013 | +0.009% | 0.959 | +4.087 | +0.789% | 1.023 | -1.610 | -0.312% | 0.627 | +26.651 | +0.342% | 0.965 | +2.431 | +0.391% | 1.058 | -4.029 | +0.563% |
| SiLU | 1.000 | +0.000 | +0.000% | 0.996 | +0.375 | +0.086% | 1.009 | -0.758 | +0.290% | 0.777 | +18.504 | +0.425% | 0.994 | +0.427 | +0.328% | 1.009 | -0.665 | +0.011% |
| Softplus | 1.000 | -0.000 | +0.000% | 0.991 | +0.889 | -0.159% | 1.005 | -0.395 | +0.406% | 0.911 | +7.556 | -0.313% | 1.002 | -0.147 | +0.355% | 1.011 | -0.850 | +0.000% |
| ELU | 1.000 | +0.004 | +0.009% | 0.989 | +1.102 | +0.101% | 1.006 | -0.537 | +0.250% | 0.907 | +7.805 | -0.378% | 0.998 | +0.155 | +0.417% | 1.009 | -0.692 | +0.029% |

For layer 2 (Table 10) Test1 remains stable (GR $\approx 1$) for most activations and classifiers; Test2 shows reduced GR and positive $\Delta_{gen}$ (notably for ReLU, Sigmoid, GELU on $p_2$). Most $\Delta_{acc}$ values are small; isolated spikes occur (e.g., GELU on $p_2$–Test2) but remain at the level of % units rather than tens of percent.

Table 11: Accuracy differences ($\Delta_{acc}$, %) and generalization metrics (GR, $\Delta_{gen}$) per activation and classifier for three-layer ACR-GNN after applying the dynamic PTQ.

| A/F | $p_1$ | | | | | | $p_2$ | | | | | | $p_3$ | | | | | |
|---|---|---|---|---|---|---|---|---|---|---|---|---|---|---|---|---|---|---|
| | Test1 | | | Test2 | | | Test1 | | | Test2 | | | Test1 | | | Test2 | | |
| | GR | $\Delta_{gen}$ | $\Delta_{acc}$ | GR | $\Delta_{gen}$ | $\Delta_{acc}$ | GR | $\Delta_{gen}$ | $\Delta_{acc}$ | GR | $\Delta_{gen}$ | $\Delta_{acc}$ | GR | $\Delta_{gen}$ | $\Delta_{acc}$ | GR | $\Delta_{gen}$ | $\Delta_{acc}$ |
| ReLU | 1.000 | -0.010 | +0.464% | 0.902 | +9.520 | +2.057% | 1.008 | -0.674 | -0.062% | 0.912 | +7.297 | -0.468% | 0.988 | +0.935 | +0.204% | 0.797 | +15.671 | +0.188% |
| ReLU6 | 0.992 | +0.545 | +0.022% | 0.961 | +2.602 | -0.187% | 0.987 | +0.876 | +0.290% | 0.878 | +8.416 | -0.342% | 0.976 | +1.588 | +0.764% | 1.133 | -8.996 | +0.534% |
| trReLU | 0.993 | +0.451 | +0.190% | 0.897 | +6.459 | +0.267% | 0.998 | +0.113 | +0.125% | 0.940 | +4.093 | -0.267% | 1.016 | -1.051 | -1.749% | 1.058 | -3.828 | +0.798% |
| GELU | 1.001 | -0.123 | +0.102% | 0.934 | +6.504 | +0.584% | 1.004 | -0.359 | +0.080% | 0.952 | +3.999 | +0.227% | 0.994 | +0.473 | +0.448% | 0.865 | +10.352 | +0.729% |
| Sigmoid | 0.986 | +0.921 | +0.115% | 0.947 | +3.452 | +0.104% | 0.994 | +0.437 | -0.281% | 0.908 | +6.288 | -0.184% | 0.996 | +0.257 | +0.342% | 1.077 | -5.281 | -0.238% |
| SiLU | 1.002 | -0.153 | -0.172% | 0.942 | +5.563 | +0.677% | 1.008 | -0.658 | -0.004% | 0.975 | +2.053 | -0.274% | 0.993 | +0.568 | +0.271% | 0.913 | +6.688 | -0.141% |
| Softplus | 0.999 | +0.094 | -0.075% | 0.969 | +2.946 | +0.512% | 1.008 | -0.677 | -0.085% | 0.955 | +3.747 | -0.259% | 0.997 | +0.213 | +0.253% | 0.996 | +0.333 | -0.022% |
| ELU | 0.998 | +0.160 | -0.071% | 0.946 | +5.162 | +0.760% | 1.008 | -0.667 | +0.018% | 0.966 | +2.820 | -0.659% | 0.993 | +0.568 | +0.413% | 0.996 | +0.295 | -0.152% |

For layer 3 (Table 11), we observed that some separation emerges: GELU and trReLU retain comparatively higher GR (especially on $p_1$ for Test1), while Test2 GR drops for multiple activations on $p_2$ and $p_3$ with increased $\Delta_{gen}$. Most $\Delta_{acc}$ remain modest; a few cells show larger swings (e.g., ReLU on $p_1$ for Test2 and some $p_3$ for Test2 cases), which suggests sensitivity to the deeper setting.

Table 12: Accuracy differences ($\Delta_{acc}$, %) and generalization metrics (GR, $\Delta_{gen}$) per activation and classifier for four-layer ACR-GNN after applying the dynamic PTQ.

| A/F | $p_1$ | | | | | | $p_2$ | | | | | | $p_3$ | | | | | |
|---|---|---|---|---|---|---|---|---|---|---|---|---|---|---|---|---|---|---|
| | Test1 | | | Test2 | | | Test1 | | | Test2 | | | Test1 | | | Test2 | | |
| | GR | $\Delta_{gen}$ | $\Delta_{acc}$ | GR | $\Delta_{gen}$ | $\Delta_{acc}$ | GR | $\Delta_{gen}$ | $\Delta_{acc}$ | GR | $\Delta_{gen}$ | $\Delta_{acc}$ | GR | $\Delta_{gen}$ | $\Delta_{acc}$ | GR | $\Delta_{gen}$ | $\Delta_{acc}$ |
| ReLU | 0.997 | +0.193 | +0.018% | 0.974 | +1.767 | +0.195% | 1.011 | -0.907 | -0.593% | 0.995 | +0.395 | -0.468% | 0.995 | +0.385 | -0.169% | 0.846 | +11.916 | +0.318% |
| ReLU6 | 0.987 | +0.846 | -0.049% | 0.959 | +2.634 | -0.231% | 1.016 | -1.078 | -0.036% | 0.928 | +4.884 | +0.411% | 0.956 | +2.907 | +0.146% | 1.105 | -7.010 | +0.397% |
| trReLU | 0.854 | +8.958 | +9.068% | 0.826 | +10.641 | +3.152% | 0.998 | +0.127 | +2.634% | 0.734 | +17.053 | +8.920% | 0.977 | +1.453 | -0.262% | 0.728 | +16.908 | +3.270% |
| GELU | 0.992 | +0.607 | +0.203% | 0.906 | +7.389 | +0.115% | 1.010 | -0.806 | -0.196% | 0.997 | +0.213 | -0.274% | 0.995 | +0.367 | -0.027% | 0.935 | +5.005 | +0.332% |
| Sigmoid | 0.984 | +1.061 | +0.208% | 0.963 | +2.390 | +0.061% | 1.023 | -1.552 | -0.651% | 0.926 | +5.049 | -0.274% | 0.975 | +1.676 | +0.182% | 1.069 | -4.585 | +0.718% |
| SiLU | 0.996 | +0.267 | -0.049% | 0.979 | +1.462 | +0.083% | 1.011 | -0.897 | -0.397% | 0.939 | +5.109 | -1.272% | 0.988 | +0.886 | +0.075% | 0.960 | +3.085 | -0.433% |
| Softplus | 0.999 | +0.086 | -0.221% | 0.987 | +0.883 | +0.025% | 1.008 | -0.694 | -0.147% | 0.995 | +0.444 | -0.141% | 0.998 | +0.126 | +0.200% | 1.001 | -0.047 | +0.043% |
| ELU | 0.997 | +0.236 | -0.053% | 0.978 | +1.529 | +0.184% | 1.009 | -0.730 | -0.495% | 0.984 | +1.291 | +0.184% | 0.993 | +0.503 | +0.222% | 0.978 | +1.676 | +0.051% |

For layer 4 (Table 12) ReLU6 stands out as robust on $p_1$ (GR $\approx$ 0.96–0.99 with smaller $\Delta_{gen}$). Larger gaps appear on Test2 for several activations and classifiers (e.g., ReLU on $p_3$ for Test2). Accuracy changes are mostly modest, with a few noticeable positive or negative shifts for trReLU and SiLU in specific cells.

Table 13: Accuracy differences ($\Delta_{acc}$, %) and generalization metrics (GR, $\Delta_{gen}$) per activation and classifier for five-layer ACR-GNN after applying the dynamic PTQ.

| A/F | $p_1$ | | | | | | $p_2$ | | | | | | $p_3$ | | | | | |
|---|---|---|---|---|---|---|---|---|---|---|---|---|---|---|---|---|---|---|
| | Test1 | | | Test2 | | | Test1 | | | Test2 | | | Test1 | | | Test2 | | |
| | GR | $\Delta_{gen}$ | $\Delta_{acc}$ | GR | $\Delta_{gen}$ | $\Delta_{acc}$ | GR | $\Delta_{gen}$ | $\Delta_{acc}$ | GR | $\Delta_{gen}$ | $\Delta_{acc}$ | GR | $\Delta_{gen}$ | $\Delta_{acc}$ | GR | $\Delta_{gen}$ | $\Delta_{acc}$ |
| ReLU | 1.001 | -0.055 | -0.102% | 0.972 | +1.914 | +0.036% | 1.009 | -0.761 | -0.325% | 0.929 | +5.950 | -0.076% | 0.983 | +1.280 | +0.195% | 0.533 | +36.010 | +0.372% |
| ReLU6 | 0.991 | +0.588 | -0.491% | 0.956 | +2.776 | -0.425% | 1.000 | +0.013 | +0.539% | 0.961 | +2.590 | +0.104% | 0.982 | +1.192 | +0.315% | 1.155 | -10.104 | +0.198% |
| trReLU | 0.974 | +1.367 | +0.000% | 0.958 | +2.239 | -0.429% | 1.021 | -1.296 | +0.000% | 0.628 | +23.419 | -0.004% | 0.936 | +3.704 | +2.828% | 1.054 | -3.134 | -28.571% |
| GELU | 0.996 | +0.302 | +0.040% | 0.978 | +1.494 | -0.025% | 1.007 | -0.571 | -0.308% | 0.901 | +8.274 | -0.155% | 0.966 | +2.642 | -0.191% | 0.460 | +41.643 | -1.649% |
| Sigmoid | 0.974 | +1.613 | -0.022% | 0.925 | +4.549 | -0.014% | 1.000 | +0.027 | -0.227% | 0.936 | +4.282 | -0.692% | 0.983 | +1.106 | +0.444% | 1.145 | -9.500 | -1.007% |
| SiLU | 0.995 | +0.328 | +0.080% | 0.976 | +1.633 | +0.040% | 1.010 | -0.811 | -0.370% | 0.972 | +2.328 | +0.216% | 0.980 | +1.590 | +0.178% | 0.636 | +28.440 | +2.505% |
| Softplus | 0.997 | +0.196 | +0.084% | 0.980 | +1.336 | +0.141% | 1.008 | -0.684 | -0.125% | 0.993 | +0.598 | -0.068% | 0.978 | +1.736 | +0.013% | 0.656 | +27.048 | -1.447% |
| ELU | 0.999 | +0.059 | -0.022% | 0.985 | +1.003 | -0.022% | 1.007 | -0.590 | -0.134% | 0.955 | +3.728 | +0.032% | 0.992 | +0.652 | +0.186% | 0.980 | +1.560 | -0.235% |

For layer 5 (Table 13) trReLU remains strong on $p_1$ for Test1 (GR $\approx$ 0.97) with small gaps, while some activations show pronounced degradation on $p_3$ for Test2 (very low GR and large $\Delta_{gen}$). Most $\Delta_{acc}$ are small, but a few outliers appear (e.g., large-magnitude entries for trReLU or GELU on $p_3$ for Test2), indicating occasional instability at this depth.

Table 14: Accuracy differences ($\Delta_{acc}$, %) and generalization metrics (GR, $\Delta_{gen}$) per activation and classifier for six layer ACR-GNN after applying the dynamic PTQ.

| A/F | $p_1$ | | | | | | $p_2$ | | | | | | $p_3$ | | | | | |
|---|---|---|---|---|---|---|---|---|---|---|---|---|---|---|---|---|---|---|
| | Test1 | | | Test2 | | | Test1 | | | Test2 | | | Test1 | | | Test2 | | |
| | GR | $\Delta_{gen}$ | $\Delta_{acc}$ | GR | $\Delta_{gen}$ | $\Delta_{acc}$ | GR | $\Delta_{gen}$ | $\Delta_{acc}$ | GR | $\Delta_{gen}$ | $\Delta_{acc}$ | GR | $\Delta_{gen}$ | $\Delta_{acc}$ | GR | $\Delta_{gen}$ | $\Delta_{acc}$ |
| ReLU | 0.997 | +0.180 | +0.088% | 0.941 | +4.048 | +0.115% | 1.007 | -0.598 | -0.192% | 0.977 | +1.908 | +0.065% | 1.006 | -0.468 | +0.186% | 0.406 | +45.319 | -1.007% |
| ReLU6 | 0.981 | +1.105 | +0.150% | 0.895 | +6.013 | -0.375% | 0.982 | +1.225 | +0.428% | 0.841 | +10.566 | -0.234% | 0.994 | +0.410 | +0.129% | 1.048 | -3.175 | +0.935% |
| trReLU | 0.986 | +0.726 | +0.747% | 0.968 | +1.697 | -0.061% | 1.020 | -1.272 | +0.000% | 0.628 | +23.443 | +0.000% | 0.976 | +1.366 | -1.052% | 0.615 | +22.343 | -1.516% |
| GELU | 0.993 | +0.453 | +0.075% | 0.979 | +1.455 | -0.086% | 1.005 | -0.382 | -0.049% | 0.912 | +7.332 | +0.612% | 1.005 | -0.425 | +0.115% | 0.868 | +10.186 | +0.173% |
| Sigmoid | 0.992 | +0.504 | +0.190% | 0.891 | +6.725 | +0.404% | 1.011 | -0.761 | -0.098% | 0.943 | +3.841 | +0.829% | 0.998 | +0.110 | +0.617% | 1.046 | -2.952 | -0.134% |
| SiLU | 0.996 | +0.246 | +0.141% | 0.979 | +1.430 | -0.122% | 1.006 | -0.481 | -0.080% | 0.961 | +3.230 | -0.173% | 1.011 | -0.813 | +0.044% | 0.630 | +28.322 | +1.750% |
| Softplus | 0.994 | +0.424 | +0.429% | 0.983 | +1.143 | +0.144% | 1.005 | -0.451 | +0.045% | 0.983 | +1.408 | +0.944% | 1.008 | -0.584 | +0.040% | 0.588 | +31.682 | +3.187% |
| ELU | 0.994 | +0.434 | +0.256% | 0.974 | +1.810 | +0.104% | 1.005 | -0.416 | -0.080% | 0.934 | +5.528 | +0.331% | 0.992 | +0.612 | -0.013% | 0.470 | +40.811 | +0.256% |

For layer 6 (Table 14), we noticed moderate GR on Test1 for many activations; Test2 again exposes larger train–evaluation gaps for several pairs (e.g., ReLU6, trReLU on $p_2$ and $p_3$). A handful of $\Delta_{acc}$ cells become larger (e.g., Softplus on $p_3$ for Test2), though many remain within a few tenths of a percent.

Table 15: Accuracy differences ($\Delta_{acc}$, %) and generalization metrics (GR, $\Delta_{gen}$) per activation and classifier for seven-layer ACR-GNN after applying the dynamic PTQ.

| A/F | $p_1$ Test1 GR | $\Delta_{gen}$ | $\Delta_{acc}$ | $p_1$ Test2 GR | $\Delta_{gen}$ | $\Delta_{acc}$ | $p_2$ Test1 GR | $\Delta_{gen}$ | $\Delta_{acc}$ | $p_2$ Test2 GR | $\Delta_{gen}$ | $\Delta_{acc}$ | $p_3$ Test1 GR | $\Delta_{gen}$ | $\Delta_{acc}$ | $p_3$ Test2 GR | $\Delta_{gen}$ | $\Delta_{acc}$ |
|---|---|---|---|---|---|---|---|---|---|---|---|---|---|---|---|---|---|---|
| ReLU | 0.995 | +0.330 | +0.212% | 0.968 | +2.176 | +0.404% | 1.006 | -0.525 | +0.045% | 0.934 | +5.460 | +0.241% | 0.997 | +0.204 | +0.311% | 0.605 | +30.815 | +0.751% |
| ReLU6 | 0.985 | +0.888 | -0.367% | 0.913 | +5.058 | +0.285% | 1.026 | -1.713 | -0.379% | 0.699 | +19.814 | -0.068% | 1.005 | -0.319 | -0.839% | 0.857 | +8.831 | -1.361% |
| trReLU | 0.983 | +0.901 | +1.605% | 0.979 | +1.126 | -0.054% | 1.023 | -1.509 | -0.512% | 0.806 | +12.505 | +3.113% | 0.963 | +2.171 | +1.611% | 0.759 | +14.011 | +3.295% |
| GELU | 0.996 | +0.257 | -0.022% | 0.973 | +1.883 | -0.177% | 1.004 | -0.367 | +0.004% | 0.942 | +4.824 | +0.551% | 0.991 | +0.684 | -0.293% | 0.995 | +0.376 | +0.004% |
| Sigmoid | 0.984 | +0.949 | -0.181% | 0.891 | +6.376 | -0.061% | 1.000 | -0.020 | -0.009% | 0.618 | +24.404 | -0.032% | 0.961 | +2.545 | -0.710% | 1.126 | -8.205 | -0.520% |
| SiLU | 0.997 | +0.222 | +0.124% | 0.958 | +2.850 | +0.130% | 1.004 | -0.346 | +0.178% | 0.980 | +1.670 | -0.007% | 0.989 | +0.826 | +0.710% | 0.619 | +29.343 | +5.363% |
| Softplus | 0.997 | +0.175 | +0.027% | 0.976 | +1.677 | -0.159% | 1.004 | -0.321 | +0.183% | 0.945 | +4.546 | +0.742% | 0.990 | +0.801 | +0.071% | 0.977 | +1.745 | -0.736% |
| ELU | 0.998 | +0.142 | +0.040% | 0.926 | +5.039 | +0.288% | 1.004 | -0.338 | +0.245% | 0.956 | +3.670 | +0.317% | 0.980 | +1.519 | +0.586% | 0.580 | +32.243 | -0.783% |

For layer 7 (Table 15) GELU and SiLU retain relatively stable Test1 GR on $p_1$; Test2 often degrades across $p_2$ and $p_3$ with increased $\Delta_{gen}$. Accuracy shifts are still mostly small, but some cells show multi-percent swings for trReLU/SiLU on $p_3$ for Test2, pointing to sensitivity at greater depth.

Table 16: Accuracy differences ($\Delta_{acc}$, %) and generalization metrics (GR, $\Delta_{gen}$) per activation and classifier for eight-layer ACR-GNN after applying the dynamic PTQ.

| A/F | $p_1$ Test1 GR | $\Delta_{gen}$ | $\Delta_{acc}$ | $p_1$ Test2 GR | $\Delta_{gen}$ | $\Delta_{acc}$ | $p_2$ Test1 GR | $\Delta_{gen}$ | $\Delta_{acc}$ | $p_2$ Test2 GR | $\Delta_{gen}$ | $\Delta_{acc}$ | $p_3$ Test1 GR | $\Delta_{gen}$ | $\Delta_{acc}$ | $p_3$ Test2 GR | $\Delta_{gen}$ | $\Delta_{acc}$ |
|---|---|---|---|---|---|---|---|---|---|---|---|---|---|---|---|---|---|---|
| ReLU | 0.998 | +0.109 | +0.018% | 0.957 | +2.952 | +0.162% | 1.004 | -0.328 | +0.281% | 0.882 | +9.788 | +0.566% | 0.978 | +1.716 | +0.613% | 1.005 | -0.352 | +0.025% |
| ReLU6 | 1.012 | -0.656 | -0.323% | 0.917 | +4.687 | +0.079% | 1.005 | -0.326 | -1.841% | 0.799 | +12.838 | -1.841% | 0.956 | +2.601 | +1.278% | 0.862 | +8.147 | -0.917% |
| trReLU | 0.996 | +0.218 | +0.964% | 0.949 | +2.831 | -0.497% | 1.015 | -0.993 | +1.364% | 0.651 | +22.393 | +10.059% | 0.979 | +1.146 | +0.719% | 0.624 | +20.858 | +8.001% |
| GELU | 0.997 | +0.173 | +0.080% | 0.974 | +1.747 | +0.119% | 1.005 | -0.382 | +0.111% | 0.900 | +8.352 | +0.303% | 0.985 | +1.140 | +0.812% | 0.815 | +14.204 | -0.267% |
| Sigmoid | 0.994 | +0.325 | -0.128% | 0.954 | +2.538 | -0.216% | 0.996 | +0.257 | -0.009% | 0.991 | +0.592 | -0.209% | 0.961 | +2.434 | -0.364% | 0.825 | +10.900 | +1.155% |
| SiLU | 0.998 | +0.170 | +0.018% | 0.956 | +3.038 | -0.025% | 1.004 | -0.324 | -0.089% | 0.934 | +5.549 | -0.173% | 0.984 | +1.205 | +0.071% | 0.990 | +0.795 | +0.014% |
| Softplus | 1.000 | +0.026 | -0.035% | 0.970 | +2.074 | +0.335% | 1.006 | -0.525 | +0.160% | 0.863 | +11.365 | -0.040% | 0.989 | +0.837 | +0.426% | 0.957 | +3.301 | -0.217% |
| ELU | 0.995 | +0.355 | +0.159% | 0.980 | +1.385 | -0.137% | 1.009 | -0.733 | +0.009% | 0.952 | +4.020 | -0.569% | 0.987 | +1.030 | +0.417% | 0.998 | +0.122 | -0.105% |

For layer 8 (Table 16) ReLU and ReLU6 keep Test1 near GR $\approx 1$ on $p_1$; several activations show reduced GR and larger $\Delta_{gen}$ on $p_2$ and on $p_3$ for Test2. A few $\Delta_{acc}$ spikes appear (e.g., trReLU on $p_2$ and $p_3$), but most entries remain within modest ranges.

Table 17: Accuracy differences ($\Delta_{acc}$, %) and generalization metrics (GR, $\Delta_{gen}$) per activation and classifier for nine-layer ACR-GNN after applying the dynamic PTQ.

| A/F | $p_1$ Test1 GR | $\Delta_{gen}$ | $\Delta_{acc}$ | $p_1$ Test2 GR | $\Delta_{gen}$ | $\Delta_{acc}$ | $p_2$ Test1 GR | $\Delta_{gen}$ | $\Delta_{acc}$ | $p_2$ Test2 GR | $\Delta_{gen}$ | $\Delta_{acc}$ | $p_3$ Test1 GR | $\Delta_{gen}$ | $\Delta_{acc}$ | $p_3$ Test2 GR | $\Delta_{gen}$ | $\Delta_{acc}$ |
|---|---|---|---|---|---|---|---|---|---|---|---|---|---|---|---|---|---|---|
| ReLU | 0.998 | +0.165 | -0.186% | 0.961 | +2.707 | +0.058% | 1.006 | -0.471 | -0.085% | 0.934 | +5.539 | +0.501% | 0.993 | +0.560 | +0.160% | 0.818 | +14.116 | -0.173% |
| ReLU6 | 1.016 | -0.866 | -0.168% | 0.963 | +2.014 | +0.097% | 0.999 | +0.096 | +0.134% | 0.827 | +11.147 | +0.184% | 1.014 | -0.833 | -1.212% | 1.120 | -7.431 | -0.307% |
| trReLU | 0.988 | +0.638 | +0.765% | 0.956 | +2.339 | -0.512% | 0.969 | +1.964 | +2.763% | 0.748 | +15.945 | +2.897% | 0.958 | +2.485 | +1.252% | 0.624 | +22.306 | +4.118% |
| GELU | 0.997 | +0.187 | +0.000% | 0.967 | +2.270 | +0.076% | 1.002 | -0.151 | +0.018% | 0.960 | +3.346 | +0.508% | 0.980 | +1.504 | +0.439% | 0.898 | +7.770 | +3.006% |
| Sigmoid | 0.984 | +0.989 | +0.062% | 0.904 | +5.792 | -0.209% | 1.019 | -1.176 | +0.209% | 0.623 | +23.768 | +0.393% | 0.965 | +2.112 | +2.046% | 0.835 | +9.827 | -1.176% |
| SiLU | 0.997 | +0.200 | -0.009% | 0.942 | +3.992 | +0.292% | 1.003 | -0.274 | +0.022% | 0.885 | +9.616 | +0.515% | 0.969 | +2.408 | -0.133% | 0.691 | +23.756 | -1.014% |
| Softplus | 0.998 | +0.137 | -0.075% | 0.917 | +5.707 | +0.317% | 1.004 | -0.331 | +0.160% | 0.901 | +8.275 | -0.522% | 0.992 | +0.634 | +0.226% | 0.728 | +21.375 | +0.527% |
| ELU | 0.999 | +0.040 | -0.018% | 0.934 | +4.498 | -0.104% | 1.005 | -0.401 | -0.058% | 0.973 | +2.232 | +0.249% | 0.990 | +0.785 | +0.781% | 0.931 | +5.326 | -0.834% |

For layer 9 (Table 17), we found that the results are patterns that mirror layer 8: Test1 is comparatively stable, where Test2 on $p_2$ and $p_3$ tends to have lower GR and larger positive $\Delta_{gen}$. Accuracy differences are mostly small, with occasional larger deviations for trReLU, Sigmoid and Softplus in specific columns.

Table 18: Accuracy differences ($\Delta_{acc}$, %) and generalization metrics (GR, $\Delta_{gen}$) per activation and classifier for ten-layer ACR-GNN after applying the dynamic PTQ.

| A/F | $p_1$ Test1 GR | $\Delta_{gen}$ | $\Delta_{acc}$ | $p_1$ Test2 GR | $\Delta_{gen}$ | $\Delta_{acc}$ | $p_2$ Test1 GR | $\Delta_{gen}$ | $\Delta_{acc}$ | $p_2$ Test2 GR | $\Delta_{gen}$ | $\Delta_{acc}$ | $p_3$ Test1 GR | $\Delta_{gen}$ | $\Delta_{acc}$ | $p_3$ Test2 GR | $\Delta_{gen}$ | $\Delta_{acc}$ |
|---|---|---|---|---|---|---|---|---|---|---|---|---|---|---|---|---|---|---|
| ReLU | 0.997 | +0.230 | +0.186% | 0.959 | +2.822 | +0.151% | 1.008 | -0.702 | -0.098% | 0.922 | +6.501 | +0.303% | 0.992 | +0.596 | +0.546% | 0.752 | +19.319 | -5.132% |
| ReLU6 | 0.977 | +1.257 | +0.433% | 0.927 | +4.003 | +0.850% | 1.019 | -1.180 | -0.316% | 0.729 | +17.091 | -0.303% | 0.986 | +0.864 | +1.580% | 1.052 | -3.290 | -0.296% |
| trReLU | 0.988 | +0.633 | +0.141% | 0.955 | +2.360 | -0.454% | 1.020 | -1.272 | +0.000% | 0.628 | +23.443 | +0.000% | 1.015 | -0.930 | -1.065% | 0.809 | +11.485 | -0.823% |
| GELU | 0.994 | +0.383 | +0.212% | 0.930 | +4.780 | +0.249% | 1.007 | -0.589 | -0.058% | 0.866 | +11.097 | +1.373% | 0.986 | +1.051 | +0.053% | 0.851 | +11.470 | +0.639% |
| Sigmoid | 0.988 | +0.628 | -0.159% | 0.929 | +3.877 | -0.259% | 1.020 | -1.263 | -0.009% | 0.627 | +23.548 | +0.414% | 0.939 | +3.545 | -1.740% | 0.497 | +29.314 | +0.985% |
| SiLU | 0.996 | +0.249 | +0.004% | 0.959 | +2.773 | +0.306% | 1.005 | -0.450 | +0.013% | 0.854 | +12.104 | -0.130% | 0.994 | +0.473 | +0.062% | 1.010 | -0.754 | -0.383% |
| Softplus | 0.998 | +0.137 | +0.004% | 0.906 | +6.402 | +1.027% | 1.007 | -0.619 | +0.009% | 0.967 | +2.750 | -0.050% | 1.002 | -0.176 | -0.626% | 0.997 | +0.219 | -0.668% |
| ELU | 0.996 | +0.284 | +0.133% | 0.941 | +4.050 | -0.141% | 1.003 | -0.287 | +0.027% | 0.935 | +5.405 | -0.306% | 0.998 | +0.177 | +0.067% | 0.510 | +37.918 | +1.555% |

For layer 10 (Table 18), the deepest layer exhibits the strongest split between Test1 and Test2. Several activations maintain reasonable Test1 GR on $p_1$ and $p_2$, but Test2—in particular on $p_3$—shows the largest gaps and the lowest GR. A few $\Delta_{acc}$ entries become sizable (e.g., ReLU on $p_3$ for Test2), yet many cells still stay within the low-percent range.

Across layers, dynamic PTQ preserves accuracy in the majority of settings: most $\Delta_{acc}$ values are small (often within a few tenths of a percent), with occasional larger swings that concentrate in deeper layers and on the more challenging Test2 split (especially for $p_2$ and $p_3$). Generalization behavior (GR, $\Delta_{gen}$) varies notably with both activation and depth: near-shallow layers and Test1 tend to remain close to GR $\approx 1$ with small gaps, Test2 consistently surfaces larger positive $\Delta_{gen}$, and certain activations (e.g., trReLU, ReLU6, GELU) are comparatively more robust in several layers, while others (e.g., Softplus or specific cases of ReLU and SiLU at depth) show reduced GR and larger gaps. Taken together, these results indicate that dynamic PTQ retains predictive performance while the choice of activation and the evaluation split (Test2) primarily govern robustness; deeper stacks accentuate these effects, so selecting stable activations (e.g., trReLU, ReLU6 and GELU in the layers where they maintain higher GR and smaller $\Delta_{gen}$) is recommended.

After comparing the original model with its dynamically quantized variant, the reviewer suggested further analysis of how accuracy varies with the quantization bit width. To address this, we applied fake quantization to the full-precision model and evaluated performance under different bit-width configurations.

The more detailed 32-bit configuration corresponds to the unmodified full-precision model and serves as the baseline. For the 8-bit configuration, we apply PyTorch's dynamic post-training quantization (PTQ), which produces genuinely quantized Linear layers with int8 weights stored in the model and quantized kernels executed at runtime. For intermediate and lower precisions (16, 7, 6, 5, 4, and 2 bits), we employ fake quantization: weights are quantized to the corresponding integer levels and then immediately dequantized back to float32. As a result, the model remains in float32 format, with no reduction in model size or inference time; only the accuracy is affected, allowing us to isolate the impact of precision on predictive performance. The quantization scheme was mentioned in the survey of Gholami et al. (2022).

The experiments were conducted with the following settings:

- Number of layers from 1 to 3.
- Activation function: ReLU, ReLU6, trReLU, GELU, Sigmoid, SiLU, Softplus, and ELU.
- Classifiers: $p_1$, $p_2$, $p_3$
- Data split. Train, Test1, and Test2 for the synthetic data.
- "Fake Quantization". [2,4,5,6,7,8,16,32]-bit.

To isolate the most significant accuracy degradations, we filtered all bit-width transitions using two criteria: (i) both bit-widths must be greater than 4, thereby excluding the trivial drop associated with the 4 to 2-bit collapse; and (ii) the relative accuracy loss must exceed 10%.

The resulting tables (19–26) report, for each activation function, classifier, the data split (Train/Test1/Test2), the layer index, the transition between bit-widths (`From bits` and `To bits`), and the computed accuracy difference. These entries highlight only the meaningful degradation patterns that occur before entering the low-precision regime.

Table 19: Accuracy drops across bit-width reductions for the ReLU activation function. Only over 10% drops are reported.

| Split | Classifier | Activation Function | Layer | From bits | To bits | Drop |
|-------|-----------|--------------------|-------|-----------|--------|-----------|
| Train | $p_1$ | ReLU | 2 | 6 | 5 | -0.112864 |
| Train | $p_1$ | ReLU | 3 | 5 | 4 | -0.104249 |
| Test1 | $p_1$ | ReLU | 2 | 6 | 5 | -0.106862 |
| Test1 | $p_1$ | ReLU | 3 | 5 | 4 | -0.117296 |
| Test2 | $p_1$ | ReLU | 2 | 6 | 5 | -0.101236 |
| Test2 | $p_1$ | ReLU | 2 | 5 | 4 | -0.115250 |
| Test2 | $p_1$ | ReLU | 3 | 5 | 4 | -0.279317 |

According to the Table19, ReLU accuracy drops become substantial once precision falls below 6 bits. Across all splits (Train, Test1, Test2), the transitions from 6 to 5 bits and especially from 5 to 4 bits

produce large degradations. These effects are consistent across layers 2 and 3, indicating that in the synthetic dataset, ReLU requires at least 6-bit precision to maintain acceptable performance.

Table 20: Accuracy drops across bit-width reductions for the RELU6 activation function. Only over 10% drops are reported.

| Split | Classifier | Activation Function | Layer | From bits | To bits | Drop |
|-------|-----------|---------------------|-------|-----------|---------|------|
| Train | $p_1$ | ReLU6 | 1 | 5 | 4 | -0.437388 |
| Train | $p_1$ | ReLU6 | 2 | 5 | 4 | -0.276845 |
| Test1 | $p_1$ | ReLU6 | 1 | 5 | 4 | -0.437130 |
| Test1 | $p_1$ | ReLU6 | 2 | 5 | 4 | -0.265718 |
| Test2 | $p_1$ | ReLU6 | 1 | 5 | 4 | -0.257773 |
| Test2 | $p_1$ | ReLU6 | 2 | 5 | 4 | -0.268905 |

According to the Table 20, ReLU6 exhibits severe vulnerability to low-bit quantization. Every observed drop occurs from 5 to 4-bit transition, with extremely large decreases across all splits and layers. This activation is the least robust in the synthetic dataset, showing catastrophic collapse once precision drops below 5 bits.

Table 21: Accuracy drops across bit-width reductions for the trRELU activation function. Only over 10% drops are reported.

| Split | Classifier | Activation Function | Layer | From bits | To bits | Drop |
|-------|-----------|---------------------|-------|-----------|---------|------|
| Train | $p_1$ | trReLU | 1 | 5 | 4 | -0.337373 |
| Train | $p_1$ | trReLU | 2 | 6 | 5 | -0.100228 |
| Train | $p_1$ | trReLU | 2 | 5 | 4 | -0.172236 |
| Test1 | $p_1$ | trReLU | 1 | 5 | 4 | -0.333982 |
| Test1 | $p_1$ | trReLU | 2 | 5 | 4 | -0.181846 |
| Test2 | $p_1$ | trReLU | 2 | 16 | 8 | -0.202003 |
| Test2 | $p_1$ | trReLU | 2 | 5 | 4 | -0.280650 |

According to the Table 21, trReLU shows a mix of moderate and severe degradation. High-precision transitions such as from 6 to 5 bits already produce notable drops, and from 5 to 4-bit transition causes very large declines.This suggests trReLU is highly sensitive to quantization noise and requires more than 6 bits for stability.

Table 22: Accuracy drops across bit-width reductions for the GELU activation function. Only over 10% drops are reported.

| Split | Classifier | Activation Function | Layer | From bits | To bits | Drop |
|-------|-----------|---------------------|-------|-----------|---------|------|
| Train | $p_1$ | GELU | 3 | 5 | 4 | -0.136041 |
| Test1 | $p_1$ | GELU | 3 | 5 | 4 | -0.128261 |
| Test2 | $p_1$ | GELU | 1 | 7 | 6 | -0.120834 |
| Test2 | $p_1$ | GELU | 1 | 5 | 4 | -0.106280 |
| Test2 | $p_1$ | GELU | 3 | 5 | 4 | -0.165616 |
| Test2 | $p_2$ | GELU | 2 | 5 | 4 | -0.152574 |

According to the Table 22, GELU shows relatively moderate deterioration from 7 to 6 bits and more severe drops from 5 to 4 bits. Although GELU performs better than ReLU6 or trReLU, it still shows meaningful degradation below 6 bits, particularly at 5 bits and below.

According to the Table 23, Sigmoid exhibits extremely large decreases across all splits from 5 to 4-bit transition. The collapse is immediate and consistent across layers, indicating that Sigmoid is highly unstable under aggressive quantization. No drops were observed at higher bit-width transitions, confirming that the sensitivity is primarily to the 4-bit regime.

Table 23: Accuracy drops across bit-width reductions for the Sigmoid activation function. Only over 10% drops are reported.

| Split | Classifier | Activation Function | Layer | From bits | To bits | Drop |
|-------|-----------|--------------------|-------|-----------|---------|------|
| Train | $p_1$ | Sigmoid | 1 | 5 | 4 | -0.461539 |
| Test1 | $p_1$ | Sigmoid | 1 | 5 | 4 | -0.465116 |
| Test2 | $p_1$ | Sigmoid | 1 | 5 | 4 | -0.383759 |

Table 24: Accuracy drops across bit-width reductions for the SiLU activation function. Only over 10% drops are reported.

| Split | Classifier | Activation Function | Layer | From bits | To bits | Drop |
|-------|-----------|--------------------|-------|-----------|---------|------|
| Train | $p_1$ | SiLU | 1 | 6 | 5 | -0.196592 |
| Train | $p_1$ | SiLU | 1 | 5 | 4 | -0.274406 |
| Train | $p_1$ | SiLU | 3 | 5 | 4 | -0.252801 |
| Train | $p_2$ | SiLU | 2 | 5 | 4 | -0.122887 |
| Test1 | $p_1$ | SiLU | 1 | 6 | 5 | -0.197586 |
| Test1 | $p_1$ | SiLU | 1 | 5 | 4 | -0.275046 |
| Test1 | $p_1$ | SiLU | 3 | 5 | 4 | -0.253869 |
| Test1 | $p_2$ | SiLU | 2 | 5 | 4 | -0.127234 |
| Test2 | $p_1$ | SiLU | 1 | 6 | 5 | -0.223691 |
| Test2 | $p_1$ | SiLU | 1 | 5 | 4 | -0.215946 |
| Test2 | $p_1$ | SiLU | 2 | 6 | 5 | -0.103613 |
| Test2 | $p_1$ | SiLU | 2 | 5 | 4 | -0.147963 |
| Test2 | $p_1$ | SiLU | 3 | 6 | 5 | -0.127175 |
| Test2 | $p_1$ | SiLU | 3 | 5 | 4 | -0.192600 |
| Test2 | $p_2$ | SiLU | 3 | 5 | 4 | -0.139352 |

According to the Table 24, SiLU displays significant instability from 6 to 5 bits and dramatic degradation from 5 to 4 bits across splits and layer. Additional small-to-moderate decreases appear from 6 to 5 bits, reinforcing that SiLU also requires at least 6 bits for reliable performance. Classifiers $p_1$ and $p_2$ both exhibit the same pattern, suggesting this behavior is activation-dependent, not classifier-dependent.

Table 25: Accuracy drops across bit-width reductions for the Softplus activation function. Only over 10% drops are reported.

| Split | Classifier | Activation Function | Layer | From bits | To bits | Drop |
|-------|-----------|--------------------|-------|-----------|---------|------|
| Train | $p_1$ | Softplus | 1 | 6 | 5 | -0.132967 |
| Train | $p_1$ | Softplus | 1 | 5 | 4 | -0.332597 |
| Train | $p_1$ | Softplus | 3 | 6 | 5 | -0.126885 |
| Train | $p_2$ | Softplus | 2 | 5 | 4 | -0.106855 |
| Train | $p_3$ | Softplus | 1 | 5 | 4 | -0.110549 |
| Test1 | $p_1$ | Softplus | 1 | 6 | 5 | -0.131886 |
| Test1 | $p_1$ | Softplus | 1 | 5 | 4 | -0.334822 |
| Test1 | $p_1$ | Softplus | 3 | 6 | 5 | -0.130383 |
| Test1 | $p_2$ | Softplus | 2 | 5 | 4 | -0.121574 |
| Test1 | $p_3$ | Softplus | 1 | 5 | 4 | -0.113730 |
| Test2 | $p_1$ | Softplus | 1 | 5 | 4 | -0.295493 |
| Test2 | $p_1$ | Softplus | 2 | 5 | 4 | -0.141838 |
| Test2 | $p_1$ | Softplus | 3 | 6 | 5 | -0.208020 |
| Test2 | $p_2$ | Softplus | 3 | 5 | 4 | -0.122095 |
| Test2 | $p_3$ | Softplus | 1 | 5 | 4 | -0.205341 |

According to the Table 25, Softplus becomes unstable below 6 bits for all classifiers and splits, especially in Test2.

Table 26: Accuracy drops across bit-width reductions for the ELU activation function. Only over 10% drops are reported.

| Split | Classifier | Activation Function | Layer | From bits | To bits | Drop |
|-------|-----------|---------------------|-------|-----------|---------|------|
| Train | $p_1$ | ELU | 1 | 6 | 5 | -0.170322 |
| Train | $p_1$ | ELU | 1 | 5 | 4 | -0.285962 |
| Train | $p_1$ | ELU | 3 | 6 | 5 | -0.105417 |
| Train | $p_1$ | ELU | 3 | 5 | 4 | -0.229503 |
| Train | $p_2$ | ELU | 2 | 5 | 4 | -0.107738 |
| Train | $p_3$ | ELU | 3 | 5 | 4 | -0.108801 |
| Test1 | $p_1$ | ELU | 1 | 6 | 5 | -0.168892 |
| Test1 | $p_1$ | ELU | 1 | 5 | 4 | -0.287780 |
| Test1 | $p_1$ | ELU | 3 | 6 | 5 | -0.103502 |
| Test1 | $p_1$ | ELU | 3 | 5 | 4 | -0.229021 |
| Test1 | $p_2$ | ELU | 2 | 5 | 4 | -0.122153 |
| Test2 | $p_1$ | ELU | 1 | 5 | 4 | -0.195698 |
| Test2 | $p_1$ | ELU | 3 | 6 | 5 | -0.168966 |
| Test2 | $p_1$ | ELU | 3 | 5 | 4 | -0.107937 |

According to the Table 26, ELU behaves similarly to Softplus and SiLU: moderate sensitivity at 6 bits and severe instability at 4 bits. Drops appear across all classifiers, layers, and splits, reinforcing the conclusion that ELU is reliable only down to approximately 6 bits.

Across all activation functions and classifiers, the synthetic dataset exhibits a strong and consistent collapse in accuracy when the precision is reduced below 6 bits. The transitions from 6 to 5 bits already introduce substantial degradation (typically 10–20%), while from 5 to 4-bit transition results in universally severe drops (20–45%). Among activation functions, ReLU6, Sigmoid, and Softplus are the most sensitive, with catastrophic failures at 4 bits, whereas GELU, ReLU, and ELU show slightly more resilience but still deteriorate sharply below 6 bits. Overall, unlike the PPI dataset, which remained stable down to 6 bits, the synthetic dataset displays high quantization sensitivity, confirming that meaningful performance is preserved only at 8–6 bits and collapses rapidly at 5 bits and below.

To test the technique not only on synthetic data, we chose the Protein-Protein Interactions (PPI) benchmark Zitnik & Leskovec (2017) as in the reference paper of Barceló et al. (2020). The PPI dataset consists of graph-level mini-batches, with separate splits for Training, Validation, and Testing.

Table 27: Dataset summary. PPI benchmark.

| Dataset | Num Graphs | Node Feature Dim | Label Dim | Avg Active Labels/Node | Avg Degree |
|---------|-----------|------------------|-----------|------------------------|------------|
| Train | 20 | 50 | 121 | 37.20 | 54.62 |
| Validation | 2 | 50 | 121 | 35.64 | 61.07 |
| Test | 2 | 50 | 121 | 36.22 | 58.64 |

In Table 27, we present a summary of the PPI dataset, which consists of 20 training graphs, 2 validation graphs, and 2 test graphs. Each graph contains nodes with 50-dimensional features and supports multi-label classification with 121 possible labels. On average, each node is associated with approximately 36 labels, indicating a densely labeled dataset. The average node degree is also high, ranging from 54.6 in the training set to 61.1 in the validation set, reflecting the dense connectivity of the protein-protein interaction graphs. The dataset presents a complex multi-label classification task with consistently rich structure across all splits.

The statistics of the dataset presented in Table 28 contain large graphs with varying sizes between the train, the validation, and the test splits. Training graphs range from 591 to 3,480 nodes, with an average of 2,245 nodes per graph, and between 7,708 and 106,754 edges (average 61,318 edges). Validation graphs are more consistent in size, with 3,230 to 3,284 nodes and 97,446 to 101,474 edges,

Table 28: Dataset statistics summary. PPI benchmark.

| | Node | | | Edge | | |
|---|---|---|---|---|---|---|
| Dataset | Min | Max | Avg | Min | Max | Avg |
| Train | 591 | 3480 | 2245.30 | 7708 | 106754 | 61318.40 |
| Validation | 3230 | 3284 | 3257.00 | 97446 | 101474 | 99460.00 |
| Test | 2300 | 3224 | 2762.00 | 61328 | 100648 | 80988.00 |

averaging 3,257 nodes and 99,460 edges. The test graphs have 2,300 to 3,224 nodes, averaging 2,762 nodes, and 61,328 to 100,648 edges, averaging 80,988. These statistics confirm that the dataset contains large and densely connected graphs and demonstrate a distributional shift in graph size and edge count between training and test data. This information is helpful in evaluating the model's ability to generalize to unseen and variable graph structures.

One key difference between the synthetic data and the PPI dataset is that the latter involves a multi-label classification task, rather than a binary classification task, because the PPI dataset is a common benchmark where each node (representing proteins) can have multiple labels, such as protein functions or interactions. Also, it is important to mention the key differences between the synthetic data and the real one. Here, the authors used the code function `EarlyStopping`: Utility for stopping training early if no further improvement is observed. The second difference is that the code is structured to run multiple experiments to collect statistics (mean and standard deviation) of the model performance, ensuring that the results are robust across different random initializations. In this case, we performed the experiments 10 times for each model, with a combination layer equal to 1 and a number of layers ranging from 1 to 10. The number of hidden dimensions is equal to 256.

We applied the same eight activation functions to train the model. We also continue the experimental flow for real-world data, focusing on running time (Table 29 and Figure 8), speedup (Figure 9), size reduction (Table 33), and analysis of accuracy.

We analyze the total training time of the ACR-GNN across ten layers for different activation functions. Table 29 reports the total runtime in seconds and minutes, while Figure 8 visualizes the results across depths.

Table 29: Training time per activation function

| A/F | ReLU | ReLU6 | trReLU | GELU | Sigmoid | SiLU | Softplus | ELU |
|---|---|---|---|---|---|---|---|---|
| Time (min) | 204.70 | 244.20 | 187.60 | 192.40 | 229.20 | 232.50 | 250.10 | 234.40 |
| Time (s) | 12286.30 | 14650.70 | 11261.60 | 11541.90 | 13745.00 | 13951.00 | 15007.00 | 14059.50 |

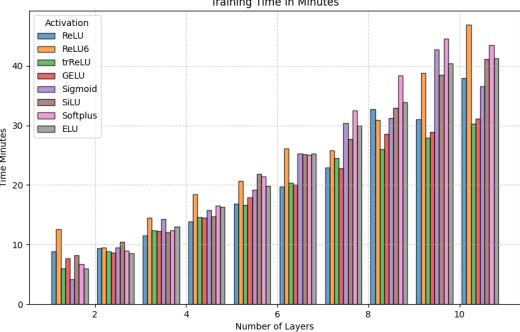

Figure 8: Training time by depth of the ACR-GNN.

The results show substantial variability depending on the activation function. Piecewise activations such as trReLU (187.6 min) and ReLU (204.7 min) yield the fastest training times, while smooth

activations such as Softplus (250.1 min), ReLU6 (244.2 min), and SiLU (232.5 min) incur significant overhead. Sigmoid also ranks among the slower functions (229.2 min).

Table 30: Slowest and fastest activation functions across the depth of the ACR-GNN.

|         | 1       | 2    | 3     | 4     | 5      | 6     | 7        | 8        | 9        | 10     |
|---------|---------|------|-------|-------|--------|-------|----------|----------|----------|--------|
| Fastest | Sigmoid | ELU  | ReLU  | ReLU  | trReLU | ReLU  | GELU     | trReLU   | trReLU   | trReLU |
| Slowest | ReLU6   | SiLU | ReLU6 | ReLU6 | SiLU   | ReLU6 | Softplus | Softplus | Softplus | ReLU6  |

As shown in Table 29, the training time for all activation functions increases dramatically after the second layer. This highlights that not only the type of activation function influences performance time, but also the depth of the model.

Depth-wise analysis (Table 30) confirms this pattern: trReLU frequently provides the lowest training time at deeper layers (5, 8–10), while ReLU6 consistently emerges as the slowest. These findings indicate that the choice of activation function significantly impacts computational efficiency on PPI, with piecewise functions offering faster convergence than their smooth counterparts.

We measured the size of the model (in Table 31) and obtained that the choice of activation function did not influence the size of the model.

Table 31: PPI. Model size in MB as a function of the number of layers.

| Layers    | 1    | 2    | 3    | 4    | 5    | 6   | 7   | 8   | 9    | 10   |
|-----------|------|------|------|------|------|-----|-----|-----|------|------|
| Size (MB) | 0.92 | 1.72 | 2.51 | 3.31 | 4.11 | 4.9 | 5.7 | 6.5 | 7.29 | 8.09 |

To assess the computational efficiency of dPTQ, we measured the elapsed time (Table 32) of each model across different activation functions and datasets.

Table 32: Total elapsed time (s) per activation function and datasets before and after applying dPTQ

|          | Train    |           | Test     |           | Validation |           |
|----------|----------|-----------|----------|-----------|------------|-----------|
| A/F      | Original | Quantized | Original | Quantized | Original   | Quantized |
| ReLU     | 22.46    | 23.13     | 2.56     | 2.57      | 3.04       | 3.02      |
| ReLU6    | 21.18    | 22.44     | 2.54     | 2.69      | 2.90       | 3.17      |
| trReLU   | 19.93    | 20.61     | 2.50     | 2.51      | 2.89       | 3.00      |
| GELU     | 23.89    | 25.37     | 2.82     | 2.91      | 3.40       | 3.25      |
| Sigmoid  | 22.52    | 24.53     | 2.72     | 2.85      | 3.17       | 3.24      |
| SiLU     | 23.14    | 24.06     | 2.72     | 2.87      | 3.51       | 3.16      |
| Softplus | 21.65    | 24.97     | 2.76     | 3.04      | 3.27       | 3.23      |
| ELU      | 26.13    | 26.05     | 3.31     | 3.22      | 3.90       | 3.52      |

Table 32 reports the total elapsed time (in seconds) for training, testing, and validation phases across activation functions, comparing original and quantized models. The results indicate that dynamic PTQ introduces only marginal differences in runtime across all phases and activations. In most cases, quantized models require slightly longer execution time (e.g., ReLU6 and Softplus), while in a few instances, minor improvements are observed (e.g., ELU in Test and Validation). Overall, the runtime overhead of quantization remains negligible, suggesting that the primary benefit of dynamic PTQ lies in memory and storage efficiency rather than acceleration.

As for the synthetic data, we measure the speedup (Figure 9) of this type of quantization technique. We report the mean dynamic PTQ speedup across 10 layers, defined as the ratio of non-quantized to quantized execution time (original time / dPTQ time), which indicates whether dynamic PTQ reduces or increases runtime.

Figure 9 reports the mean speedup values across layers for different activation functions. Overall, dynamic PTQ yields values close to 1, indicating only minor runtime benefits. ELU demonstrates the most consistent improvement, with speedup up to 1.14 in validation and above 1.07 in test, followed

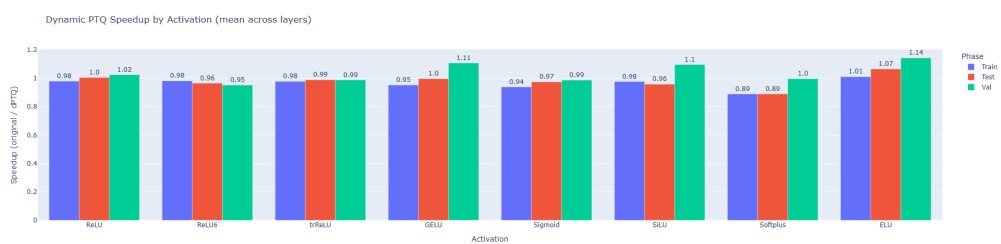

Figure 9: PPI. Dynamic PTQ Speedup by Activation (mean across layers).

by SiLU and GELU, which also provide modest acceleration during validation. In contrast, Softplus incurs consistent slowdowns (speedup $\approx 0.89$ in training and testing), while Sigmoid and ReLU6 remain below 1, showing limited suitability for quantized execution. These results indicate that smooth activations such as ELU, SiLU, and GELU are better aligned with quantized computation, whereas Softplus and Sigmoid are unfavorable for efficient PTQ deployment.

We report the results in Table 33 about the difference of the model's size. We calculated the $\Delta_{Size}$ and Reduction (%) across the depth. The main result of this experiment is the following: the total reduction in size is $\approx 74\%$. That is really good and significant, for example, for the application part of the quantization, where the model can be used on a low-power computer.

Table 33: PPI. Influence of the dPTQ on the size of the model

| Layers | Original (MB) | Quantized (MB) | Delta | Reduction (%) |
|--------|---------------|----------------|----------|---------------|
| 1 | 0.922108 | 0.242060 | 0.680048 | -73.7 |
| 2 | 1.718266 | 0.450790 | 1.267476 | -73.8 |
| 3 | 2.514808 | 0.659584 | 1.855224 | -73.8 |
| 4 | 3.311350 | 0.868378 | 2.442972 | -73.8 |
| 5 | 4.107892 | 1.077172 | 3.030720 | -73.8 |
| 6 | 4.904370 | 1.285972 | 3.618398 | -73.8 |
| 7 | 5.700912 | 1.494783 | 4.206129 | -73.8 |
| 8 | 6.497390 | 1.703594 | 4.793796 | -73.8 |
| 9 | 7.293933 | 1.912405 | 5.381528 | -73.8 |
| 10 | 8.090486 | 2.121216 | 5.969270 | -73.8 |

As for the synthetic data, we constructed tables with specific structural requirements to better examine the influence of dynamic PTQ on PPI data. The impact of dynamic PTQ was assessed by calculating the Generalization Ratio (GR), the Generalization Gap ($\Delta_{gen}$), and the accuracy difference between the original and quantized models ($\Delta_{acc}$). The results are presented in Tables 34 –Tables 43. Below, we summarize the principal observations layer by layer.

For layer 1 (Table 34) ReLU6 achieves the strongest validation GR (0.917, $\Delta_{gen} = +0.045$) with negligible $\Delta_{acc}$. Softplus performs worst (GR 0.667/0.802 with the largest gaps).

For layer 2 (Table 35) ELU and Sigmoid are comparatively stable (GR $\approx 0.72$–$0.74$), whereas ReLU degrades (GR 0.606/0.637 with $\Delta_{gen} > 0.22$). Accuracy changes remain within $\pm 0.03\%$.

For layer 3 (Table 36) GELU and trReLU lead (GR $\approx 0.73$), while Softplus is lowest (0.551/0.531). $\Delta_{acc}$ remains negligible.

For layer 4 (Table 37) ReLU6 clearly dominates (0.902/0.878, small $\Delta_{gen} \approx 0.05$–$0.07$). ELU is weakest ($\approx 0.60/0.56$, large gaps). $\Delta_{acc} \approx 0\%$.

For layer 5 (Table 38) trReLU is strongest (0.921/0.906, minimal $\Delta_{gen}$). Softplus and ELU are lowest ($\approx 0.59$). Sigmoid shows a small positive $\Delta_{acc}$ on Test (+0.031%), while SiLU has a small negative $\Delta_{acc}$ on Validation (-0.033%).

Table 34: PPI. Accuracy differences ($\Delta_{acc}$, %) and generalization metrics (GR, $\Delta_{gen}$) per activation and dataset for one-layer ACR-GNN after applying the dynamic PTQ.

| A/F | Test | | | Validation | | |
|---|---|---|---|---|---|---|
| | GR | $\Delta_{gen}$ | $\Delta_{acc}$ | GR | $\Delta_{gen}$ | $\Delta_{acc}$ |
| ReLU | 0.821 | +0.108 | -0.018% | 0.866 | +0.081 | +0.001% |
| ReLU6 | 0.835 | +0.090 | +0.007% | 0.917 | +0.045 | +0.006% |
| trReLU | 0.728 | +0.143 | +0.001% | 0.722 | +0.146 | +0.000% |
| GELU | 0.716 | +0.168 | -0.005% | 0.848 | +0.090 | -0.006% |
| Sigmoid | 0.791 | +0.109 | -0.003% | 0.741 | +0.135 | -0.001% |
| SiLU | 0.765 | +0.138 | -0.010% | 0.854 | +0.086 | -0.006% |
| Softplus | 0.667 | +0.197 | +0.025% | 0.802 | +0.118 | +0.016% |
| ELU | 0.719 | +0.156 | +0.015% | 0.774 | +0.125 | +0.007% |

Table 35: PPI. Accuracy differences ($\Delta_{acc}$, %) and generalization metrics (GR, $\Delta_{gen}$) per activation and dataset for two-layer ACR-GNN after applying the dynamic PTQ.

| A/F | Test | | | Validation | | |
|---|---|---|---|---|---|---|
| | GR | $\Delta_{gen}$ | $\Delta_{acc}$ | GR | $\Delta_{gen}$ | $\Delta_{acc}$ |
| ReLU | 0.606 | +0.241 | -0.001% | 0.637 | +0.222 | +0.031% |
| ReLU6 | 0.703 | +0.161 | +0.004% | 0.668 | +0.180 | -0.009% |
| trReLU | 0.694 | +0.157 | -0.001% | 0.687 | +0.161 | +0.002% |
| GELU | 0.681 | +0.195 | +0.011% | 0.723 | +0.170 | +0.009% |
| Sigmoid | 0.743 | +0.133 | +0.004% | 0.735 | +0.137 | -0.000% |
| SiLU | 0.677 | +0.197 | +0.015% | 0.635 | +0.223 | +0.006% |
| Softplus | 0.681 | +0.200 | -0.004% | 0.655 | +0.216 | +0.005% |
| ELU | 0.717 | +0.172 | -0.003% | 0.721 | +0.170 | +0.008% |

Table 36: PPI. Accuracy differences ($\Delta_{acc}$, %) and generalization metrics (GR, $\Delta_{gen}$) per activation and dataset for three-layer ACR-GNN after applying the dynamic PTQ.

| A/F | Test | | | Validation | | |
|---|---|---|---|---|---|---|
| | GR | $\Delta_{gen}$ | $\Delta_{acc}$ | GR | $\Delta_{gen}$ | $\Delta_{acc}$ |
| ReLU | 0.576 | +0.261 | -0.006% | 0.629 | +0.228 | -0.004% |
| ReLU6 | 0.598 | +0.216 | +0.034% | 0.588 | +0.221 | +0.008% |
| trReLU | 0.727 | +0.143 | +0.040% | 0.691 | +0.162 | +0.007% |
| GELU | 0.730 | +0.164 | -0.027% | 0.720 | +0.170 | -0.014% |
| Sigmoid | 0.749 | +0.123 | +0.005% | 0.627 | +0.183 | +0.007% |
| SiLU | 0.646 | +0.214 | -0.000% | 0.713 | +0.173 | -0.001% |
| Softplus | 0.551 | +0.282 | +0.001% | 0.531 | +0.295 | +0.000% |
| ELU | 0.653 | +0.215 | +0.024% | 0.639 | +0.223 | -0.010% |

For layer 6 (Table 39) Sigmoid emerges as best (0.880/0.910, smallest gaps), followed by trReLU (0.870/0.879). SiLU and ELU are weaker ($\approx$ 0.62). Softplus shows the largest negative $\Delta_{acc}$ (-0.025/-0.031%), though still small.

For layer 7 (Table 40) Sigmoid and GELU lead (0.866/0.841 and 0.744/0.759). SiLU is weakest ($\approx$ 0.59 with largest $\Delta_{gen}$). Accuracy changes remain near zero.

For layer 8 (Table 41) trReLU and Sigmoid are strongest (0.855/0.845 and 0.859/0.823). Softplus is lowest ($\approx$ 0.63/0.61). $\Delta_{acc}$ values are minimal.

For layer 9 (Table 42) trReLU again achieves best generalization (0.864/0.846). Softplus is weakest ($\approx$ 0.59). $\Delta_{acc}$ is small, with mixed signs for Sigmoid (+0.023% Test, -0.015% Validation splits).

Table 37: PPI. Accuracy differences ($\Delta_{acc}$, %) and generalization metrics (GR, $\Delta_{gen}$) per activation and dataset for four-layer ACR-GNN after applying the dynamic PTQ.

| A/F | Test | | | Validation | | |
|-----|------|------|------|------------|------|------|
| | GR | $\Delta_{gen}$ | $\Delta_{acc}$ | GR | $\Delta_{gen}$ | $\Delta_{acc}$ |
| ReLU | 0.564 | +0.269 | +0.001% | 0.570 | +0.265 | -0.002% |
| ReLU6 | 0.902 | +0.053 | -0.000% | 0.878 | +0.065 | -0.004% |
| trReLU | 0.708 | +0.150 | -0.000% | 0.707 | +0.151 | -0.004% |
| GELU | 0.589 | +0.252 | +0.006% | 0.597 | +0.247 | -0.003% |
| Sigmoid | 0.822 | +0.088 | +0.003% | 0.762 | +0.118 | +0.003% |
| SiLU | 0.612 | +0.231 | -0.001% | 0.595 | +0.241 | +0.001% |
| Softplus | 0.674 | +0.200 | -0.005% | 0.653 | +0.213 | +0.007% |
| ELU | 0.600 | +0.252 | -0.010% | 0.556 | +0.280 | -0.003% |

Table 38: PPI. Accuracy differences ($\Delta_{acc}$, %) and generalization metrics (GR, $\Delta_{gen}$) per activation and dataset for five-layer ACR-GNN after applying the dynamic PTQ.

| A/F | Test | | | Validation | | |
|-----|------|------|------|------------|------|------|
| | GR | $\Delta_{gen}$ | $\Delta_{acc}$ | GR | $\Delta_{gen}$ | $\Delta_{acc}$ |
| ReLU | 0.705 | +0.180 | +0.003% | 0.673 | +0.200 | +0.018% |
| ReLU6 | 0.786 | +0.110 | -0.006% | 0.754 | +0.126 | +0.001% |
| trReLU | 0.921 | +0.039 | +0.002% | 0.906 | +0.046 | -0.002% |
| GELU | 0.581 | +0.252 | -0.001% | 0.566 | +0.261 | +0.018% |
| Sigmoid | 0.770 | +0.112 | +0.031% | 0.755 | +0.120 | +0.006% |
| SiLU | 0.671 | +0.195 | +0.001% | 0.720 | +0.166 | -0.033% |
| Softplus | 0.598 | +0.246 | -0.003% | 0.586 | +0.253 | +0.001% |
| ELU | 0.582 | +0.251 | +0.000% | 0.574 | +0.255 | +0.000% |

Table 39: PPI. Accuracy differences ($\Delta_{acc}$, %) and generalization metrics (GR, $\Delta_{gen}$) per activation and dataset for six-layer ACR-GNN after applying the dynamic PTQ.

| A/F | Test | | | Validation | | |
|-----|------|------|------|------------|------|------|
| | GR | $\Delta_{gen}$ | $\Delta_{acc}$ | GR | $\Delta_{gen}$ | $\Delta_{acc}$ |
| ReLU | 0.730 | +0.161 | -0.003% | 0.717 | +0.169 | +0.004% |
| ReLU6 | 0.672 | +0.168 | +0.000% | 0.663 | +0.173 | +0.000% |
| trReLU | 0.870 | +0.066 | +0.012% | 0.879 | +0.062 | +0.000% |
| GELU | 0.642 | +0.214 | -0.006% | 0.628 | +0.223 | -0.002% |
| Sigmoid | 0.880 | +0.057 | -0.001% | 0.910 | +0.043 | -0.001% |
| SiLU | 0.619 | +0.222 | +0.000% | 0.611 | +0.227 | -0.015% |
| Softplus | 0.641 | +0.216 | -0.025% | 0.685 | +0.190 | -0.031% |
| ELU | 0.624 | +0.232 | +0.004% | 0.626 | +0.231 | +0.010% |

For layer 10 (Table 43) Sigmoid is strongest (0.805/0.809). ReLU is weakest (0.554/0.571, $\Delta_{gen} \approx$ 0.25). SiLU shows the largest absolute $\Delta_{acc}$ (+0.071/+0.079%), but still below 0.1%.

Dynamic PTQ preserves accuracy across all layers, with $|\Delta_{acc}| < 0.1\%$ in nearly every case. Generalization robustness varies by activation: trReLU and ReLU6 perform most consistently across layers, Sigmoid becomes increasingly stable in deeper layers, while Softplus is the weakest choice, and plain ReLU tends to degrade with depth. These findings confirm that quantized models retain generalization performance on PPI, with activation choice being the primary factor for robustness under PTQ.

After comparing the original model with its dynamically quantized variant, the reviewer suggested further analysis of how accuracy varies with the quantization bit width. To address this, we applied

Table 40: PPI. Accuracy differences ($\Delta_{acc}$, %) and generalization metrics (GR, $\Delta_{gen}$) per activation and dataset for seven-layer ACR-GNN after applying the dynamic PTQ.

| A/F | Test | | | Validation | | |
|---|---|---|---|---|---|---|
| | GR | $\Delta_{gen}$ | $\Delta_{acc}$ | GR | $\Delta_{gen}$ | $\Delta_{acc}$ |
| ReLU | 0.714 | +0.172 | -0.009% | 0.686 | +0.189 | +0.007% |
| ReLU6 | 0.641 | +0.182 | -0.001% | 0.615 | +0.195 | -0.002% |
| trReLU | 0.696 | +0.155 | +0.000% | 0.688 | +0.159 | +0.000% |
| GELU | 0.744 | +0.151 | +0.010% | 0.759 | +0.143 | -0.000% |
| Sigmoid | 0.866 | +0.061 | +0.014% | 0.841 | +0.072 | -0.000% |
| SiLU | 0.591 | +0.239 | +0.002% | 0.585 | +0.243 | +0.001% |
| Softplus | 0.655 | +0.210 | +0.003% | 0.665 | +0.204 | -0.003% |
| ELU | 0.713 | +0.177 | +0.003% | 0.732 | +0.165 | -0.001% |

Table 41: PPI. Accuracy differences ($\Delta_{acc}$, %) and generalization metrics (GR, $\Delta_{gen}$) per activation and dataset for eight-layer ACR-GNN after applying the dynamic PTQ.

| A/F | Test | | | Validation | | |
|---|---|---|---|---|---|---|
| | GR | $\Delta_{gen}$ | $\Delta_{acc}$ | GR | $\Delta_{gen}$ | $\Delta_{acc}$ |
| ReLU | 0.590 | +0.245 | +0.002% | 0.585 | +0.247 | +0.000% |
| ReLU6 | 0.795 | +0.103 | -0.005% | 0.796 | +0.102 | -0.005% |
| trReLU | 0.855 | +0.072 | -0.001% | 0.845 | +0.077 | +0.003% |
| GELU | 0.664 | +0.199 | +0.000% | 0.656 | +0.204 | +0.000% |
| Sigmoid | 0.859 | +0.064 | -0.007% | 0.823 | +0.080 | -0.012% |
| SiLU | 0.717 | +0.162 | +0.001% | 0.714 | +0.164 | +0.000% |
| Softplus | 0.631 | +0.222 | -0.006% | 0.613 | +0.233 | +0.006% |
| ELU | 0.645 | +0.211 | +0.012% | 0.675 | +0.193 | -0.004% |

Table 42: PPI. Accuracy differences ($\Delta_{acc}$, %) and generalization metrics (GR, $\Delta_{gen}$) per activation and dataset for nine-layer ACR-GNN after applying the dynamic PTQ.

| A/F | Test | | | Validation | | |
|---|---|---|---|---|---|---|
| | GR | $\Delta_{gen}$ | $\Delta_{acc}$ | GR | $\Delta_{gen}$ | $\Delta_{acc}$ |
| ReLU | 0.614 | +0.225 | -0.002% | 0.610 | +0.228 | -0.002% |
| ReLU6 | 0.699 | +0.155 | +0.001% | 0.690 | +0.159 | -0.002% |
| trReLU | 0.864 | +0.066 | -0.003% | 0.846 | +0.075 | -0.015% |
| GELU | 0.634 | +0.216 | -0.001% | 0.627 | +0.221 | -0.000% |
| Sigmoid | 0.797 | +0.093 | +0.023% | 0.840 | +0.073 | -0.015% |
| SiLU | 0.760 | +0.144 | -0.002% | 0.752 | +0.148 | -0.003% |
| Softplus | 0.589 | +0.248 | -0.001% | 0.586 | +0.249 | -0.004% |
| ELU | 0.690 | +0.187 | +0.005% | 0.710 | +0.175 | -0.014% |

fake quantization to the full-precision model and evaluated performance under different bit-width configurations. The experiments were conducted with the following settings:

- Number of layers from 1 to 3.
- Activation function: ReLU, ReLU6, trReLU, GELU, Sigmoid, SiLU, Softplus, ELU.
- Data split. Train, Validation, and Test for the PPI data.
- "Fake Quantization". [2,4,5,6,7,8,16,32]-bit. Here, 32-bit corresponds to the original full-precision model, and 8-bit corresponds to the dynamic PTQ configuration reported before.

To isolate the most significant accuracy degradations, we filtered all bit-width transitions using two criteria: (i) both bit-widths must be greater than 4, thereby excluding the trivial drop associated with

Table 43: PPI. Accuracy differences ($\Delta_{acc}$, %) and generalization metrics (GR, $\Delta_{gen}$) per activation and dataset for ten-layer ACR-GNN after applying the dynamic PTQ.

| A/F | Test | | | Validation | | |
|---|---|---|---|---|---|---|
| | GR | $\Delta_{gen}$ | $\Delta_{acc}$ | GR | $\Delta_{gen}$ | $\Delta_{acc}$ |
| ReLU | 0.554 | +0.260 | +0.008% | 0.571 | +0.250 | -0.006% |
| ReLU6 | 0.720 | +0.138 | +0.004% | 0.711 | +0.143 | +0.004% |
| trReLU | 0.732 | +0.133 | -0.008% | 0.715 | +0.141 | -0.006% |
| GELU | 0.621 | +0.225 | -0.023% | 0.588 | +0.245 | -0.008% |
| Sigmoid | 0.805 | +0.088 | +0.019% | 0.809 | +0.087 | +0.007% |
| SiLU | 0.602 | +0.232 | +0.071% | 0.586 | +0.241 | +0.079% |
| Softplus | 0.729 | +0.160 | -0.004% | 0.705 | +0.175 | +0.004% |
| ELU | 0.749 | +0.152 | -0.014% | 0.726 | +0.166 | -0.019% |

the 4 to 2-bit collapse; and (ii) the relative accuracy loss must exceed 1%. Note that we first searched for 'lose more than 10% of accuracy', but we did not observe a decrease in accuracy.

The resulting tables (44– 51) report, for each activation function, the data split (Train/Validation/Test), the layer index, the transition between bit-widths (`From bits` and `From bits`), and the computed accuracy difference. These entries highlight only the meaningful degradation patterns that occur before entering the low-precision regime.

Table 44: PPI. Accuracy drops across bit-width reductions for the RELU activation function. Only over 1% drops are reported.

| Split | Activation Function | Layer | From bits | From bits | Drop |
|---|---|---|---|---|---|
| Test | RELU | 1 | 32 | 16 | -0.010267 |
| Test | RELU | 1 | 7 | 6 | -0.027213 |
| Test | RELU | 3 | 32 | 16 | -0.013466 |
| Test | RELU | 3 | 5 | 4 | -0.032393 |
| Validation | RELU | 2 | 16 | 8 | -0.015078 |

According to the Table44, ReLU activation exhibits only small accuracy degradations across bit-widths, all below 3.3%. Drops occur primarily in the Test split, with notable decreases from 7 to 6 bits and from 5 to 4 bits, and mild drops from 32 to 16 bits. A single Validation drop is observed from 16 to 8 bits.

Table 45: PPI. Accuracy drops across bit-width reductions for the RELU6 activation function. Only over 1% drops are reported.

| Split | Activation Function | Layer | From bits | To bits | Drop |
|---|---|---|---|---|---|
| Train | RELU6 | 1 | 8 | 7 | -0.011276 |
| Train | RELU6 | 2 | 5 | 4 | -0.012176 |

According to the Table45, ReLU6 shows very limited sensitivity to quantization. Only two drops exceed the 1% threshold: one in the Train split from 8 to 7 bits, and one from 5 to 4 bits for layer 2. No drops were observed in the Validation or Test sets, indicating that ReLU6 behaves consistently and is robust under moderate bit-width reductions.

According to the Table46, trReLU is the most sensitive activation among those tested. Drops appear across all splits and layers, with several consistent degradations from 32 to 16 bits, from 7 to 6 bits, from 6 to 5 bits, and 5 to 4 bits. Both Test and Validation splits show multiple decreases, some approaching 3%. This pattern indicates that trReLU is more vulnerable to precision reduction and benefits from bit-widths greater that 6 bits.

Table 46: PPI. Accuracy drops across bit-width reductions for the trRELU activation function. Only over 1% drops are reported.

| Split | Activation Function | Layer | From bits | To bits | Drop |
|---|---|---|---|---|---|
| Test | trRELU | 1 | 32 | 16 | -0.026129 |
| Test | trRELU | 1 | 7 | 6 | -0.023775 |
| Test | trRELU | 1 | 5 | 4 | -0.013549 |
| Test | trRELU | 2 | 16 | 8 | -0.029678 |
| Test | trRELU | 2 | 6 | 5 | -0.012522 |
| Test | trRELU | 2 | 5 | 4 | -0.014027 |
| Validation | trRELU | 1 | 32 | 16 | -0.014119 |
| Validation | trRELU | 1 | 7 | 6 | -0.014122 |
| Validation | trRELU | 1 | 5 | 4 | -0.010713 |
| Validation | trRELU | 2 | 8 | 7 | -0.011061 |
| Validation | trRELU | 3 | 16 | 8 | -0.017004 |

Table 47: PPI. Accuracy drops across bit-width reductions for the GELU activation function. Only over 1% drops are reported.

| Split | Activation Function | Layer | From bits | To bits | Drop |
|---|---|---|---|---|---|
| Train | GELU | 1 | 8 | 7 | -0.011261 |
| Test | GELU | 1 | 32 | 16 | -0.012628 |
| Test | GELU | 1 | 7 | 6 | -0.025353 |
| Test | GELU | 1 | 5 | 4 | -0.017917 |
| Test | GELU | 3 | 7 | 6 | -0.015957 |
| Test | GELU | 3 | 5 | 4 | -0.016658 |
| Validation | GELU | 1 | 32 | 16 | -0.012274 |
| Validation | GELU | 1 | 7 | 6 | -0.014351 |
| Validation | GELU | 3 | 32 | 16 | -0.012390 |
| Validation | GELU | 3 | 8 | 7 | -0.010392 |
| Validation | GELU | 3 | 5 | 4 | -0.070996 |

According to the Table47, GELU displays a mixed pattern: it is stable at higher bit-widths but shows several consistent drops once bit-width falls below 6 bits. The most substantial drop occurs in the Validation split, from 5 to 4 bits (approximately 7%). Additional degradations appear across layers from 32 to 16 bits, from 7 to 6 bits, and from 5 to 4 bits. While GELU is reliable at 16–8 bits, its performance deteriorates noticeably at 6 bits and below.

Table 48: PPI. Accuracy drops across bit-width reductions for the Sigmoid activation function. Only over 1% drops are reported.

| Split | Activation Function | Layer | From bits | To bits | Drop |
|---|---|---|---|---|---|
| Test | Sigmoid | 1 | 32 | 16 | -0.011923 |
| Test | Sigmoid | 1 | 5 | 4 | -0.011844 |
| Test | Sigmoid | 2 | 6 | 5 | -0.010145 |
| Test | Sigmoid | 3 | 7 | 6 | -0.021927 |
| Validation | Sigmoid | 1 | 5 | 4 | -0.012812 |
| Validation | Sigmoid | 2 | 16 | 8 | -0.015463 |
| Validation | Sigmoid | 2 | 6 | 5 | -0.015374 |
| Validation | Sigmoid | 3 | 5 | 4 | -0.011994 |

According to the Table48, Sigmoid shows moderate sensitivity, with drops distributed across the Test and Validation splits. Small decreases occur from 32 to 16, from 7 to 6, and from 6 to 5 bits, with additional declines from 5 to 4 bits in both splits. These degradations are consistent but remain relatively small ( less than 2%), suggesting mild vulnerability to quantization, mainly below 6 bits.

Table 49: PPI. Accuracy drops across bit-width reductions for the SiLU activation function. Only over 1% drops are reported.

| Split | Activation Function | Layer | From bits | To bits | Drop |
|---|---|---|---|---|---|
| Test | SiLU | 1 | 32 | 16 | -0.012687 |
| Test | SiLU | 1 | 7 | 6 | -0.015057 |
| Test | SiLU | 2 | 5 | 4 | -0.022375 |
| Test | SiLU | 3 | 16 | 8 | -0.015068 |
| Validation | SiLU | 2 | 5 | 4 | -0.019180 |

According to the Table49, SiLU exhibits stable accuracy at higher precisions but shows drops from 32 to 16 bits and from 7 to 6 bits in the Test split. The most notable drop occurs from 5 to 4 bits, appearing in both Test and Validation splits. This indicates that SiLU maintains robustness down to 6 bits but experiences degradation at 5 bits and below.

Table 50: PPI. Accuracy drops across bit-width reductions for the Softplus activation function. Only over 1% drops are reported.

| Split | Activation Function | Layer | From bits | To bits | Drop |
|---|---|---|---|---|---|
| Test | Softplus | 1 | 7 | 6 | -0.027224 |
| Test | Softplus | 2 | 32 | 16 | -0.020117 |
| Test | Softplus | 2 | 5 | 4 | -0.010378 |
| Validation | Softplus | 1 | 6 | 5 | -0.014675 |
| Validation | Softplus | 2 | 32 | 16 | -0.016789 |
| Validation | Softplus | 2 | 5 | 4 | -0.012770 |

According to the Table50, Softplus shows several drops across both Test and Validation splits. The most substantial decreases occur from 32 to 16 bits and from 7 to 6 bits, with additional drops from 5 to 4 bits. These results indicate that while Softplus performs reliably at 16–8 bits, accuracy becomes unstable at lower precisions.

Table 51: PPI. Accuracy drops across bit-width reductions for the ELU activation function. Only over 1% drops are reported.

| Split | Activation Function | Layer | From bits | To bits | Drop |
|---|---|---|---|---|---|
| Validation | ELU | 2 | 7 | 6 | -0.012918 |
| Validation | ELU | 2 | 6 | 5 | -0.013019 |

According to the Table51, ELU is one of the most stable activations, showing only two Validation drops, both small from 7 to 6 and from 6 to 5 bits. No degradations were observed in the Test split. This suggests ELU is highly robust to quantization down to 6 bits, with only minor sensitivity at very low precision.

As a conclusion of this analysis, we found that across all activation functions, quantization down to 8 and 6 bits generally preserved the accuracy of the PPI models, with most degradations remaining below 1–2%. Noticeable accuracy drops emerged primarily in the transitions from 7 to 6 bits and from 6 to 5 bits, while the most substantial losses consistently appeared from 5 to 4 bits, regardless of activation function. Among the functions tested, trRELU and Softplus showed the greatest sensitivity to reduced precision, with multiple drops across splits and layers, whereas ReLU, ReLU6, and ELU demonstrated the highest robustness, exhibiting only occasional and comparatively minor decreases. Overall, these results indicate that ACR-GNNs retain stable performance at moderate bit-widths (16–8–6 bits), and significant accuracy degradation occurs mainly when entering the very low-precision regime (less that 5 bits).

The experiments were conducted on a Samsung Galaxy Book4 laptop equipped with an Intel Core i7-150U processor, 16 GB of RAM, and 1 TB of SSD storage. Additional experiments were conducted using Kaggle's cloud platform with an NVIDIA Tesla P100 GPU (16 GB RAM).

