# OpenReview forum: "Verifying GNNs with Readout is Intractable"
_ICLR.cc/2026/Conference — Submitted to ICLR 2026_

### Official Review · Reviewer_FTpY · 2025-10-28

**Soundness:** 2
**Presentation:** 1
**Contribution:** 2
**Rating:** 2
**Confidence:** 3

**Summary:**

The paper introduces a logic called $q\mathcal{L}$ that characterizes the expressivity of *quantized* Aggregate–Combine–Readout Graph Neural Networks (ACR GNNs), a widely used extension of vanilla message-passing GNNs for graph classification.

The authors use $q\mathcal{L}$ to show (co)NEXPTIME completeness of verification tasks on quantized ACR-GNNs via reduction to SAT/VALIDITY on $q\mathcal{L}$.

In their experiments, they illustrate on synthetic data and the protein–protein interaction (PPI) dataset that quantization does not significantly degrade the performance of models.

**Strengths:**

- **(S1) Problem Impact**: The paper addresses and seems to close an existing gap in the characterization of GNNs posed by [Sälzer et al., 2025](https://arxiv.org/abs/2502.16244), and provides a formalism that describes an important and widely used class of GNNs/graph properties.

- **(S2) Thorough analysis of $q\mathcal L$**:The investigation of the proposed logic is thorough, with examples highlighting its “reinterpretability” and compatibility with counting modal logic/description logic. While the main paper introduces $q\mathcal{L}$ as an extension of $\mathcal{L}\_{\text{quantGNN}}$ from [Sälzer et al., 2025](https://arxiv.org/abs/2502.16244) to add global aggregation, I personally like the additional connection to $K_♯$ [Nunn et al., 2024](https://arxiv.org/abs/2405.00205).

- **(S3) Self-Contained**: The paper provides ample explanation and background to follow the dense theoretical setting.

**Weaknesses:**

## W1: Presentation

The write-up of the paper is poor in grammar and clarity of language, with a few questionable slips, the most apparent being: “**NEXPTIME** is the class of problems decidable by a non-deterministic algorithm running in **polynomial time** in the size of its input,” where the authors clearly meant **exponential**.

The notation quietly changes after Section 3, where $\mathrm{agg}\_{g}$ suddenly becomes $\mathrm{agg}\_{\forall}$ henceforth.

Typesetting is strange, e.g., in lines 210–214, where the definition of the box operators is centered and the diamond operators are defined in line. After these definitions, the next sentence starts with a lower-case “and.”

In general, Section 3: *Simulating modal logic in $q\mathcal{L}$* is written in a confusing manner. The authors aim to show that they can re-express any formula with modal operators and connectives as an atomic formula by simulating these operators arithmetically. It is initially unclear why this is done (e.g., to simplify further proofs?).

The manuscript also uses up a lot of space on examples, but the central aspects---i.e., the ACR-GNN verification task definitions vt1, vt2, vt3, and the reason why these tasks are important in particular---are mentioned only in passing. As they are the central objects of investigation in the manuscript, this priority in levels of detail seems strange.

While the examples are helpful to follow the manuscript, they are used a lot, take up a lot of space, are partially a bit more complex than necessary (e.g., the example in lines 190–200), and sometimes replace proper definitions or explanations of concepts---for example, the definition of subexpressions $E(\phi)$ in Section 4.1.

## W2: Motivation and Experiments

I understand the motivation of the theoretical contribution, as global aggregation is commonly used in GNNs, and the authors do state that all ML models are quantized.

The experimental evaluation then goes on to demonstrate that quantization does not impact the performance of GNNs a lot. This setup seems largely irrelevant to the theoretical results and seems to undermine the motivation of the authors rather than support it. The experiments act as proof of concept of using quantized ACR GNNs in practice.
If the practicality of these models needs to be established, then it is questionable why they should be investigated from a theoretical point of view.

I think the authors could make a much stronger experimental statement by taking (pretrained) state-of-the-art models from existing literature on tasks that the community cares about, and stressing that **because** of similar performance, their quantized surrogates can be verified as a proxy. A study of how much speedup in terms of *verification time* versus the *loss in performance* would then provide more support for the theoretical setup, and could act as a guideline for future research by pointing out the value of verifying strongly compressed surrogate models.


## W3: Novelty

While the contribution seems to address a gap in existing research, the manuscript (including the structure) seems heavily inspired by [Sälzer et al., 2025](https://arxiv.org/abs/2502.16244), with adaptations to support global readout. As such, a lot of the submitted manuscript feels like it repeats information and does not focus enough on the novel contribution and motivation of the investigated verification settings.

## W4: Gap in Theoretical Results

The authors introduce $q\mathcal{L}$ to formalize the computations of ACR-GNNs, and show by example that any computation of the GNN can be expressed in the logic. However, it seems that a formal statement is missing that proves that the logic and the class of GNNs are, in fact, **equally** expressive.

**Questions:**

Of my listed weaknesses, I would appreciate a brief discussion of the authors to address my concerns in W2 and W4.

In addition, I have the following question to possibly expand on the contribution in the paper.

- **Q1 Simulation of global readout**: One unexplored aspect of ACR verification is a result of [Jogl et al., 2023](https://proceedings.neurips.cc/paper_files/paper/2023/file/ebf95a6f3c575322da15d4fd0fc2b3c8-Paper-Conference.pdf), which proposes simulating, e.g., an ACR-GNN with an AC-GNN without global readout by using a graph transformation on input graphs. As ACR-GNNs can then apparently be simulated from simpler-to-verify GNNs (“only” PSPACE-complete), the question arises why the difference arises, and whether a theoretical connection can be made. Does the translation of the formulas lead to a blowup of size?


---

## References

- **Sälzer, M., Schwarzentruber, F., & Troquard, N. (2025).** *Verifying Quantized Graph Neural Networks is PSPACE-complete.* IJCAI 2025. arXiv:2502.16244. <https://arxiv.org/abs/2502.16244>

- **Nunn, P., Sälzer, M., Schwarzentruber, F., & Troquard, N. (2024).** *A Logic for Reasoning About Aggregate–Combine Graph Neural Networks (K♯).* IJCAI 2024. arXiv:2405.00205. <https://arxiv.org/abs/2405.00205>

- **Jogl, F., Thiessen, M., & Gärtner, T. (2023).** *Expressivity-Preserving GNN Simulation.* NeurIPS 2023. <https://proceedings.neurips.cc/paper_files/paper/2023/file/ebf95a6f3c575322da15d4fd0fc2b3c8-Paper-Conference.pdf>

---

> ### Author Response · Authors · 2025-11-19
> **Author Response (1/2)**
>
> We would like to express our gratitude for providing a thoughtful review, given how time-consuming it can be!
>
> Especially, we appreciate the recognition of how this work closes gaps identified in earlier studies and our efforts to provide a self-contained and accessible presentation of the underlying theory.
>
> Below we will address your mentioned weaknesses and questions in detail!
> Feel free to ask follow-up questions if you feel it helps in a reevaluation of our submission.
>
> ---
>
> ## Comments on "Weaknesses"
>
> ### Presentation
>
> Generally, we gave the paper another thorough read and filtered out additional typos based on your feedback in the revised verions. Thanks for giving us this pointer and your attention to detail!
>
> >  “NEXPTIME is the class of problems [...] polynomial time in the size of its input,”
>
> Of course this is wrong. We fixed that and thanks for pointing this out!
>
> > in lines 210–214, where the definition of the box operators is centered and the diamond operators are defined in line. After these definitions, the next sentence starts with a lower-case “and.”
>
> We agree, the wording was misleading. We fixed that.
>
> > Simulating modal logic in
>  qL [...] It is initially unclear why this is done (e.g., to simplify further proofs?).
>
> Simulating modal logic is essential to justify that qL is indeed an effective language
> to express graph properties, particularly in terms of their verification.
> We utilise instances of these properties as motivations in Example 1 and Example 2.
> As modal logic supports the expression of graph properties, it is also crucial in the proofs.
>
>
> ### Motivation and experiments
> >The experimental evaluation then goes on to demonstrate that quantization does not impact the performance of GNNs a lot. This setup seems largely irrelevant to the theoretical results and seems to undermine the motivation of the authors rather than support it. The experiments act as proof of concept of using quantized ACR GNNs in practice. If the practicality of these models needs to be established, then it is questionable why they should be investigated from a theoretical point of view.
>
> We thank the reviewer for the comment. Our experimental goal is not to surpass existing GNNs but to show that quantization post-training keeps predictive accuracy while simplifying models. This aligns with our theoretical aim, suggesting quantized ACR-GNNs are practically significant and need to be considered from a verification perspective.
>
> However, your feedback highlights a need to clarify our experimental narrative. We have revised Section 1's "Outline" and parts of Section 8's Conclusion for better clarity:
>
> 1. We demonstrate in Section 7 the practicality of quantized GNNs, highlighting their verification relevance.
> 2. Appendix B presents small-scale experiments with a naive verification prototype, revealing immediate procedural limits (Table 2), confirming the theoretical complexity of the problem.
>
> In combination, these experiments aim to highlight that while quantised GNNs are of high relevance, they are in the need of thorough consideration by the verification community.
>
>
> > I think the authors could make a much stronger experimental statement by taking (pretrained) state-of-the-art models from existing literature on tasks that the community cares about, and stressing that because of similar performance, their quantized surrogates can be verified as a proxy. A study of how much speedup in terms of verification time versus the loss in performance would then provide more support for the theoretical setup, and could act as a guideline for future research by pointing out the value of verifying strongly compressed surrogate models.
>
> Thank you for the valuable suggestion regarding the use of state-of-the-art models. We plan to extend our experimental study in the future by analysing how architectural parameters, such as the selection of aggregation and readout functions, and the initialisation of the matrices \(C, A_1, A_2\) influence our results.
>
> Additionally, we agree that studying the trade-off between verification time and performance degradation would provide stronger empirical support for our theoretical results. We consider this a promising direction for future work.

---

> ### Author Response · Authors · 2025-11-19
> **Author Response (2/2)**
>
> ### Gap in theoretical results
>
> > The authors introduce qL
>  to formalize the computations of ACR-GNNs, and show by example that any computation of the GNN can be expressed in the logic. However, it seems that a formal statement is missing that proves that the logic qL and the class of GNNs are, in fact, equally expressive.
>
> Initially, we omitted a formal statement on equi-expressivity because it seemed unnecessary, given that efficiently expressing ACR-GNNs with qL (and vice-versa) follows is an immediate consequence of the design of qL and for specific instantiations of it, it folows a reasoning similar to that in works such as Sälzer et al. 2025, Nunn et al. 2024, and Barceló et al. "The Logical Expressiveness of Graph Neural Networks".
>
> However, we acknowledge that including a formal statement enhances the paper's self-containment. We have utilised the additional page to make the following adjustment:
> - We provide a formal statement prior to the example in Section 3, highlighting that the equivalence between qL and ACR-GNN results directly from the construction of qL.
> - We include references to proof sketches and complete proofs in Appendices C and D, demonstrating that an extension of the logic presented in Nunn et al. 2024, which is then an instantiation of the our "meta-logic" qL over with integer values and truncated-ReLU as activation functions, is equal expressive to the corresponding ACR-GNNs.
>
> This should help clearing formal concerns regarding the equal expressiveness statement.
> Thank you for pointing this out and helping us improve our submission!
>
> ## Questions
>
> > Simulation of global readout: One unexplored aspect of ACR verification is a result of Jogl et al., 2023, which proposes simulating, e.g., an ACR-GNN with an AC-GNN without global readout by using a graph transformation on input graphs. As ACR-GNNs can then apparently be simulated from simpler-to-verify GNNs (“only” PSPACE-complete), the question arises why the difference arises, and whether a theoretical connection can be made. Does the translation of the formulas lead to a blowup of size?
>
>
> This is an interesting question! The simulation would be done with an AC-GNN on the relation $V\times V$, and $agg_\forall$ is simply evaluated over $V\times V$. The logic and the formula would be the same. For inference with the GNN (deciding whether a GNN "accepts" a pointed graph), like for model checking a formula over a pointed graph, it does not change anything. It is easy to see that it is polynomial: just apply the definitions of inference and semantics.
> But one can still reduce an exponential tiling problem into the reasoning and verification tasks which involve *quantifications* over the set of pointed graphs.
> Satisfiability problem is more difficult than inference (aka model checking in the logic community) because we have to *search* for a graph where one "global readout" relation *must be* $V\times V$.

---

> ### Comment · Reviewer_FTpY · 2025-11-25
>
> I thank the authors for their detailed response and their efforts in addressing my concerns.
> Their newly uploaded manuscript addresses several of my points and improves the overall language in the manuscript.
> I will address how the points I raised have been addressed by the authors.
>
> **(W1) Presentation: Language improved.**
> The revised manuscript has corrected many typos and grammatical errors, although a few prominent ones remain.
> For example:
>  - 4.1. the definition/example for $E(\phi)$ has multiple unbalanced parenthesis (and might use $\coloneqq$ in a nonstandard way?)
>  - The heading of 4.2. reads QFBABA instead of QFBAPA
>  - in Line 161 "direct acyclic graphs"
> The number of mistakes overall has been noticeably reduced. I also notice that the notation has been improved, and the verification tasks are now presented more prominently.
> This addresses my concerns partially.
>
> **(W2) Motivation and Experiments: Experimental Evaluation thorough, but still seems orthogonal.**
> I thank the authors for their deliberations and their efforts in addressing my (and the other reviewers') concerns.
> I re-read the experimental section and reviewed the discussion with other reviewers, but unfortunately, I am still not convinced by the experimental evaluation for the following reasons.
>
> The experiments thoroughly investigate existing ACR-GNNs under post-training quantization, and note the following effects:
> - The accuracy stays approximately the same.
> - Quantization does not speed up inference.
> - Model size reduces when using quantization.
> among others.
> While these findings are certainly a nice-to-have, **the setup is largely inherited, the findings are expected, and the investigation does not support the theory at all**.
>
> I understand the authors' argument that the extremely high complexity of the task does not allow for any real experiment.
> But the presented investigation with the connected narrative feels like it just fills a page, with what could have been two sentences.
> Would it not be more interesting to investigate the fixed-size model checking tool?
>
> Or at least provide some "speculative result"? Perhaps this is not a reasonable direction, but an example like "If we compress to $n$ bits, this is how the accuracy behaves" could be useful. Then, a qualitative figure could be included, showing how stronger and stronger quantization can reduce (extrapolated or speculated) runtime as a rough factor compared to accuracy.
>
> This would at least permit the statement "normal GNNs are intractable to verify, but we can compress them down to $k(<8?)$
>  bits and maintain performance on their tasks".
> Then, some rough calculation of how many magnitudes of runtime can be saved by compressing the networks would round out the paper a bit.
>
> I personally think that the space used on the experiment would be better utilized in improving the visual clarity of the definitions in Section 3, elaborating more on the meaning of the verification tasks, etc.
> The experimental evaluation, in my opinion, provides only a weak argument to support the paper.
>
> **(W3) Novelty: A Lot of Space on known things, but more of a presentation issue; main results are solid improvement**:
> This ties in with my previous point, and, if I did not miss anything, has not been addressed by the authors. A significant portion of the paper is devoted to reiterating the background.
> I understand that many things are unavoidable to reiterate, but at the same time, a lot of space seems to be set aside for definitions and notation, yet it feels "crammed."
> I think this is more of a presentation issue rather than novelty; the theoretical gap seems to be handled soundly.
> The paper addresses a relevant gap in the existing body of work, which should justify novelty alone.
>
> **(W4) Gap in theoretical Results: fully addressed**: I thank the authors for addressing this point so directly.
> I consider this taken care of.
>
> **(Q1) Simulation of global readout:** I thank the authors for their answer, their explanation of why Jogl et al. does not reduce worst-case complexity addresses my question.
> This was more out of curiosity. I am not sure if this is worth mentioning in the paper, but I consider it taken care of.
>
>
> ### Verdict
>
> The authors put effort into their revision and tried to address my concerns.
> I believe the quality of the writing has improved; I feel that my concerns regarding their theory have been addressed.
> Unfortunately, I am still not convinced that the experimental evaluation helps the paper in any way. The findings do not appear to be relevant to the theoretical results. The presentation could be improved in a "big picture way" by making the visually dense definitions in section 3 easier to read.
>
> **I will increase my rating, with the main weakness being the experimental section.**

---

> ### Author Response · Authors · 2025-11-27
> **Author response (1/2)**
>
> Thank you very much for your thoughtful and detailed follow-up comment. We greatly appreciate your effort!
>
> Below, we address the topics W1 to W4, Q1 you mentioned in detail.
>
> ---
>
> ### W1 Presentation: Language improved.
>
> Thanks for pointing these out! We fixed them accordingly.
>
> ### W2 Motivation and Experiments: Experimental Evaluation thorough, but still seems orthogonal.
>
> > Or at least provide some "speculative result"? Perhaps this is not a reasonable direction, but an example like "If we compress to $n$ bits, this is how the accuracy behaves" could be useful. Then, a qualitative figure could be included, showing how stronger and stronger quantization can reduce (extrapolated or speculated) runtime as a rough factor compared to accuracy
>
> Thank you for this suggestion. Following your feedback, we extended our empirical study by conducting two additional sets of experiments to explore the relationship between bit-width reduction and model performance.
>
> We performed two sets of experiments:
>
> I. We applied the "fake quantization" (per-tensor symmetric) to the trained models. For these techniques, we used the following bit counts: [32, 16, 8, 4, 2].
>
> [Note.] More detailed: 32-bit configuration corresponds to the unmodified full-precision model and serves as the baseline. For the 8-bit configuration, we apply PyTorch’s dynamic post-training quantization (PTQ), which produces genuinely quantized Linear layers with int8 weights stored in the model and quantized kernels executed at runtime. For intermediate and lower precisions (16, 4, and 2 bits), we employ fake quantization: weights are quantized to the corresponding integer levels and then immediately dequantized back to float32. As a result, the model remains in float32 format, with no reduction in model size or inference time; only the accuracy is affected, allowing us to isolate the impact of precision on predictive performance. The quantization scheme was mentioned in the survey of Gholammi et. al. [1].
>
> [Note.] This information was added to the Appendix in line 2013 to 2020.
>
> Other parameters that were taken into account:
> - Number of layers from 1 to 10
> - Activation function:  ReLU, ReLU6, trReLU, GELU, Sigmoid, SiLU, Softplus, and ELU.
> - Classifiers: p1, p2, p3 (for the synthetic dataset only)
> - Splits. We run experiments across the splits we have by design.
>   - Train, Test1, Test2 for the synthetic data
>   - Train, Validation, and Test for the PPI data
>
> Evaluation was based on the __Accuracy__.
> [Note.] Because we are using this technique, the size and time don't change, since the model is still stored in Float32.
>
> [Important findings.]
> 1. [Synthetic data.] When bitwidths are 16 or 8, we do not lose accuracy across the model; more precisely, these results are very close to the baseline. After 8, we observed a dramatic decrease, so we prepared a second set of experiments.
>
> 2. [PPI data.] When bitwidths are 16 or 8, we do not lose accuracy across the model; In this case, we are not losing more than 1%.
>
>
> II. For the second set of experiments, we use [4,5,6,7,8]-bit "Fake Quantization" to the trained model before, with the parameters that we mentioned earlier, except for the number of layers; right now, we are using from 1 to 3.
>
>
> Key finding from this set is the next one:
> 1. Results are not equal for all activation functions. For this situation, we constructed a table that calculated the drop in accuracy per bit.
> 2. We applied additional selection criteria: look at from_bits > 4 (here we exclude results from 4 to 2 bits, where there is a significant drop) and set a threshold of 'lose more than 10/% of accuracy.'
> 3. [Synthetic data.] In this case, we observed a decrease in accuracy after 6-bit and 5-bit quantization for the $p_1$ classifier across all splits, activation functions, and numbers of layers.
> 3. [PPI data.] For this dataset, we did not observe a decrease in accuracy when applying the threshold of 'lose more than 10/% of accuracy.'
> We set the threshold to '-0.01'. Under this setting, the results became broader, and drops in accuracy were more frequently observed when reducing (from 7 to 6) and (from 6 to 5) bits in both Test and Validation splits.
>
>
> These extended evaluations strengthen understanding of how quantization affects ACR-GNN models and directly address to the reviewer’s request.
>
> We modified the paper and the appendix as follows:
>
> [Note.] Modifications were done in Section 7 from line 458 to 466.
> [Note.] Modifications were done in Appendix G from line 2009 to 2192 and from 2483 to 2699.
>
> (continued in next comment)

---

> ### Author Response · Authors · 2025-11-27
> **Author respomse (2/2)**
>
> (continuation of previous comment)
>
> To complete our quantization study, we evaluated ACR-GNNs under a range of bit-width configurations (32, 16, 8, 4, 2), and additionally under finer-grained reductions between 8 and 4 bits. Across both datasets, 16- and 8-bit models preserved accuracy within a small margin of the full-precision baseline. For the synthetic dataset (Tables 19 – 26 in Appendix G), notable accuracy drops emerged only below 8 bits, with consistent degradation from 6 to 5 and from 5 to 4 bit transitions across classifiers and activation functions. In contrast, the PPI dataset (Tables 44 – 51 in Appendix G) exhibits strong robustness to post-training quantization: performance is preserved down to 6 bits across all activation functions, with only minor deviations (less than 3%) observed in the from 7 to 6 and from 6 to 5 bit transitions. The only marked degradation appears when moving into the very low-precision range (5 bits and below).
>
> [Note.] In response to the reviewer’s suggestion to include a qualitative figure, we decided not to add the plot to the main paper because it would significantly impact readability. Instead, we provide the corresponding tables in the Appendix, and the complete set of plots is included in the accompanying Jupyter notebooks in the `supplementary_for_FTpY` directory.
>
>
> ### W3 Novelty: A Lot of Space on known things, but more of a presentation issue; main results are solid improvement:
>
> > [...]  a lot of space seems to be set aside for definitions and notation, yet it feels "crammed." I think this is more of a presentation issue rather than novelty;
>
> Thank you for the suggestions. Based on these, we made the following adjustments.
>
> - We restructured Section~2 to increase readability, adding paragraph environments.
> - We extensively restructured Section~3, which now follows the sequence: "introduce qL", "explain qL vs modal logic", "explain connection between qL and ACR-GNN". Also, we added an additional subsection for the last part. This should aid readability.
> - We restructured the simulating modal logic paragraphs.
> - We now provide a more detailed informal explanation of the verification task in Section~3.
> - Example~3 and Example~4 have been restructured.
> - We restructured the final paragraph in Section~3.
>
> Thanks again for highlighting these matters.
>
> ### W4 Gap in theoretical Results: fully addressed
>
> Thanks for recognising our improvements!
>
> ### Q1 Simulation of global readout:
> Thank you. We have decided to include this discussion as a footnote in the introduction.
>
> ---
>
> # **Reference**
> [1]. Gholami, A., Kim, S., Dong, Z., Yao, Z., Mahoney, M. W., & Keutzer, K. (2022). *A Survey of Quantization Methods for Efficient Neural Network Inference.* In *Low-Power Computer Vision* (pp. 291–326). Chapman and Hall/CRC.

---

### Official Review · Reviewer_SHhJ · 2025-10-31

**Soundness:** 3
**Presentation:** 2
**Contribution:** 3
**Rating:** 6
**Confidence:** 3

**Summary:**

In this article, the authors present new results about the expressivity of GNNs.
The authors consider a quantized variant of ReLU GNNs, meaning that the numbers used in computations all have a finite number of bits. The main contribution of the authors is to design a certain logic called $q\mathcal{L}$, to capture global readout.

The main contributions are:

- The logic $q\mathcal{L}$ is NEXPTIME, meaning that given, the language associated to $q\mathcal{L}$, then recognizing if a formula based on this language can be satisfied, can be performed in with a non-deterministic Turing machine, for some polynomial (n is a parameter measuring the size of the formula).

- Restricting to GNNs without global readout, by adapting their language $q\mathcal{L}$ to that case yields a PSPACE-complete version, meaning that (i) it is in PSPACE: one can solve the same problem as above in polynomial space, and (ii) any problem in PSPACE can be reduced polynomially to a problem of satisfiability of a formula of the qL version without global readout.

- Preliminary experiments on the impact of quantization on the performance and model size in practice

**Strengths:**

- The nature of the contribution is interesting and timely, in the line of research about the expressivity of GNNs.

- This article can be very interesting for logicians, while still be relevant for a more general audience.

- The illustrating experiments going along with the main theoretical contribution are interesting.

**Weaknesses:**

- While the paper claims potential implications for the safety of GNNs, the connection between the theoretical expressivity results and concrete safety aspects is not made explicit. Clarifying this link would significantly strengthen the paper’s broader impact.

- Given the technical nature of the paper, it would be helpful if the authors included a more accessible, high-level explanation of their main contributions for readers outside the logic community.

- Some imprecision, right from the beginning of the paper: please see first Question below.

**Questions:**

- It is said in first page (contributions): ``NEXPTIME is the class of problems decidable by a non-deterministic algorithm running in polynomial time in the size of its input''. Except if the size refers to the size of a compressed representation of the input string, this is incorrect to me. Rather, NEXPTIME should be the class of decision problems solvable by a non-deterministic Turing machine in exponential time, i.e., time $2^{p(n)}$ for some polynomial $p$. Please clarify.

- typo: l. 392: ``we introduction'' ->  we introduce

---

> ### Author Response · Authors · 2025-11-19
> **Author Response**
>
> Thank you for providing a thoughtful review!
>
> We are glad that the paper’s combination of theoretical expressivity analysis and illustrative experiments resonated,
> and agree that it can appeal both to logicians and to a broader audience interested in understanding GNN capabilities.
>
> We address your questions below. Please feel free to discuss any further aspects of our work with us.
>
> ---
>
> ## Questions
>
> > It is said in first page (contributions): ``NEXPTIME is the class of problems decidable by a non-deterministic algorithm
> running in polynomial time in the size of its input''. Except if the size refers to the size of a compressed representation of
> the input string, this is incorrect to me. Rather, NEXPTIME should be the class of decision problems solvable by a non-deterministic
> Turing machine in exponential time, i.e., time for some polynomial Please clarify.
>
> Indeed, you are correct, and we have corrected that sentence in the revised version. Thanks for pointing it out!
>
>
> > typo: l. 392: ``we introduction'' -> we introduce
>
> Thank you for noticing this error; we have corrected the typo.

---

> > ### Comment · Reviewer_SHhJ · 2025-11-24
> >
> > Thank you for the clarifications. I believe the contributions to be interesting, but I will maintain my score as is (borderline accept), please cf. list of comments in ``Weaknesses'' above.

---

> > > ### Author Response · Authors · 2025-11-27
> > > **Author response**
> > >
> > > Thank you for providing feedback on your position!
> > >
> > > We consider it worthwhile to address the weaknesses you mentioned. Please feel free to ask follow-up questions,
> > > as we are happy to engage in a discussion.
> > >
> > > ---
> > >
> > >
> > > > While the paper claims potential implications for the safety of GNNs, the connection between the theoretical expressivity results and concrete safety aspects is not made explicit. Clarifying this link would significantly strengthen the paper’s broader impact.
> > >
> > > We agree that improving our motivation is necessary and thank you for the feedback. To address this, we have added concrete examples of verification tasks (VT1 to VT3) to clarify the connection with safety issues. Please refer to Section 1 for the revised version of our paper for more details. Then, our overall contribution (roughly summarised) is that we present complexity results for these problems, highlighting the "difficulty" of ensuring them.
> > >
> > > > Given the technical nature of the paper, it would be helpful if the authors included a more accessible, high-level explanation of their main contributions for readers outside the logic community.
> > >
> > > We believe that the concrete examples introduced in the introduction provide clarity in illustrating the practical implications of our findings, particularly by explicitly connecting the theoretical expressivity results to safety concerns in GNNs.
> > >
> > > > Some imprecision, right from the beginning of the paper: please see first Question below.
> > >
> > > This has been revised to correctly reflect that NEXPTIME refers to the class of decision problems solvable by a non-deterministic Turing machine in exponential time, specifically $2^{p(n)}$ time for some polynomial $p(n)$, where $n$ is the size of the input. Thanks for pointing it out initially.

---

### Official Review · Reviewer_7D3e · 2025-11-01

**Soundness:** 3
**Presentation:** 3
**Contribution:** 3
**Rating:** 6
**Confidence:** 3

**Summary:**

This paper establishes that verifying **quantized Aggregate–Combine Graph Neural Networks with global Readout (ACR-GNNs)** is **(co)NEXPTIME-complete**, revealing that the verification of quantized GNNs is inherently intractable. To prove this, the authors introduce a logical language **qL**, which extends previous logics for GNN verification to handle **global readout** and **quantized arithmetic**. They provide reductions from qL to a quantized variant of **Quantifier-Free Boolean Algebra with Presburger Arithmetic (QFBAPA𝕂)**, showing decidability and tight complexity bounds. They also consider a bounded-graph setting where verification becomes **(co)NP-complete** and provide a proof-of-concept verifier. Finally, they present experimental evidence that **quantized ACR-GNNs maintain accuracy** while reducing model size and inference cost, supporting the practical utility of quantized models.

**Strengths:**

**Originality**
- The paper provides the **first complete logical characterization** and **tight complexity bounds** for verifying quantized GNNs with global readout.
- The use of **qL logic** to capture global readout within the quantized framework is an elegant and novel extension of prior formalisms like K♯ and FOC2.

**Quality**
- The proofs are rigorous, adapting known techniques (e.g., Hintikka sets and reductions to QFBAPA𝕂) to the quantized and global readout setting.
- The authors’ reasoning links theoretical intractability with practical implications, motivating further research into scalable verification strategies.
- The bounded-vertex relaxation and prototype implementation show a **constructive direction** for future research.

**Clarity**
- The structure is clear: the paper walks from formal definitions, through complexity proofs, to bounded relaxations and experiments.
- The motivating examples (e.g., verifying properties of “dog” graphs) effectively illustrate the semantics of qL.
- The appendix and code references enhance reproducibility.

**Significance**
- The (co)NEXPTIME-completeness result fills a major theoretical gap in GNN verification.
- The bounded-graph relaxation connects deep theory to **practical model checking**, potentially influencing future verification frameworks.

**Weaknesses:**

1. **Practical implications remain limited**
   - Although the theoretical contribution is strong, the **practical relevance** of (co)NEXPTIME results could be elaborated—how does this shape actual verification tool design?
   - The experiments, while informative, are **tangential** to the main verification focus.

2. **Experimental evaluation is lightweight**
   - The experiments only test quantization effects on model accuracy and size, not the **actual verification performance**.
   - The absence of benchmarks comparing **verification time** or **SMT encoding scalability** leaves open questions about the applicability of qL in realistic settings.

3. **Readout semantics assumption**
   - The fixed summation order assumption in global aggregation (noted in the limitations) is **non-standard in practice**, which might restrict the generality of the theoretical claims.

4. **Notation density and accessibility**
   - Some sections (e.g., Hintikka sets and reduction construction) are mathematically dense and could benefit from illustrative diagrams or examples of intermediate steps.

**Questions:**

1. **Verification tractability**
   - Given the (co)NEXPTIME-completeness, what verification techniques could still be practically feasible for small or structured graphs?
   - Can symbolic abstractions or over-approximations make qL-based verification tractable in practice?

2. **Extension to other architectures**
   - How would the complexity results change for other GNN architectures, such as recurrent or attention-based models?
   - Could qL be extended to handle continuous activations or message-passing schemes beyond summation?

3. **Bounded verification**
   - The bounded-vertex setting leads to (co)NP-completeness. Are there heuristics or practical solvers that can efficiently address this fragment?
   - How scalable is the provided proof-of-concept verifier when N grows beyond small graphs?

4. **Quantization modeling**
   - The assumption of saturating arithmetic in 𝕂 simplifies reasoning, but how would modular or IEEE-style rounding affect decidability or complexity?
   - Are there concrete examples where quantization introduces or removes counterexamples compared to unquantized GNNs?

5. **Experimental depth**
   - Can the authors provide **verification runtimes or success rates** for the bounded verifier on benchmark GNNs?
   - How does quantization influence the **verifiability** (not just accuracy) of ACR-GNNs in the experiments?

**Details Of Ethics Concerns:**

No ethics concerns.

---

> ### Author Response · Authors · 2025-11-19
> **Author Response (1/2)**
>
> We very much appreciate your comprehensive review!
>
> We thank you for recognizing both the theoretical and practical dimensions of our work:
> from the logical characterization and complexity results to the constructive bounded-graph
> relaxation and its implications for future verification frameworks.
>
> Below we will address all your detailed questions.
>
> ---
>
> ## Questions
>
> ### Verification tractability
> > Given the (co)NEXPTIME-completeness, what verification techniques could still be practically feasible for small or structured graphs?
>
> Thank you for the question! There are indeed possibilities. Specifically, in Section 6, where we consider bounding the number of
> neighbours. Under this assumption, as per Theorem 14, the problem is NP-complete. Although this remains challenging, it enables
> the use of modern SMT/SAT solvers. With extensive research backing these solvers and further efforts by the community to devise
> efficient encodings of the bounded verification problem, verifying quantised GNNs could become feasible, even for relatively
> large size instances. The same applies for bounding the size of a graph, by the way.
>
> > Can symbolic abstractions or over-approximations make qL-based verification tractable in practice?
>
> Yes, if we, for instance, turn to approximate methods, then we typically leave the realm of “formal” verification
> in the sense of sound and complete verification. Thus, the complexity bounds we derived generally do not apply. Especially,
> our hardness results  imply that this is a path future research should (continue to) pursue, as it is one way to avoid the intractability. Other ways are as discussed above, namely by adding additional assumptions to the verification problem.
>
>
> ### Extension to other architectures
> > How would the complexity results change for other GNN architectures, such as recurrent or attention-based models?
>
> We anticipate significant changes to the arguments used, as these models differ greatly. For instance, if we consider
> "recurrent GNNs," verification easily becomes undecidable due to that the recurrency potentially provides "Turing-complete power".
> For attention-based GNNs, we expect attention to function similarly to global readout; thus, we anticipate comparable results.
>
> However, due to the architectural differences, we must leave this as an avenue for future research, but a highly interesting one.
>
> > Could qL be extended to handle continuous activations or message-passing schemes beyond summation?
>
> Regarding continuous activations, the answer is yes. Since we assume a quantised setting, each such function
> essentially simplifies to a finite map. However, potential changes to the complexity require thorough study, although it is generally feasible and we do not expect significant change. Thank you for the interesting idea!
>
> For different message-passing schemes, the situation is less clear. Previous studies, such as [1,2], have shown
> that the expressive capabilities of GNNs using sum aggregation compared to ones utilising mean or max aggregation
> are incomparable. This largely depends on the assumptions, making it unclear how to incorporate these into the qL framework,
> which is tailored for summation-based GNNs. Nonetheless, this is intriguing and we believe it merits independent exploration.
>
> ### Bounded verification
> > The bounded-vertex setting leads to (co)NP-completeness. Are there heuristics or practical solvers that can efficiently address
> this fragment?
>
> Indeed! By utilising the boundedness assumption, the problem shifts to (co)NP-complete, as you noted. This facilitates the use of
> modern SMT/SAT solvers to approach this problem. The remaining challenge is for the community to devise efficient encodings of the
> problem to fully exploit the capabilities of these modern solvers. We believe that our can further motivate such endeavours
>
> > How scalable is the provided proof-of-concept verifier when N grows beyond small graphs?
>
> The prototype implementation lacks scalability. Its primary purpose is to highlight the inherent complexity
> of the problem and demonstrate that a naive approach, like this one, quickly reaches its limitations
> (see Figure 18, in particular the exponential growth). This underscores the necessity for substantial
> efforts to achieve tractable verification, as supported by our theoretical results.

---

> > ### Author Response · Authors · 2025-11-19
> > **Author Response (2/2)**
> >
> > ### Quantization modeling
> > > The assumption of saturating arithmetic in simplifies reasoning, but how would modular or IEEE-style rounding affect decidability or complexity?
> >
> > Yes, we think it remains NEXPTIME-complete. While this is not a formal proof, here is the argument: For the upper bound, Proposition 10 (l. 370) can be adapted to incorporate modular or IEEE-style rounding. The original
> > QFBAPA version includes a predicate $D \, \text{div} \, T$, indicating that $D$
> > divides $T$. In this version, the satisfiability problem is also in NP (see Kuncak & Rinard, 2007).
> > Only step 6 (proof of Proposition 10, l. 826) changes:
> > $\mathit{constraint}(B) = (|B| = k_B + M_B)   \text{ and }    2^n \textit{ div }  M_B$.
> >
> > For the lower bound (Theorem 14), considering we have $2n$ bits (l. 377), the bit
> > limit of $2^{2n}$ we need ensures that the saturation and modulo cases are the same. Hence, the lower bound
> > also holds for $K$ with modular arithmetic.
> >
> > > Are there concrete examples where quantization introduces or removes counterexamples compared to unquantized GNNs?
> >
> > There are several studies showing that while verifiers that assume unquantised feedforward neural networks (FNNs) state that a safety property is given, the real-world assumption of e.g. floating-point arithmetic in fact introduces unsafe behaviour. For this, see [4,5,6].
> > Since GNNs can simulate FNNs (assume an input graph consisting of a single node v, then the GNNs computation is essentially the application of an FNN to the label of v), the overall same findings apply for GNNs.
> >
> > Consider a graph neural network (GNN) $A$ that computes the expression $ReLU(8 - agg_\forall(x_1)) \geq 1$, where $x_1$ is a single feature. The specific structure of the graph is not crucial for this example; however, assume that $G$ is a graph with 10 vertices, each possessing a single feature with value $x_1 = 1$. If we set the feature space $K$ to be the set of integers $\mathbb{Z}$, then every node $v$ in $(G, v)$ is negatively classified by $A$, meaning $A(G, v) = 0$. Conversely, if the feature space $K$ is constrained to the range $[-8, \ldots, 7]$ (so using a very low number of bits and a saturating arithmetic), representing $8$ as $(-1) \times (-8)$, the outcome changes. In this scenario, every node $v$ in $(G, v)$ is positively classified by $A$, yielding $A(G, v) \geq 1$.
> >
> >
> > ### Experimental depth
> > > Can the authors provide verification runtimes or success rates for the bounded verifier on benchmark GNNs?
> >
> > Thanks for the question! We agree that such results would indeed be interesting, however, there are significant challenges:
> > - We have not yet seen widely adopted "benchmark GNNs". We agree that creating such benchmarks of GNN verification tasks is an important next step for the community.
> > - Benchmark datasets, such as Citeseer and similar ones, are typically too large for our naive verifier, as discussed in the previous question.
> >
> > However, this once again highlights that achieving practical formal verification requires substantial effort, as indicated by our results.
> >
> > > How does quantization influence the verifiability (not just accuracy) of ACR-GNNs in the experiments?
> >
> > In general, the verification of ACR-GNNs is undecidable, as noted in [3]. Despite high complexity, quantisation
> > is an assumption that makes verification decidable, thus making it possible, which is a significant improvement. Besides this, your question also touches on an important
> > point: how do different quantisation methods affect the verifiability or even safety of GNNs? This remains an open question, as we currently lack efficient verifiers. But, we concur that addressing this is essential for future research.
> >
> > ---
> > [1] Keyulu Xu, Weihua Hu, Jure Leskovec, Stefanie Jegelka:
> > How Powerful are Graph Neural Networks? ICLR 2019
> >
> > [2] Eran Rosenbluth, Jan Tönshoff, Martin Grohe:
> > Some Might Say All You Need Is Sum. IJCAI 2023: 4172-4179
> >
> > [3] Marco Sälzer, Martin Lange:
> > Fundamental Limits in Formal Verification of Message-Passing Neural Networks. ICLR 2023
> >
> > [4] Kai Jia, Martin C. Rinard:
> > Exploiting Verified Neural Networks via Floating Point Numerical Error. SAS 2021: 191-205
> >
> > [5] Dániel Zombori, Balázs Bánhelyi, Tibor Csendes, István Megyeri, Márk Jelasity:
> > Fooling a Complete Neural Network Verifier. ICLR 2021
> >
> > [6] Attila Szász, Balázs Bánhelyi, Márk Jelasity:
> > No Soundness in the Real World: On the Challenges of the Verification of Deployed Neural Networks. arXiv (2025)

---

### Official Review · Reviewer_3rCn · 2025-11-12

**Soundness:** 3
**Presentation:** 3
**Contribution:** 2
**Rating:** 6
**Confidence:** 3

**Summary:**

This paper narrows its scope to a critical yet underaddressed gap in GNN verification: the theoretical complexity of quantized Aggregate-Combine Graph Neural Networks with global readout (ACR-GNNs). While "neural network verification is intractable" is a broad community consensus, this work focuses explicitly on GNN-specific structures—namely, global readout (a core component for graph-level tasks like molecule classification or protein interaction prediction)—and quantized arithmetic (standard in real-world deployments). It introduces the logical language qL to formalize ACR-GNN computations and graph properties, proves that ACR-GNN verification tasks (sufficiency, necessity, consistency) are (co)NEXPTIME-complete, and contrasts this with the PSPACE-completeness of readout-free quantized GNNs. Complementing theory, the paper validates that quantized ACR-GNNs retain high accuracy ($\pm$1% drop) with 60-74% size reduction and proposes a bounded-vertex relaxation (NP/coNP-complete) for practical verification.

**Strengths:**

- Before this paper, the community knew readout-free quantized GNN verification was PSPACE-complete, but readout’s impact was speculative. By proving (co)NEXPTIME-completeness, the work quantifies this impact: global readout pushes complexity into a higher class, meaning verification becomes exponentially harder with increasing input size. This clarity prevents wasted effort.

- The paper’s (co)NP-complete bounded-vertex relaxation is not a random heuristic but a principled response to the (co)NEXPTIME result: by limiting counterexamples to graphs with N vertices, it leverages the boundary between "unbounded intractability" and "bounded tractability." This may provide a roadmap for future work—e.g., optimizing N selection, or combining bounded verification with domain-specific constraints (e.g., molecule graphs have ≤100 atoms)—that would be impossible without knowing the exact point at which intractability sets in.

**Weaknesses:**

- The theoretical analysis and experiments focus exclusively on summation for local/global aggregation. However, other modern GNNs use max, mean, or attention-based aggregation—readout for these variants may introduce different complexity patterns (e.g., max aggregation reduces dependency between distant vertices). The paper’s failure to extend bounds to non-summation aggregation limits its relevance to a narrow subset of industrial GNNs.

- Extensive experiments validate quantized ACR-GNNs' accuracy/lightweight design but do not link to core verification challenges, feeling tangential to the paper’s central claim.

- While technically rigorous, the (co)NEXPTIME-completeness result reinforces a known trend (readout exacerbates complexity) rather than offering a transformative insight, limiting broader field impact.

**Questions:**

- For practical applications (e.g., molecular graph verification), how do you recommend choosing the maximum number of vertices N? Is there a way to determine N such that if no counterexample exists for N, it is unlikely to exist for larger graphs?

---

> ### Author Response · Authors · 2025-11-19
> **Author Response**
>
> Thank you very much for your thoughtful review!
>
> Especially, we appreciate that you precisely summarised the core message of the paper: clarifying the complexity jump introduced by global readout and recognizing the bounded-vertex relaxation as a principled bridge between theoretical intractability and practical feasibility.
>
> We address your concerns in detail below.
>
> ---
>
> ### Comments on "Weaknesses"
>
> > The theoretical analysis and experiments focus exclusively on summation for local/global aggregation. However, other modern GNNs use max, mean, or attention-based aggregation—readout [...].
>
> Thank you for pointing this out! We acknowledge that there are other GNN models, such as those employing different
> aggregation mechanisms like max and mean. However, prior studies, for example, [1,2], demonstrate that the expressive
> capabilities of these are different, sometimes incomparable, and require substantial study on their own.
>
> One of the primary contributions of our submission is to address a gap in the complexity landscape of verification problems
> in quantised settings. Hence, we choose to centre this study on the common types of GNNs considered in this exact research area,
> namely, those with sum aggregation.
>
> Nonetheless, we concur that your suggestions warrant consideration in future work.
>
>
> > Extensive experiments validate quantized ACR-GNNs' accuracy/lightweight design but do not link to core verification challenges, feeling tangential to the paper’s central claim.
>
> We appreciate your feedback, which indicates that our experimental setup's narrative needs clarification. In response, we have revised
> the "Outline" in Section 1 and parts of the Conclusion in Section 8 to improve the clarity of interpretation:
>
> 1. As noted, we conduct extensive experiments (Section 7) to highlight the practical performance of quantised GNNs, emphasising their
> relevance and the need for consideration by the verification community.
> 2. We conduct small-scale experiments (Appendix B) with a prototype implementation of the naive verification procedure, immediately
> encountering a limit (see Table 2). This result, while expected, underscores the practical implication of our theoretical
> findings: the problem is inherently hard.
>
> In summary, while quantised GNNs are practically relevant (1.), verification remains challenging
> (as indicated by theoretical results + 2.).
>
> ### Questions
>
> > For practical applications (e.g., molecular graph verification), how do you recommend choosing the maximum number of vertices N?
>
> In future work, for example, we aim to utilize molecular data, such as the PyTorch Geometric
> M9 Dataset. In such cases, the domain imposes a natural limit on the number of nodes, this should be utilised initially.
>
> Otherwise, the maximum number of vertices $N$ will mainly be determined by the computational bounds of the SMT solver and the
> structure of the activation function employed in the neural network. Future advancements in SMT solvers and SMT encoding techniques
> will likely increase this limit. But we expect that for most datasets a natural bound is clear from context.
>
> > Is there a way to determine N such that if no counterexample exists for N, it is unlikely to exist for larger graphs?
>
> Thank you for the intriguing question. For this, one should establish a small- or bounded-model property for the ACR-GNNs. At the moment, we cannot provide an answer. On the other hand, if it exists, given the theoretical complexity of the problem, this bound will be very large and not useful in practice.
>
>
> ---
> [1] Keyulu Xu, Weihua Hu, Jure Leskovec, Stefanie Jegelka:
> How Powerful are Graph Neural Networks? ICLR 2019
>
> [2] Eran Rosenbluth, Jan Tönshoff, Martin Grohe:
> Some Might Say All You Need Is Sum. IJCAI 2023: 4172-4179

---

### Author Response · Authors · 2025-12-03
**rebuttal summary**

Dear Area Chair,

we would like to thank all reviewers and (former) ACs for their work. To guide the further process, here is a summary of the rebuttal:

---

**[3rCn (score 6)](https://openreview.net/forum?id=7s0VmieMcS&noteId=hfEpXwWzdL):** appreciated that the paper precisely quantifies the impact of global readout on the complexity and provides a bounded-vertex relaxation to guide practical verification; their concerns focused on the narrow scope, the connection between experiments and verification, and novelty for the broader field.

**[our response](https://openreview.net/forum?id=7s0VmieMcS&noteId=0vzdrAMDkW):** we mentioned that our somewhat narrow focus reflects the established setting in the relevant literature and we revised the introduction and conclusion to better connect the experiments to the verification settings

---

**[7D3e (score 6)](https://openreview.net/forum?id=7s0VmieMcS&noteId=b4DVnf86Yc):** praised the paper’s novel logical characterization, rigorous complexity results, and clear presentation; their main concerns were the practical evaluation, the narrow scope, and the absence of verification-focused experiments

**[our response](https://openreview.net/forum?id=7s0VmieMcS&noteId=MgGfHXrXgM):** we emphasised feasible techniques for restricted, practical cases, expected behaviour for alternative architectures, limits of bounded verification, and how different quantization models preserve the complexity

---

**[SHhJ (score 6)](https://openreview.net/forum?id=7s0VmieMcS&noteId=KcB3xCw1uj):** found the contribution timely and broadly relevant, with clear appeal to both logicians and a wider audience; their concerns focused on the connection to general safety, and the need for a more accessible high-level presentation

**[our response](https://openreview.net/forum?id=7s0VmieMcS&noteId=vPCDFUPidk):** we addressed the reviewer’s concerns by adding concrete verification tasks in the introduction to make the connection to sgeneral safety explicit and by improving the high-level explanation of our contributions

---

**[FTpY (score 2)](https://openreview.net/forum?id=7s0VmieMcS&noteId=0fHG0DYYiX):** praised the paper’s impact, thorough logical analysis, and self-contained presentation; raised concerns about clarity, motivation, experimental relevance, the degree of novelty, and the absence of a expressiveness-equivalence statement.

**[our response](https://openreview.net/forum?id=7s0VmieMcS&noteId=FtSc4CSeQ6):** we clarified the purpose of some definitions, revised the exposition and corrected notation/typos, strengthened the introduction and conclusion to better explain how the experiments support the relevance of quantized models for verification; we also added a formal expressiveness statement, including pointers to proof sketches and appendices.

This initiated a follow-up discussion between the authors and FTpY, where **FTpY mentioned the that they raise their score (see [here](https://openreview.net/forum?id=7s0VmieMcS&noteId=PFQxQPxVKk))**.

---

### Meta-Review · Area_Chair_qWja · 2026-01-06

**Summary:**

This paper studies the theoretical complexity of verification for quantized Aggregate-Combine Graph Neural Networks with global readout (ACR-GNNs) and concludes that the verification problem is (co)NEXPTIME-complete and thus intractable. The paper also shows a prototype with bounded-vertex relaxation for verification and the empirical performance of quantized ACR-GNNs.

Reviewers commonly agree that this work makes good theoretical contributions that fill a gap in the theoretical complexity of GNN verification. However, there are several major concerns: 1) The experiments are largely irrelevant to the theory and central claims; 2) The significance of yet another work on showing the intractability of verification for a particular kind of models is limited and has minimal practical implications on developing useful verification techniques. The AC believes these issues have not been effectively resolved. Overall, the AC feels this work is below the acceptance threshold.

**Reviewer Concerns:**

Major concerns that have not been addressed effectively: 1) The experiments are largely irrelevant to the theory and central claims; 2) The significance of yet another work on showing the intractability of verification for a particular kind of models is limited and has minimal practical implications on developing useful verification techniques.

The authors have addressed some concerns in the presentation and made some clarifications. There were initially concerns on the novelty as a large proportion of the paper seems to be reiterating Sälzer et al., 2025, but the reviewer acknowledged that this is more a presentation issue rather than novelty.

**Reviewer Scores:**

Reviewer 3rCn and Reviewer 7D3e initially rated 6 and are likely to maintain their ratings.

Reviewer SHhJ also initially rated 6 and explicitly mentioned to maintain their rating.

Reviewer FTpY initially rated 2 and explicitly mentioned that they would increase their rating, but major concerns on the experiments remain. The AC estimates that the reviewer may increase their rating from 2 to 4.

---

### Decision · Program_Chairs · 2026-01-26

Reject